# Conservative Contextual Bandits: Beyond Linear Representations

**Rohan Deb**
University of Illinois
Urbana-Champaign
rd22@illinois.edu

**Mohammad Ghavamzadeh**
Amazon AGI
ghavamza@amazon.com

**Arindam Banerjee**
University of Illinois
Urbana-Champaign
arindamb@illinois.edu

## ABSTRACT

Conservative Contextual Bandits (CCBs) address safety in sequential decision making by requiring that an agent's policy, along with minimizing regret, also satisfies a safety constraint: the performance is not worse than a baseline policy (e.g., the policy that the company has in production) by more than $(1 + \alpha)$ factor. Prior work developed UCB-style algorithms for this problem in the multi-armed (Wu et al., 2016) and contextual linear (Kazerouni et al., 2017) settings. However, in practice the cost of the arms is often a non-linear function, and therefore existing UCB algorithms are ineffective in such settings. In this paper, we consider CCBs beyond the linear case and develop two algorithms `C-SquareCB` and `C-FastCB`, using Inverse Gap Weighting (IGW) based exploration and an online regression oracle. We show that the safety constraint is satisfied with high probability and that the regret for `C-SquareCB` is sub-linear in horizon $T$, while the regret for `C-FastCB` is first-order and is sub-linear in $L^*$, the cumulative loss of the optimal policy. Subsequently, we use a neural network for function approximation and online gradient descent as the regression oracle to provide $\tilde{\mathcal{O}}(\sqrt{KT} + K/\alpha)$ and $\tilde{\mathcal{O}}(\sqrt{KL^*} + K(1 + 1/\alpha))$ regret bounds respectively. Finally, we demonstrate the efficacy of our algorithms on real world data, and show that they significantly outperform the existing baseline while maintaining the performance guarantee.

## 1 INTRODUCTION

Contextual bandits provide a framework to make sequential decisions over time by actively interacting with the environment. In each time step, the learner observes $K$ context vectors associated with corresponding arms, selects an arm based on the history of interaction and observes the corresponding noise corrupted cost[1] of playing that arm. The objective of the learner is to minimize the cumulative sum of costs over the entire horizon of length $T$, or equivalently to minimize the *regret*. Although a lot of progress had been made in the multi-armed (Auer et al., 2002; Agrawal & Goyal, 2012; Bubeck et al., 2012; Bubeck & Slivkins, 2012) and linear formulation (Chu et al., 2011; Abbasi-Yadkori et al., 2011; Agrawal & Goyal, 2013), until recently solutions for the general non-linear cost function did not exist. A series of work on neural contextual bandits (Zahavy & Mannor, 2020; Zhou et al., 2020; Zhang et al., 2021) have provided algorithms and guarantees for general non-linear cost functions, paving the way for practical use of bandit algorithms in real-world problems. Distinct from the previous set of works, Foster & Rakhlin (2020) and Foster & Krishnamurthy (2021) developed general reductions from the bandit problem to online regression using the Inverse Gap Weighting (IGW) idea (Abe & Long, 1999; Abe et al., 2003). This reduction works for general cost functions and uses only a mild realizability assumption (see Assumption 1).

In addition to non-linear cost functions, safety is another crucial consideration that significantly enhances the practical use of these algorithms in real-world. In this work, we consider a specific notion of safety called *safety with respect to a baseline* (Kazerouni et al., 2017). Algorithms that are safe, meaning it is assured to perform at least as well as an established (possibly already deployed) baseline, are more likely to be used in practice. While existing online algorithms for bandits are expected to eventually identify an optimal or high-performing policy, their performance during the

---

[1]We use the cost formulation instead of the more common reward formulation in this paper.

initial learning phase can be unpredictable and often unsafe. To ensure safety in such algorithms, it is important to regulate their exploration, by making them more *conservative*. This is done by making sure that the cumulative cost of the algorithm at any stage is not worse than that of the baseline by more than a $(1 + \alpha)$ factor (cf. Definition 2.2). Such a conservative bandit formulation has been studied in the multi-armed setting (Wu et al., 2016) and the contextual linear setting (Kazerouni et al., 2017; Garcelon et al., 2020), but algorithms and regret guarantees for the general case do not exist.

Existing conservative bandit algorithms in Wu et al. (2016); Kazerouni et al. (2017); Garcelon et al. (2020) have considered standard multi-armed bandits and linear contextual bandits, using a suitable variant of the popular Upper Confidence Bound (UCB) approach. In this work, we are interested in Conservative Contextual Bandits (CCBs) beyond the linear case. One simple and lazy way to extend the analysis to general non-linear functions would be to modify the Neural UCB algorithm (Zhou et al., 2020), and extend the regret analysis to the conservative setup. However, a recent work (Deb et al., 2024a) has shown that the regret bound for Neural UCB in Zhou et al. (2020) (and Neural Thompson Sampling Zhang et al. (2021)) that depends on the efective dimension $\tilde{d}$, is $\Omega(T)$ in the worst case even with an oblivious adversary. This also extends to any modification for the conservative case, and therefore we avoid this approach.

In this paper, we consider CCBs with general functions and make the following contributions. First, as our main contribution, under the assumption of an online regression oracle for such general functions, we propose CCB algorithms utilizing such regression oracle and doing exploration using inverse gap weighting (IGW) (Abe & Long, 1999; Foster & Rakhlin, 2020; Foster & Krishnamurthy, 2021). The regret of our proposed algorithms, respectively based on squared loss and KL-loss regression (Sections 3 and 4), can be expressed in terms of the regret of the corresponding regression oracle, while ensuring that the conservative performance guarantee is not violated with high probability. Our analysis differs substantially from the standard UCB based analysis, since our algorithms do not maintain high confidence sets around the true cost functions, which is challenging for general functions. Our analysis also differs from the standard IGW analysis as the proposed CCB algorithms have to guarantee the safety constraint by a careful balance between actions chosen based on IGW exploration and using the baseline algorithm. Second, we instantiate the proposed CCB algorithms by using online neural regression, leverage $O(\log T)$ regret for neural regression with both square-loss and KL-loss, and provide regret bounds for CCBs with neural networks (Section 5). A more detailed description of existing works leading up to the current work can be found in Appendix A.

Next we summarize our specific technical contributions below:

1. **Reduction using Squared loss:** We provide an algorithm for conservative bandits for general cost functions using an oracle for online regression with squared loss (see Algorithm 1). We subsequently prove a $\mathcal{O}(\sqrt{KT \, \texttt{Reg}_{\texttt{Sq}}(T)} + K\texttt{Reg}_{\texttt{Sq}}(T)/\alpha)$ regret bound, where $\texttt{Reg}_{\texttt{Sq}}(T)$ is the regret of online regression with squared loss, and also ensure that the performance constraint is satisfied in high probability (see Theorem 3.1).

2. **Reduction using KL loss:** Next, we provide an algorithm using an oracle for online regression with KL loss (see Algorithm 2) and prove a $\mathcal{O}(\sqrt{KL^* \log(L^*) \, \texttt{Reg}_{\texttt{KL}}(T)} + K \, \texttt{Reg}_{\texttt{KL}}(T)(1+1/\alpha)$ regret bound. Here, $\texttt{Reg}_{\texttt{KL}}(T)$ is the regret of online regression with KL loss and $L^*$ is the cumulative cost of the optimal policy, while ensuring that the performance constraint is satisfied in high probability (see Theorem 4.1). This is a *first order* regret bound and is data-dependent in the sense that it scales with the cumulative cost of the best policy $L^*$, instead of the horizon length $T$.

3. **Regret Bounds using Neural Networks:** We instantiate the online regression oracle with Online Gradient Descent (OGD) and the function approximator with a feed-forward neural network to give an end-to-end regret bound of $\mathcal{O}\big(\sqrt{KT \, \log(T)} + K \log(T)/\alpha\big)$ for Algorithm 1 (Theorem 5.1) and $\mathcal{O}\big(\sqrt{KL^* \log(L^*) \, \log(T)} + K \log T + K \log(T)/\alpha\big)$ for Algorithm 2 (Theorem 5.2).

4. **Experiments:** Finally, we compare our proposed algorithms with existing baselines for conservative bandits and show that our algorithms consistently perform better (see Section 6).

## 2 PROBLEM FORMULATION

**Contextual Bandits:** We consider a contextual bandit problem where a learner needs to make sequential decisions over $T$ time steps. At any round $t \in [T]$, the learner observes the context for $K$ arms $\mathcal{X}_t = \{\mathbf{x}_{t,1}, , ..., \mathbf{x}_{t,K}\} \subseteq \mathbb{R}^d$, where the contexts can be chosen adversarially unlike in Agarwal et al. (2014); Simchi-Levi & Xu (2020); Ban et al. (2022) where the contexts are chosen i.i.d. from

a fixed distribution. The learner chooses an arm $a_t \in [K]$ and then the associated cost of the arm $y_{t,a_t} \in [0,1]$ is observed. We make the following assumption on the cost.

**Assumption 1** (**Realizability**). *The conditional expectation of $y_{t,a}$ given $\mathbf{x}_{t,a}$ is given by some $h \in \mathcal{H}$, where $\mathcal{H}$ is the function class such that $h : \mathbb{R}^d \mapsto [0,1]$, i.e., $\mathbb{E}[y_{t,a}|\mathbf{x}_{t,a}] = h(\mathbf{x}_{t,a})$. Further, the context vectors satisfy $\|\mathbf{x}_{t,a}\| \leq 1$, $t \in [T], a \in [K]$.*

**Definition 2.1** (**Regret**). *The learner's goal is to minimize the regret, defined as the expected difference between the cumulative cost of the algorithm and that of the optimal policy:*

$$\texttt{Reg}_{\texttt{CB}}(T) = \mathbb{E}\Big[ \sum_{t=1}^{T} \big( y_{t,a_t} - y_{t,a_t^*} \big) \Big] = \sum_{t=1}^{T} \big( h(\mathbf{x}_{t,a_t}) - h(\mathbf{x}_{t,a_t^*}) \big) \ , \tag{1}$$

*where $a_t^* = \mathrm{argmin}_{a \in [K]} h(\mathbf{x}_{t,a})$, minimizes the expected cost in round $t$. The subscript $\texttt{CB}$ stands for Contextual Bandits and subsequently differentiates it from the regret of online regression.*

**Conservative Contextual Bandits:** There exists a baseline policy $\pi_b$ that at each round $t$, selects action $b_t \in [K]$ and receives the expected cost $h(\mathbf{x}_{t,b_t})$. This baseline policy is to be interpreted as the default or status quo policy that the company follows and knows to provide a reasonable performance. However, the company wants to improve the policy but at the same time not incur a high cost while trying to do so. Thus, it insists on the following performance constraint on any algorithm:

**Definition 2.2** (**Performance Constraint**). *At each round $t$, the cumulative loss of the agent's policy should remain below $(1 + \alpha)$ times the cumulative loss of the baseline policy for some $\alpha > 0$, i.e.,*

$$\sum_{i=1}^{t} h(\mathbf{x}_{i,a_i}) \leq (1 + \alpha) \sum_{i=1}^{t} h(\mathbf{x}_{i,b_i}) \ , \ \forall t \in \{1, \ldots, T\}. \tag{2}$$

The parameter $\alpha > 0$ controls how conservative the agent has to be with respect to the baseline policy. When $\alpha$ is very small, the cumulative loss by the agent's policy cannot be very large in comparison to baseline cumulative loss and as $\alpha$ is increased the agent can take larger risks to explore more. We assume that the expected costs of the actions taken by the baseline policy, $h(\mathbf{x}_{t,b_t})$, are known. This is a reasonable assumption as argued in Kazerouni et al. (2017); Garcelon et al. (2020), since we usually have access to a large amount of data generated by the baseline policy as this is the default strategy of the company. We can also relax this to the assumption that we have an un-biased estimate of the baseline cost and modify our algorithms slightly (see Appendix E).

Next, we make the following assumption on the baseline gap and the costs of the baseline actions.

**Assumption 2** (**Baseline Gap and Cost Bounds**). *Let $\Delta_{t,b_t} := h(\mathbf{x}_{t,b_t}) - h(\mathbf{x}_{t,a_t^*})$ be the baseline gap. There exist $0 \leq \Delta_l \leq \Delta_h$ and $0 < y_l < y_h$, such that for all $t \in [T]$, we have*

$$\Delta_l \leq \Delta_{t,b_t} \leq \Delta_h \quad and \quad y_l \leq y_{t,b_t} \leq y_h.$$

The assumption ensures a minimum level of performance by the baseline action and is standard in conservative bandits. Assumption 3 in both Kazerouni et al. 2017 and Garcelon et al. (2020) are exactly as Assumption 2 in this work, while the regret bound provided in Theorem 2 of Wu et al. (2016) implicitly depends on similar quantities.

## 3 REDUCTION TO ONLINE REGRESSION WITH SQUARED LOSS

In this section, we develop an algorithm for Conservative Bandits with general output functions by reducing it to a black-box online regression oracle with squared loss. In Section 5, we instantiate the oracle by online gradient descent and give end-to-end regret guarantees. Before proceeding to the algorithm, we briefly describe the online regression formulation below. For a more detailed treatment, see Hazan (2021); Shalev-Shwartz (2012); Bubeck (2011).

**Online Regression with Squared Loss:** We assume access to an oracle $\texttt{Sq-Alg}$ that takes as input all data points until time $t - 1$, $\mathcal{D}_{t-1} = \{(\mathbf{x}_{i,a_i}, y_{i,a_i}) : 1 \leq i \leq t - 1\}$ and makes the prediction $\hat{y}_{t,a} = \texttt{Sq-Alg}(\mathcal{D}_{t-1}, \mathbf{x}_{t,a})$ in $[0,1]$ for input $\mathbf{x}_{t,a}$ at time $t$. We further make the following assumption on the regret incurred by the oracle $\texttt{Sq-Alg}$:

**Assumption 3** (**Online Regression Regret for Squared Loss**). *The regret of the online regression oracle $\texttt{Sq-Alg}$ is bounded by $\texttt{Reg}_{\texttt{Sq}}(T) \geq 1$, i.e.,*

$$\sum_{t=1}^{T} \ell_{\texttt{sq}}(\hat{y}_{t,a_t}, y_{t,a_t}) - \inf_{g \in \mathcal{H}} \sum_{t=1}^{T} \ell_{\texttt{sq}}(g(\mathbf{x}_{t,a_t}), y_{t,a_t}) \leq \texttt{Reg}_{\texttt{Sq}}(T), \tag{3}$$

*where the squared loss is given by $\ell_{\texttt{sq}}(\hat{y}_{t,a_t}, y_{t,a_t}) = (\hat{y}_{t,a_t} - y_{t,a_t})^2$.*

---

**Algorithm 1** Conservative SquareCB (`C-SquareCB`)

---

1: **Input:** $\alpha$
2: **Hyper-parameter:** Exploration parameter $\gamma_t$
3: **Initialize:** $\mathcal{S}_0 = \emptyset$, and let $m_0 = 0, m_t := |\mathcal{S}_t|, t \in [T]$
4: **for** $t = 1, \ldots, T$ **do**
5:      Receive contexts $\mathbf{x}_{t,1}, \ldots, \mathbf{x}_{t,K}$ and compute $\hat{y}_{t,k}, \ \forall k \in [K]$ using `Sq-Alg`
6:      Let $z_t = \underset{a \in [K]}{\arg\min} \ \hat{y}_{t,a}$, and compute

$$p_{t,a} = \frac{1}{K + \gamma_t(\hat{y}_{t,a} - \hat{y}_{t,z_t})}, \ \forall k \in [K] \setminus \{z_t\}; \quad p_{t,z_t} = 1 - \sum_{a \neq z_t} p_{t,a} \ .$$

7:      Sample $\tilde{a}_t \sim p_t$
8:      **if** the *safety condition* in (4) is satisfied **then**
9:          Play the IGW action $a_t = \tilde{a}_t$ and observe output $y_{t,a_t}$
10:        Set $\mathcal{S}_t = \mathcal{S}_{t-1} \cup t, \ \ \mathcal{S}_t^c = \mathcal{S}_{t-1}^c$
11:        Set $\mathcal{D}_t = \mathcal{D}_{t-1} \cup \{(\mathbf{x}_{t,a_t}, y_{t,a_t})\}$ and update the oracle `Sq-Alg`
12:      **else**
13:        Play $a_t = b_t$ and observe output $h(\mathbf{x}_{t,b_t})$
14:        Set $\mathcal{S}_t = \mathcal{S}_{t-1}, \ \ \mathcal{S}_t^c = \mathcal{S}_{t-1}^c \cup t, \ \ \mathcal{D}_{t+1} = \mathcal{D}_t$

---

We refer to our algorithm as `C-SquareCB`, whose pseudo-code is reported in Algorithm 1. At a high level, `C-SquareCB` does the following: **1)** It samples an action from the IGW distribution using the outputs of the oracle `Sq-Alg`, **2)** It then verifies if a certain *safety condition* is met, **3)** If yes, it then plays the sampled action, otherwise turns conservative and plays the baseline action. We use $\mathcal{S}_t \subseteq [T]$ and its complement $\mathcal{S}_t^c \subseteq [T]$ to denote the subsets containing the time-steps until round $t$ when the IGW and baseline actions were played, respectively. We denote the cardinality of these sets by $m_t = |\mathcal{S}_t|$ and $n_t = |\mathcal{S}_t^c|$.

At every round $t$, the agent receives $K$ contexts $\mathbf{x}_{t,1}, \ldots, \mathbf{x}_{t,K}$ and estimates the cost for every arm $\hat{y}_{t,a}$ using the online regression oracle (line 5). It then finds the arm with the lowest estimate $z_t$ (see line 6) and computes the Inverse Gap Weighted (IGW) distribution using the estimate gaps $\hat{y}_{t,a} - \hat{y}_{t,z_t}$ and the exploration parameter $\gamma_t$. Next it samples a candidate action $\tilde{a}_t$ in line 7 and verifies a *safety condition* in line 7 (corresponding to (2)) by checking if the following inequality holds:

$$\underbrace{\hat{y}_{t,\tilde{a}_t} + \sum_{i \in \mathcal{S}_{t-1}} \sum_{a \in [K]} p_{i,a} \hat{y}_{i,a}}_{(A)} + \underbrace{\sum_{i \in \mathcal{S}_{t-1}^c} h(\mathbf{x}_{i,b_i})}_{(B)} + \underbrace{16 \sqrt{m_{t-1}\left( \texttt{Reg}_{\texttt{Sq}}(m_{t-1}) + \log(4/\delta) \right)}}_{(C)}$$
$$\leq (1+\alpha)\sum_{i=1}^{t} h(\mathbf{x}_{i,b_i}). \tag{4}$$

Here, term $(A)$ sums up the expected costs of the regression oracle for all rounds when the IGW action was played and the cost of the current IGW action $\tilde{a}_t$ under consideration. Term $(B)$ simply sums up the baseline costs for all the rounds when the baseline action was played. To ensure that the performance constraint (2) is not violated, in our proof we show that (see proof of Lemma 4)

$$(A) \ - \sum_{i \in \mathcal{S}_{t-1} \cup \{t\}} h(\mathbf{x}_{i,a_i}) \geq -16\sqrt{m_{t-1}(\texttt{Reg}_{\texttt{Sq}}(m_{t-1}) + \log(4/\delta))}.$$

Observe that now term $(C)$ compensates for the above gap and immediately implies that the constraint in (2) is satisfied. Note that an easy way to ensure that (2) holds would be to replace $(A)$ with the observed costs $y_{t,a_t}$ and use Azuma-Hoeffding to bound $\sum_{i \in \mathcal{S}_{t-1}} (y_{i,a_i} - h(\mathbf{x}_{i,a_i}))$. However this approach does not let us control the number of times the baseline action is played by the algorithm, which is crucial to bound the final regret (see Step 2 in the *proof of Theorem* 3.1). If the *safety condition* in (4) is satisfied, then the IGW action $a_t = \tilde{a}_t$ is played and the output $y_{t,a_t}$ is observed in line 9. The current time step is added to $\mathcal{S}_t$ and the current input-output pair $(\mathbf{x}_{t,a_t}, y_{t,a_t})$ is added to the online regression dataset $\mathcal{D}_t$ (lines 10 and 11). Otherwise we play the baseline action $b_t$ in

line 13, observe the true output $h(\mathbf{x}_{t,b_t})$ and add the current time step to $\mathcal{S}_t^c$ in line 14. We now state the main theoretical result of this section that bounds the regret of C-SquareCB (Algorithm 1) along with satisfying the performance constraint in (2) in high probability.

**Theorem 3.1** (**Regret Bound for C-SquareCB**). *Suppose Assumptions 1,2 and 3 hold. With probability at least $1-\delta$, C-SquareCB (Algorithm 1) with $\gamma_t = \sqrt{K|\mathcal{S}_t|}/(\text{Reg}_{\text{Sq}}(m_T) + \log(8\delta^{-1}))$ satisfies the performance constraint in (2) and has the following regret bound:*

$$\text{Reg}_{\text{CB}}(T) = \mathcal{O}\bigg( \underbrace{\sqrt{KT}\Big(\sqrt{\text{Reg}_{\text{Sq}}(T)} + \sqrt{\log(8\delta^{-1})}\Big)}_{I} + \underbrace{\frac{K(\text{Reg}_{\text{Sq}}(T) + \log(8\delta^{-1}))}{\alpha y_l(\Delta_l + \alpha y_l)}}_{II} \bigg). \quad (5)$$

**Remark 3.1** (**Term interpretations**). Term $I$ and $II$ in (5) correspond to the regret of playing the IGW and baseline actions, respectively. Note that term $II$ grows with $\text{Reg}_{\text{Sq}}(T)$, unlike the linear case where the second term is independent of the horizon $T$ (see Theorem 5 in Kazerouni et al. 2017). However, in Section 5, when we instantiate the oracle with OGD and the function approximator with a neural network, $\text{Reg}_{\text{Sq}}(T)$ only contributes a $\log T$ factor to the regret to the second term.

**Remark 3.2** (**Infinite actions**). The regret in (5) scales with the number of actions $K$, and thus, holds for finite number of actions. In case of infinite actions, a straightforward extension of our results following the analysis of Theorem 1 in Foster et al. (2020) will lead to a regret that scales with the dimension of the action space instead of $K$.

*Proof of Theorem 3.1* The proof of the theorem follows along the following steps. We report the proof of the intermediate lemmas in Appendix B.

1. **Regret Decomposition:** We begin by decomposing the regret in (1) into two parts following Kazerouni et al. (2017): the regret accumulated by playing the IGW and baseline actions, terms $I$ and $II$ in the regret bound (5), respectively.

   **Lemma 3.1.** *Let Assumptions 1 and 2 hold. Then, the regret defined in (1) can be bounded as*

   $$\text{Reg}_{\text{CB}}(T) \leq \sum_{t \in \mathcal{S}_T} \Big( h(\mathbf{x}_{t,a_t}) - h(\mathbf{x}_{t,a_t^*}) \Big) + n_T \Delta_h, \quad (6)$$

   *where the set $\mathcal{S}_T$ consists of the rounds until the horizon $T$ when C-SquareCB played an IGW action and $n_T = |\mathcal{S}_T^c|$ is the number of times until $T$ where a baseline action was played.*

2. **Upper Bound on $n_T$:** The *safety condition* in (4) determines how many times the baseline action is played. In what follows, we use $m_t := |\mathcal{S}_t|$ and $\tau := \max\{1 \leq t \leq T : a_t = b_t\}$, i.e., the last time step at which C-SquareCB played an action according to the baseline strategy.

   (a) The following lemma upper-bounds $n_T$ in terms of $m_\tau$ and $\text{Reg}_{\text{Sq}}(m_{\tau-1})$.

   **Lemma 3.2.** *Suppose Assumption 1,2 and 3 holds. Then, with probability $1-\delta/4$ the number of times the baseline action is played by C-SquareCB is bounded as*

   $$n_T \leq \frac{1}{\alpha y_l}\bigg\{ -(m_{\tau-1}+1)(\Delta_l + \alpha y_l) + 64\sqrt{K}\sqrt{(m_{\tau-1}+1)}\Big(\sqrt{\text{Reg}_{\text{Sq}}(T)} + \sqrt{\log(8\delta^{-1})}\Big) \bigg\}. \quad (7)$$

   (b) Note that the second term in (7) grows as $\sqrt{m_{\tau-1}}$ and the first term decreases linearly in $m_\tau$, and therefore, one can find the maximum and further bound $n_T$ as in the following lemma.

   **Lemma 3.3.** *Suppose Assumption 1,2 and 3 holds. Then, with probability $1-\delta/4$ the number of times the baseline action is played by C-SquareCB is bounded as follows:*

   $$n_T \leq \mathcal{O}\left( \frac{K(\text{Reg}_{\text{Sq}}(T) + \log(8\delta^{-1}))}{\alpha y_l(\Delta_l + \alpha y_l)} \right). \quad (8)$$

3. **Bounding the Final Regret:** The first term in (6) can be bounded along the lines of the analysis in (Foster & Rakhlin, 2020). Note that $\mathcal{D}_T$ only contains the input-output pairs at time steps when the IGW action was picked, i.e., all $t \in \mathcal{S}_T$, and therefore, using $m_T = |\mathcal{S}_T|$, (3) reduces to

   $$\sum_{t \in \mathcal{S}_T} (\hat{y}_{t,a_t} - y_{t,a_t})^2 - \inf_{g \in \mathcal{H}} \sum_{t \in \mathcal{S}_T} \big( g(\mathbf{x}_{t,a_t}) - y_{t,a_t} \big)^2 \leq \text{Reg}_{\text{Sq}}(m_T). \quad (9)$$

However, unlike Foster & Rakhlin (2020), we need an a time varying exploration parameter $\gamma_t$ that depends on the size of $\mathcal{S}_t$ for all $t \in [T]$ in order to bound $n_T$ in Step 2. The next lemma bounds the regret of the first term in (6) with an such adaptive $\gamma_t$.

**Lemma 3.4.** *Suppose Assumptions 1 and 3 hold. Then, for $\delta > 0$ and $\gamma_t = \sqrt{K|\mathcal{S}_t|/(\texttt{Reg}_{\texttt{Sq}}(T) + \log(4\delta^{-1}))}$, with probability $1 - \delta/4$, C-SquareCB guarantees*

$$\sum_{t \in \mathcal{S}_T} \left( h(\mathbf{x}_{t,a_t}) - h(\mathbf{x}_{t,a_t^*}) \right) \leq \mathcal{O}\left( \sqrt{Km_T \texttt{Reg}_{\texttt{Sq}}(T)} + \sqrt{Km_T \log(8\delta^{-1})} \right). \quad (10)$$

Note that $m_T \leq T$. Combining (6), (8), and (10), and taking a union bound over the high probability events shows that the regret bound in (5) holds with probability $1 - \delta/2$.

4. **Performance Constraint:** Finally the following lemma shows that the condition in Line 7 of C-SquareCB ensures that the *performance constraint* in (2) is satisfied.

**Lemma 3.5.** *Let Assumptions 1, 2 and 3 hold. Then, for $\delta > 0$ and $\gamma_t = \sqrt{K|\mathcal{S}_t|/(\texttt{Reg}_{\texttt{Sq}}(m_T) + \log(8\delta^{-1}))}$, with probability $1 - \delta/2$, C-SquareCB satisfies the performance constraint in (2).*

Taking a union bound with the high probability regret bound in Step 3, we have that with probability $1 - \delta$, C-SquareCB simultaneously satisfies the performance constraint in (2) and the regret upper-bound in (5), which concludes the proof. $\quad\square$

**Remark 3.3** (**Bounding baseline regret**). The analysis in Foster & Rakhlin (2020) does not have a safety condition and therefore our analysis bounding $n_T$ (the number of times the baseline action is played) in Step 2 and the performance constraint satisfaction in Step 4 of proof of Theorem 3.1 are original contributions. One of the important parts of the analysis involves bounding $n_T$, the number of times the baseline actions are played. In the linear case (Kazerouni et al., 2017), the analysis crucially uses the upper and lower confidence bounds for the parameter estimates. For general function classes it is difficult to maintain such confidence bounds around estimates, and further the estimates from the regression oracle $\hat{y}_{t,a_t}$ do not provide any such guarantees. Therefore our analysis crucially relates $n_T$ to squared loss and through that gives a reduction to online regression.

**Remark 3.4** (**Time dependent Exploration**). The analysis from Foster & Rakhlin (2020) cannot be directly used to bound the regret for the time steps when the IGW actions were picked (term $I$ in (5)). This is because we need to carefully choose a time dependent exploration parameter $\gamma_t$, to simultaneously ensure that term $I$ is $\sqrt{T}$ while ensuring that $n_T$ is small. In the process, we extend the analysis in Foster & Rakhlin (2020) to time-dependent $\gamma_t$ and bound the regret in $I$. $\quad\square$

## 4 FIRST ORDER REGRET BOUND WITH LOG LOSS

In this section, we use an oracle with KL loss, KL-Alg, and provide a reduction from the conservative contextual bandit (CCB) problem to online regression. The objective of this reduction is to provide a *first order* data dependent[2] regret bound, i.e., a bound that scales with $L^* = \sum_{t=1}^{T} L^*(t)$, where $L^*(t) = h(\mathbf{x}_{t,a_t^*})$ is the cost of the optimal action at time $t$. Note that $L^* \leq T$, since $h(\mathbf{x}) \in [0, 1]$ for all $\mathbf{x}$, but in practice we may have $L^* \ll T$.

**Online Regression with KL Loss:** We assume access to an oracle KL-Alg that takes as input all data points until time $t - 1$, $\mathcal{D}_{t-1} = \{(\mathbf{x}_{i,a_i}, y_{i.a_i}) : 1 \leq i \leq t - 1\}$ and makes the prediction $\hat{y}_{t,a} = $ KL-Alg$(\mathcal{D}_{t-1}, \mathbf{x}_{t,a})$ in $[0, 1]$ for input $\mathbf{x}_{t,a}$ at time $t$. We further make the following assumption on the regret incurred by the oracle KL-Alg:

**Assumption 4** (**Online Regression Regret for KL Loss**). *The regret of the online regression oracle KL-Alg is bounded by $\texttt{Reg}_{\texttt{KL}}(T) \geq 1$, i.e.,*

$$\sum_{t=1}^{T} \ell_{\text{KL}}(\hat{y}_{t,a_t}, y_{t,a_t}) - \inf_{g \in \mathcal{H}} \sum_{t=1}^{T} \ell_{\text{KL}}\left( g(\mathbf{x}_{t,a_t}), y_{t,a_t} \right) \leq \texttt{Reg}_{\texttt{KL}}(T), \quad (11)$$

*where the KL loss is given by $\ell_{\text{KL}}(\hat{y}, y) = y \log(1/\hat{y}) + (1 - y) \log(1/(1 - \hat{y}))$.*

---

[2]See Appendix A for more details on Data Dependent Bounds

---

**Algorithm 2** Conservative FastCB (`C-FastCB`)

---
1: **Input:** $\alpha$
2: **Hyper-parameter:** Exploration parameter $\gamma_t$
3: **Initialize:** $\mathcal{S}_0 = \emptyset$
4: **for** $t = 1, \dots, T$ **do**
5:     Receive contexts $\mathbf{x}_{t,1}, \dots, \mathbf{x}_{t,K}$ and compute $\hat{y}_{t,k}, \ \forall k \in [K]$ using `KL-Alg`
6:     Let $z_t = \operatorname*{argmin}_{k \in [K]} \hat{y}_{t,k}$ and compute

$$p_{t,k} = \frac{\hat{y}_{t,z_t}}{K\hat{y}_{t,z_t} + \gamma_t(\hat{y}_{t,k} - \hat{y}_{t,z_t})} \forall k \in [K] \setminus \{z_t\}; \quad p_{t,z_t} = 1 - \sum_{a \neq z_t} p_{t,a}$$

7:     Sample $\tilde{a}_t \sim p_t$
8:     **if**

$$\hat{y}_{t,\tilde{a}_t} + \sum_{i \in \mathcal{S}_{t-1}} \sum_{a \in [K]} p_{i,a} \hat{y}_{i,a} + \sum_{i \in \mathcal{S}_{t-1}^c} h(\mathbf{x}_{i,b_i}) + 16\sqrt{m_{t-1}\mathtt{Reg_{KL}}(T)} \leq (1+\alpha)\sum_{i=1}^{t} h(\mathbf{x}_{i,b_t})$$

    **then**
9:         Play $a_t = \tilde{a}_t$ and observe output $y_{t,a_t}$
10:        Set $\mathcal{S}_t = \mathcal{S}_{t-1} \cup t, \ \mathcal{S}_t^c = \mathcal{S}_{t-1}^c$
11:        Set $\mathcal{D}_t = \mathcal{D}_{t-1} \cup \{(\mathbf{x}_{t,a_t}, r_{t,a_t})\}$ and update the oracle `KL-Alg`
12:     **else**
13:        Play $a_t = b_t$ and observe output $h(\mathbf{x}_{t,b_t})$
14:        Set $\mathcal{S}_t = \mathcal{S}_{t-1}, \ \mathcal{S}_t^c = \mathcal{S}_{t-1}^c \cup t, \ \mathcal{D}_{t+1} = \mathcal{D}_t$

---

We refer to the resulting algorithm as `C-FastCB`. It follows the same structure as `C-SquareCB` (Algorithm 1) and is summarized in Algorithm 2. We now state the main theory of this section that bounds the regret of `C-FastCB` along with satisfying the performance constraint in high probability.

**Theorem 4.1** (**Regret Bound for C-FastCB**). *Let Assumptions 1, 2 and 4 hold. With probability $1 - \delta$, `C-FastCB` (Algorithm 2) with $\gamma_i$ chosen in ($\gamma_i$-`Schedule`), satisfies the performance constraint in (2) and has the following bound on the expected regret (expectation is for the action distributions):*

$$\mathbb{E}\left[\mathtt{Reg_{CB}}(T)\right] = \mathcal{O}\left(\sqrt{KL^* \log(L^*) \, \mathtt{Reg_{KL}}(T)} + \frac{K\mathtt{Reg_{KL}}(T)}{\alpha y_l(\Delta_l + \alpha y_l)} \log\left(\frac{e\sqrt{K\mathtt{Reg_{KL}}(T)}}{\Delta_l + \alpha y_l}\right)\right). \quad (12)$$

**Remark 4.1** (**First Order Regret**). Note that the regret in (12) depends on $\sqrt{L^*}$ instead of $\sqrt{T}$, where $L^* = \sum_{t=1}^{T} L^*(i)$ is the cumulative loss of the optimal policy and depends on the complexity of the bandit instance, $L^* \ll T$, thus improving the performance of the learner. Such a data dependent regret is referred to as a *first-order regret* (Agarwal et al., 2017a; Foster & Krishnamurthy, 2021).

**Remark 4.2** (**Challenges**). We face similar set of challenges as in Theorem 5.1 in trying to bound $n_T$, and our analysis relates $n_T$ to the KL loss using the sampling strategy and reduces it to online regression with KL loss. We face an additional challenge. In Foster & Krishnamurthy (2021), the exploration parameter $\gamma_t$ is set to a fixed value $\gamma = \max(\sqrt{KL^*/3\mathtt{Reg_{KL}}(T)}, 10K)$. In our analysis we need a time dependent $\gamma_t$ to ensure that we can bound the regret contributed by both the IGW and baseline actions (cf. decomposition in (6)). However, unlike in Algorithm 1, we crucially need to set $\gamma_t$ in an episodic manner to ensure that the final regret does not have a $\sqrt{T}$ dependence. By having $\log(L^*)$ episodes and keeping $\gamma_t$ constant within an episode, we derive our final regret in (12), in which term $I$ has only an additional $\sqrt{\log(L^*)}$ factor. A more detailed description of the exact choice of $\gamma_t$ along with the episodic analysis has been pushed to Appendix C, for clarity.

*Proof of Theorem 4.1.* The proof broadly follows the same sequence of steps as in the proof of Theorem 3.1, and owing to limited space, has been reported in Appendix C. □

## 5 NEURAL CONSERVATIVE BANDITS

In this section, we instantiate the online regression oracles `Sq-Alg` (Algorithm 1) and `KL-Alg` (Algorithm 2) by (projected) Online Gradient Descent (OGD), and use feed-forward neural networks for function approximation. The setup closely follows the one in Deb et al. (2024a), which we restate it here for completeness. We consider a feed-forward neural network whose output is given by

$$f(\theta_t; \mathbf{x}) := m^{-1/2} \mathbf{v}_t^\top \phi(m^{-1/2} W_t^{(L)} \phi(\cdots \phi(m^{-1/2} W_t^{(1)} \mathbf{x}) \cdots)), \tag{13}$$

where $L$ is the number of hidden layers and $m$ is the width of the network. Further, $W_t^{(1)} \in \mathbb{R}^{m \times d}$ and $W_t^{(l)} = [w_{t,i,j}^{(l)}] \in \mathbb{R}^{m \times m}$ for all $l \in \{2, \dots, L\}$ are layer-wise weight matrices, and $\mathbf{v}_t \in \mathbb{R}^m$ is the last layer vector. Similar to Du et al. (2019); Banerjee et al. (2023), we consider a (point-wise) smooth and Lipschitz activation function $\phi(\cdot)$. We define $\theta_t \in \mathbb{R}^p$, where $\theta_t := (\mathbf{vec}(W_t^{(1)})^\top, \dots, \mathbf{vec}(W_t^{(L)})^\top, \mathbf{v}^\top)^\top$, as the vector of all parameters in the network, and make the following assumption on the initialization of the network (Liu et al., 2020; Banerjee et al., 2023).

**Assumption 5.** *We initialize $\theta_0$ with $w_{0,ij}^{(l)} \sim \mathcal{N}(0, \sigma_0^2)$ for $l \in [L]$, where $\sigma_0 = \frac{\sigma_1}{2\left(1 + \frac{\sqrt{\log m}}{\sqrt{2m}}\right)}, \sigma_1 > 0$, and $\mathbf{v}_0$ is a random unit vector with $\|\mathbf{v}_0\|_2 = 1$.*

Next, we define the Neural Tangent Kernel (NTK) matrix (Jacot et al., 2018) at $\theta$ as $K_{\text{ntk}}(\theta) := [\langle \nabla f(\theta; \mathbf{x}_t), \nabla f(\theta; \mathbf{x}_{t'}) \rangle] \in \mathbb{R}^{T \times T}$, and make the following assumption on this matrix which is common in the deep learning literature (Du et al., 2019; Arora et al., 2019; Cao & Gu, 2019). Note that our NTK is defined for a specific sequence of $\mathbf{x}_t$'s where $\mathbf{x}_t$ depends on the choice of arms played, and our assumption on the NTK matrix is for all sequences, which is equivalent to the assumption for the $(TK \times TK)$ NTK matrix as in Zhou et al. (2020); Zhang et al. (2021).

**Assumption 6.** *The matrix $K_{\text{ntk}}(\theta_0)$ is positive definite, i.e., $K_{\text{ntk}}(\theta_0) \succeq \lambda_0 \mathbb{I}$ for some $\lambda_0 > 0$.*

The assumption can be ensured if no two context vectors $\mathbf{x}_t$ overlap. Note that this assumption is used by all existing regret bounds for neural bandits (see Assumption 4.2 in Zhou et al. 2020, Assumption 3.4 in Zhang et al. 2021, Assumption 5.1 in Ban et al. 2022 and Assumption 5 in Deb et al. 2024a). The choice of the width of the network $m$ depends on $\lambda_0$ and is similar to the width requirements in Zhou et al. (2020) and Zhang et al. (2021).

We define a perturbed network as in Deb et al. (2024a) as follows:

$$\tilde{f}(\theta_t, \mathbf{x}_t, \boldsymbol{\varepsilon}) = f(\theta_t; \mathbf{x}_t) + c_p \sum_{j=1}^p \frac{(\theta_t - \theta_0)^T e_j \varepsilon_j}{m^{1/4}}, \tag{14}$$

where $\{e_j\}_{j=1}^p$ are standard basis vectors, $\boldsymbol{\varepsilon} = (\varepsilon_1, \dots, \varepsilon_p)^T$ is an i.i.d. random Rademacher vector, i.e., $P(\varepsilon_j = +1) = P(\varepsilon_j = -1) = 1/2$, and $c_p$ is the perturbation constant. As in Deb et al. (2024a), we use an ensemble of $S = \mathcal{O}(\log T)$ random networks as follows:

$$\tilde{f}^{(S)}\left(\theta; \mathbf{x}_t, \boldsymbol{\varepsilon}^{(1:S)}\right) = \frac{1}{S} \sum_{s=1}^S \tilde{f}(\theta; \mathbf{x}_t, \boldsymbol{\varepsilon}_s), \tag{15}$$

where each $\boldsymbol{\varepsilon}_s$ is a Rademacher vector. We run projected OGD on the loss function

$$\mathcal{L}_{\text{Sq}}^{(S)}\left(y_t, \{\tilde{f}(\theta; \mathbf{x}_t, \boldsymbol{\varepsilon}_s)\}_{s=1}^S\right) := \frac{1}{S} \sum_{s=1}^S \ell_{\text{Sq}}\left(y_t, \tilde{f}(\theta; \mathbf{x}_t, \boldsymbol{\varepsilon}_s)\right), \tag{16}$$

which with the projection operator $\prod_B(\theta) = \arginf_{\theta' \in B} \|\theta' - \theta\|_2$ gives us the following update:

$$\theta_{t+1} = \prod_B \left(\theta_t - \eta_t \nabla \mathcal{L}_{\text{Sq}}^{(S)}\left(y_{t,a_t}, \{\tilde{f}(\theta; \mathbf{x}_{t,a_t}, \boldsymbol{\varepsilon}_s)\}_{s=1}^S\right)\right). \tag{17}$$

We now prove a regret bound for `C-SquareCB` with feed-forward neural networks (neural `C-SquareCB`) and OGD as a regression oracle.

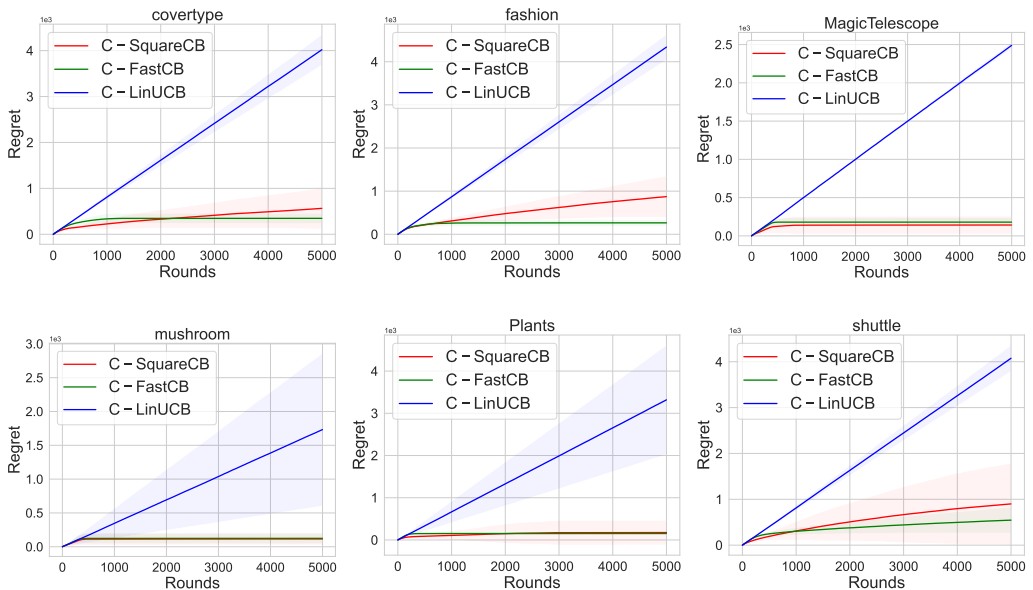

Figure 1: Comparison of cumulative regret of `C-SquareCB` and `C-FastCB` with the baseline `C-LinUCB` on openml datasets (averaged over 10 runs).

**Theorem 5.1** (**Regret bound for Neural `C-SquareCB`**). *We instantiate* `Sq-Alg` *with the predictor* $\hat{y}_{t,a_t} = \tilde{f}^{(S)}\left(\theta; \mathbf{x}_t, \boldsymbol{\varepsilon}^{(1:S)}\right)$ *from* (15) *and update the parameters using OGD in* (17). *Under Assumptions 1,2, 5 and 6 with* $\gamma_t$ *as in Theorem 5.1, step-size sequence* $\{\eta_t\}$, *width* $m$, *perturbation constant* $c_p$, *and projection ball B, with high probability* $(1 - \mathcal{O}(\delta))$, *the performance constraint in* (2) *is satisfied by* `C-SquareCB` *and the regret is given by*

$$\texttt{Reg}_{\text{CB}}(T) \leq \mathcal{O}\left(\sqrt{KT\log T} + \sqrt{KT\log(16\delta^{-1})} + \frac{K(\log T + \log(16\delta^{-1}))}{\alpha y_l(\Delta_l + \alpha y_l)}\right).$$

Next, for the *first-order* bound, we use the following ensembled network as the predictor:

$$\sigma\left(\tilde{f}^{(S)}\left(\theta; \mathbf{x}_t, \boldsymbol{\varepsilon}^{(1:S)}\right)\right) = \frac{1}{S}\sum_{s=1}^{S}\sigma(\tilde{f}(\theta; \mathbf{x}_t, \boldsymbol{\varepsilon}_s)) \tag{18}$$

where $\tilde{f}(\theta; \mathbf{x}_t, \boldsymbol{\varepsilon}_s)$ is as defined in (14) and $\sigma(\cdot)$ is the sigmoid function. Our next theorem provides a first-order regret bound for `C-FastCB` when coupled with feed-forward networks and OGD.

**Theorem 5.2** (**Regret bound for Neural `C-FastCB`**). *We instantiate* `Sq-Alg` *with the predictor* $\hat{y}_{t,a_t} = \tilde{f}^{(S)}\left(\theta; \mathbf{x}_t, \boldsymbol{\varepsilon}^{(1:S)}\right)$ *from* (15) *and update the parameters using OGD in* (17). *Under Assumptions 1,2, 4, 5 and 6 with* $\gamma_t$ *chosen as in* ($\gamma_{\texttt{i}}$-`Schedule`), *step-size sequence* $\{\eta_t\}$, *width* $m$, *perturbation constant* $c_p$, *and projection ball B, with probability* $(1 - \mathcal{O}(\delta))$, *the performance constraint in* (2) *is satisfied by* `C-FastCB` *and the expected regret is given by*

$$\mathbb{E}\,\texttt{Reg}_{\text{CB}}(T) \leq \mathcal{O}\left(\sqrt{KL^*\log L^*\log T} + K\log T + \frac{K\log T}{\alpha y_l(\Delta_l + \alpha y_l)}\right).$$

## 6 EXPERIMENTS

We evaluate our algorithms `C-SquareCB` and `C-FastCB` and compare the regret bounds with the existing baseline - Conservative Linear UCB (`C-LinUCB`) (Kazerouni et al., 2017). The algorithm estimates the parameter associated with the cost function using least squares regression and uses existing results on high probability confidence bounds around the estimate (Abbasi-Yadkori et al., 2011) to set up a safety condition. When the safety condition is satisfied, it plays actions according to Linear UCB (Chu et al., 2011; Abbasi-Yadkori et al., 2011), otherwise switches to the baseline action. We tune the ridge parameter $\lambda$ in $\{0.001, 0.005, 0.01, 0.05, 0.1\}$.

Figure 2: Comparison of Percentage of Constraints violated by `C-SquareCB` and `C-FastCB` with their vanilla non conservative versions on openml datasets (averaged over 100 runs).

We use the evaluation setting for bandit algorithms developed in Bietti et al. (2021) and subsequently used in Zhou et al. (2020); Zhang et al. (2021); Ban et al. (2022); Deb et al. (2024a). We consider a series of multiclass classification problems from the openml.org platform. We transform each $d$-dimensional input into $K$ different context vectors of dimension $dK$, where $K$ is the number of classes as follows: $\mathbf{x}_{t,1} = (\mathbf{x}_t, \mathbf{0}, \mathbf{0}, \ldots, \mathbf{0})^T$, $\mathbf{x}_{t,2} = (\mathbf{0}, \mathbf{x}_t, \mathbf{0}, \ldots, \mathbf{0})^T), \ldots, \mathbf{x}_{t,K} = (\mathbf{0}, \mathbf{0}, \ldots, \mathbf{0}, \mathbf{x}_t)^T$. The $K$ vectors correspond to the $K$ different action choices in the bandit problem. We assign a cost of $1$ to all the context vectors associated with the incorrect classes, and a cost of $0.01$ to the correct class. Note that when an action corresponding to an incorrect class is selected, the learner does not learn the identity of the action with the lowest cost. For each of the datasets, we fix one action as the baseline action, and the baseline policy corresponds to always choosing this pre-defined action.

`C-SquareCB` and `C-FastCB` use a two layer neural network with ReLU activation and width 100. We update the network parameter every 10-th round and do a grid search over step sizes $\{0.01, 0.005, 0.001\}$. In `C-SquareCB` we set $\gamma_i = c\sqrt{t/\log(\delta^{-1})}$ and tune $c$ in $\{10, 20, 50, 100, 200, 500, 1000\}$. For `C-FastCB`, since the optimal loss $L_i^*$ is not known in advance, the exploration parameter $\gamma_i$ is treated as a hyper-parameter in our experiments. We set $\gamma_i = \gamma$ and tune it in $\{10, 20, 50, 100, 200, 500, 1000\}$. Deb et al. (2024a) tune for different choices of the perturbation constant (see Appendix F in Deb et al. (2024a)) and show that the unperturbed version perform almost as good as the perturbed ones, and are computationally more efficient. We saw a similar behavior in our experiments and report the final plots for only the unperturbed networks.

We compare the cumulative regret of the algorithms in Figure 1. Note that `C-SquareCB` and `C-FastCB` consistently show a sub-linear trend in regret and beat the existing benchmark, with `C-FastCB` performing better in some of the datasets, owing to it's *first order* order regret guarantee. We also compare it with another heuristic choice, where we substitute $\sum_{i=1}^{t} L_i^*$ by the sum of the observed losses until time $t-1$, i.e., $\sum_{i=1}^{t-1} L_i$ to choose $\gamma_t$, and note that it produces good results in the majority of environments (See Figure 3 in Appendix F). We also compare the performance of the algorithms for various choices of width of the network (see Appendix F).

Finally, we compare the percentage of constraints violated by our algorithms `C-SquareCB` and `C-FastCB` compared to their vanilla counterparts that does not use any *safety condition* in Figure 2. Our algorithms maintain the performance constraint while minimizing the regret.

## 7 CONCLUSION

In this paper, we developed two new algorithms, `C-SquareCB` and `C-FastCB`, for the problem of Conservative Contextual Bandits with general non-linear cost functions. Our algorithms use Inverse Gap Weighting (IGW) for exploration and rely on an online regression oracle for prediction. We provided regret guarantees for both algorithms, showing that `C-SquareCB` achieves a sub-linear regret in $T$, while `C-FastCB` achieves a first-order regret in terms of the cumulative loss of the optimal policy $L^*$. We also extended our approach by using neural networks for function approximation and provide end-to-end regret bounds. Finally, through experiments on real-world data, we showed that our methods outperform existing baseline while maintaining safety guarantee. Adapting our methods to other safe bandit frameworks such as the stage-wise setting (Moradipari et al., 2019; Amani et al., 2019) and to the more general reinforcement learning framework following Foster et al. (2023b) and Foster et al. (2023a) is left for future work.

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

# A    RELATED WORKS

**Contextual Bandits.** The study of bandit algorithms, especially in the contextual bandit setting, has seen significant development over the years. Initial works on linear bandits, such as those by Abe et al. (2003), Chu et al. (2011), and Abbasi-Yadkori et al. (2011), laid the foundation for exploration strategies with provable regret bounds. These works primarily leveraged linear models, achieving near-optimal performance in various settings. Agrawal & Goyal (2012) provided regret guarantee for the Thompson sampling algorithm in the multi-armed case and later extended it to the linear setting with provable guarantees (Agrawal & Goyal, 2013). The success of linear bandits naturally led to their extension to more complex settings, particularly nonlinear models. Generalized linear bandits (GLBs) explored by Filippi et al. (2010) and Li et al. (2017) introduced non-linearity through a link function, while preserving a linear dependence on the context.

**Contextual Bandits beyond linearity.** More recently, the rise of deep learning has led to interest in neural models for contextual bandits. Early attempts to incorporate neural networks into the bandit framework relied on using deep neural networks (DNNs) as feature extractors, with a linear model learned on top of the last hidden layer of the DNN (Lu & Van Roy, 2017; Zahavy & Mannor, 2020; Riquelme et al., 2018). Although these methods demonstrated empirical success, they lacked theoretical regret guarantees. The NeuralUCB (Zhou et al., 2020) algorithm combined neural networks with UCB-based exploration, and provided regret guarantees. This approach was further extended to Thompson Sampling in the work of Zhang et al. (2021), with both methods drawing on neural tangent kernels (NTKs) (Jacot et al., 2018; Allen-Zhu et al., 2019) and the notion of effective dimension $\tilde{d}$. However rcently Deb et al. (2024a) showed that these bounds are $\Omega(T)$ in the worst case even with an oblivious adversary. These methods also suffer from the computational complexity of inverting large matrices at each step of the algorithm remained a limitation, as the inversion scales with the number of neural network parameters. In response, Ban et al. (2022) introduced a novel approach that achieved regret bounds independent of the effective dimension $\tilde{d}$, though this method required specific distributional assumptions on the context.

**Agnostic Contextual Bandits.** Concurrently, agnostic algorithms for bandit problems were also studied starting from Dudik et al. (2011); Agarwal et al. (2014). Foster et al. (2018) provided an approach to leverage an offline weighted least squares regression oracle and demonstrated that this approach performs well compared to other existing contextual bandit algorithms. However, despite its success, the algorithm was theoretically sub-optimal, potentially incurring high regret in the worst case. Subsequently (Foster & Rakhlin, 2020) adapted the inverse gap weighting idea from Abe & Long (1999); Abe et al. (2003) related the bandit regret to the regret of online regression with square loss, while Foster & Krishnamurthy (2021) modified (Foster & Rakhlin, 2020), with binary Kullback–Leibler (KL) loss and a re-weighted inverse gap weighting scheme to provide a *first-order* regret bound. Further, Simchi-Levi & Xu (2020) showed that an offline regression oracle with $\mathcal{O}(\log T)$ calls can also be used to derive optimal regret gurantees for the general realizable case. This improves ove the $\mathcal{O}(T)$ calls by Foster & Krishnamurthy (2021) and (Foster & Rakhlin, 2020) and also relaxes the assumption to offline oracles instead of online, however it needs to make a strong assumption about the contexts - they are drawn i.i.d. from a fixed distribution.

**Constrained Bandits.** Bandit problems under constraints have also been studied extensively. The Bandits with Knapsacks problem looks at cumulative reward maximization under budget constraints (Badanidiyuru et al., 2013; Agrawal & Devanur, 2016; Immorlica et al., 2022; Sivakumar et al., 2022; Deb et al., 2024b). The general cost function case as in this work has been studied in Slivkins et al. (2023); Han et al. (2023) and provided sub-linear regret bounds using the Inverse gap weighting idea from Abe & Long (1999); Foster & Rakhlin (2020); Foster & Krishnamurthy (2021). In the stage-wise constraint setup, each arm generates both reward and cost signals from unknown distributions. The objective is to maximize cumulative rewards while ensuring the expected cost stays below a threshold at each round. Amani et al. (2019) and Moradipari et al. (2019) investigated this setting in the context of linear bandits, developing and evaluating explore-exploit algorithm and a Thompson sampling algorithm respectively. The setup in this work, *conservative bandits* was introduced in Wu et al. (2016) and subsequently studied in the linear setting Kazerouni et al. (2017); Garcelon et al. (2020), and all existing methods use a modified version of UCB. To the best of our knowledge neither a Thompson Sampling version has been studied, nor an oracle based approach for the general function case.

**Data Dependent Regret Bounds.** Adaptive algorithms can often perform better if the environment it is operating in is comparatively easier. A data dependent regret bound tries to capture such a phenomena. In a *first-order regret bounds,* the regret scales as in $L_* = \sum_{t=1}^{T} \ell_{t,a_t^*}$, the cumulative loss/cost of the optimal policy. It has a rich history, with Freund & Schapire (1997) proving the first such bound for the full information setting (or the classical expert setting) using Exponential Weights algorithm. For the $K$-armed bandit setting (with no contexts), first order bounds were provided in Agarwal et al. (2016). For the adversarial setting Agarwal et al. (2017b) provided a $\mathcal{O}(L_*^{2/3})$ bound and subsequently also posed an open problem at COLT - '*Can first-order regret bounds be developed for contextual bandits ?*'. Allen-Zhu et al. (2018) responded by providing a first order bound with an inefficient algorithm, and subsequently Foster & Krishnamurthy (2021) provided an algorithm with a reduction to online regression that was both efficient and provided a first order regret.

Cesa-Bianchi et al. (2006a) first posed the question of whether further improvements could be achieved by deriving *second-order* (variance-like) bounds on the regret for the full information setting. They provided two choices for second order bounds, one that depends on $\sum_{t=1}^{T} \ell_{t,a_t^*}^2$ (variance across time) and another that depends on $\sum_{k \leq K} p_{k,t} (\hat{\ell}_t - \ell_{k,t})^2$ (variance across actions), where $\hat{\ell}_t = \sum_{k=1}^{K} p_{t,k} \ell_{t,k}$, and $p_{k,t}$ is the probability with which expert $k$ is chosen in round $t$. For a more detailed discussion of second order bounds we refer the reader to Ito et al. (2020); Gaillard et al. (2014); Freund (2016); Ito et al. (2020); Cesa-Bianchi et al. (2006b); Pacchiano (2024).

## B  PROOF OF REGRET BOUND FOR `C-SquareCB`

**Lemma 3.1.** *Let Assumptions 1 and 2 hold. Then, the regret defined in* (1) *can be bounded as*

$$\texttt{Reg}_{\texttt{CB}}(T) \leq \sum_{t \in \mathcal{S}_T} \Big( h(\mathbf{x}_{t,a_t}) - h(\mathbf{x}_{t,a_t^*}) \Big) + n_T \Delta_h, \tag{6}$$

*where the set $\mathcal{S}_T$ consists of the rounds until the horizon $T$ when* `C-SquareCB` *played an IGW action and $n_T = |\mathcal{S}_T^c|$ is the number of times until $T$ where a baseline action was played.*

*Proof.* The decomposition follows as in Proposition 2 in (Kazerouni et al., 2017), and we reproduce the proof here for completeness. Recall that $\mathcal{S}_T = \{t \in [T] : a_t = b_t\}$ is the set of time steps when the baseline action was chosen and $\mathcal{S}_T^c = \{t \in [T] : a_t = \tilde{a}_t\}$ is the set of time steps when the SquareCB action was played. Then, we can decompose the regret as follows:

$$\begin{aligned}
\texttt{Reg}_{\texttt{CB}}(T) &= \sum_{t=1}^{T} h(\mathbf{x}_{t,a_t}) - \sum_{t=1}^{T} h(\mathbf{x}_{t,a_t^*}) \\
&\overset{(a)}{=} \sum_{t \in \mathcal{S}_T} \Big( h(\mathbf{x}_{t,a_t}) - h(\mathbf{x}_{t,a_t^*}) \Big) + \sum_{t \in \mathcal{S}_T^c} \Big( h(\mathbf{x}_{t,b_t}) - h(\mathbf{x}_{t,a_t^*}) \Big) \\
&\overset{(b)}{=} \sum_{t \in \mathcal{S}_T} \Big( h(\mathbf{x}_{t,a_t}) - h(\mathbf{x}_{t,a_t^*}) \Big) + \sum_{t \in \mathcal{S}_T^c} \Delta_{b_t}^t \\
&\overset{(c)}{\leq} \sum_{t \in \mathcal{S}_T} \Big( h(\mathbf{x}_{t,a_t}) - h(\mathbf{x}_{t,a_t^*}) \Big) + n_T \Delta_h,
\end{aligned}$$

where $(a)$ follows because $\mathcal{S}_T \cup \mathcal{S}_T^c = [T]$, $(b)$ follows by the definition of $\Delta_{b_t}^t = h(\mathbf{x}_{t,b_t}) - h(\mathbf{x}_{t,a_t^*})$, and $(c)$ follows by Assumption 2. $\qquad\square$

**Lemma 3.2.** *Suppose Assumption 1,2 and 3 holds. Then, with probability $1 - \delta/4$ the number of times the baseline action is played by* `C-SquareCB` *is bounded as*

$$\begin{aligned}
n_T \leq \frac{1}{\alpha y_l} \Big\{ &- (m_{\tau-1} + 1)(\Delta_l + \alpha y_l) \\
&+ 64\sqrt{K}\sqrt{(m_{\tau-1} + 1)} \Big( \sqrt{\texttt{Reg}_{\texttt{Sq}}(T)} + \sqrt{\log(8\delta^{-1})} \Big) \Big\}.
\end{aligned} \tag{7}$$

*Proof.* Let $\tau$ be the last round at which the algorithm plays the conservative action, i.e.,

$$\tau = \max\{1 \le t \le T | a_t = b_t\}.$$

Recall that $m_t = |\mathcal{S}_t|$ and $n_t = |\mathcal{S}_t^c|$. By the definition of $\tau$, we have that at round $\tau$

$$\hat{y}_{\tau,\tilde{a}_\tau} + \sum_{i\in\mathcal{S}_{\tau-1}}\sum_{a\in[K]} p_{i,a}\hat{y}_{i,a} + \sum_{i\in\mathcal{S}_{\tau-1}^c} h(\mathbf{x}_{i,b_i}) + 16\sqrt{m_{\tau-1}\Big(\texttt{Reg}_{\texttt{Sq}}(T) + \log(2/\delta)\Big)}$$

$$> (1+\alpha)\sum_{i=1}^{\tau} h(\mathbf{x}_{i,b_i}).$$

Therefore, we may write

$$\alpha\sum_{i=1}^{\tau} h(\mathbf{x}_{i,b_i}) < \sum_{i\in\mathcal{S}_{\tau-1}}\sum_{a\in[K]} p_{i,a}\hat{y}_{i,a} + \hat{y}_{\tau,\tilde{a}_\tau} - \sum_{i\in\mathcal{S}_{\tau-1}}(h(\mathbf{x}_{i,b_i}) + h(\mathbf{x}_{\tau,b_\tau}))$$

$$+ 16\sqrt{m_{\tau-1}\Big(\texttt{Reg}_{\texttt{Sq}}(T) + \log(2/\delta)\Big)}$$

$$= \sum_{i\in\mathcal{S}_{\tau-1}}\sum_{a\in[K]} p_{i,a}\hat{y}_{i,a} + \hat{y}_{\tau,\tilde{a}_\tau} - \sum_{i\in\mathcal{S}_{\tau-1}}\sum_{a\in[K]} p_{i,a}h(\mathbf{x}_{i,a_i^*})$$

$$+ \sum_{i\in\mathcal{S}_{\tau-1}}\sum_{a\in[K]} p_{i,a}h(\mathbf{x}_{i,a_i^*}) + \sum_{a\in[K]} p_{\tau,a}h(\mathbf{x}_{\tau,a_\tau^*})$$

$$- \sum_{a\in[K]} p_{\tau,a}h(\mathbf{x}_{\tau,a_\tau^*}) - \sum_{i\in\mathcal{S}_{\tau-1}}(h(\mathbf{x}_{i,b_i}) + h(\mathbf{x}_{\tau,b_\tau}))$$

$$+ 16\sqrt{m_{\tau-1}\Big(\texttt{Reg}_{\texttt{Sq}}(T) + \log(2/\delta)\Big)}$$

$$= \underbrace{\sum_{i\in\mathcal{S}_{\tau-1}}\Big(h(\mathbf{x}_{i,a_i^*}) - h(\mathbf{x}_{i,b_i})\Big) + \Big(h(\mathbf{x}_{\tau,a_\tau^*}) - h(\mathbf{x}_{\tau,b_\tau})\Big)}_{I}$$

$$+ \underbrace{\sum_{i\in\mathcal{S}_{\tau-1}}\sum_{a\in[K]} p_{i,a}\Big(\hat{y}_{i,a} - h(\mathbf{x}_{i,a_i^*})\Big)}_{II} + \underbrace{\sum_{a\in[K]} p_{\tau,a}\Big(\hat{y}_{\tau,\tilde{a}_\tau} - h(\mathbf{x}_{\tau,a_\tau^*})\Big)}_{III} \qquad (19)$$

$$+ 16\sqrt{m_{\tau-1}\Big(\texttt{Reg}_{\texttt{Sq}}(T) + \log(2/\delta)\Big)}$$

First consider term $I$. Using Assumption 2 we have that $\Delta_l \le h(\mathbf{x}_{i,a_i^*}) - h(\mathbf{x}_{i,b_i}) \le \Delta_h$. Also recall that $m_{\tau-1} = |S_{\tau-1}|$. Combining these we have:

$$\sum_{i\in S_{\tau-1}}\Big(h(\mathbf{x}_{i,a_i^*}) - h(\mathbf{x}_{i,b_i})\Big) + \Big(h(\mathbf{x}_{\tau,a_\tau^*}) - h(\mathbf{x}_{\tau,b_\tau})\Big) < -(m_{\tau-1}+1)\Delta_l$$

Next consider term $II$. Adding and subtracting $h(\mathbf{x}_{i,a})$, we obtain

$$\sum_{i\in S_{\tau-1}}\sum_{a\in[K]} p_{i,a}\Big(\hat{y}_{i,a} - h(\mathbf{x}_{i,a_i^*})\Big) = \underbrace{\sum_{i\in S_{\tau-1}}\sum_{a\in[K]} p_{i,a}\Big(h(\mathbf{x}_{i,a}) - h(\mathbf{x}_{i,a_i^*})\Big)}_{II(a)}$$

$$+ \underbrace{\sum_{i\in S_{\tau-1}}\sum_{a\in[K]} p_{i,a}\Big(\hat{y}_{i,a} - h(\mathbf{x}_{i,a})\Big)}_{II(b)}.$$

Consider term $II(a)$. Using Lemma 3 in Foster & Rakhlin (2020) we have

$$\sum_{a \in [K]} p_{i,a} \left[ h(\mathbf{x}_{i,a}) - h(\mathbf{x}_{i,a_i^*}) - \frac{\gamma_i}{4} \big( \hat{y}_{i,a} - h(\mathbf{x}_{i,a}) \big)^2 \right] \le \frac{K}{\gamma_i}.$$

Now summing for all $i \in \mathcal{S}_{\tau-1}$ we have

$$\sum_{i \in \mathcal{S}_{\tau-1}} \sum_{a \in [K]} p_{i,a} \left[ h(\mathbf{x}_{i,a}) - h(\mathbf{x}_{i,a_i^*}) - \frac{\gamma_i}{4} \big( \hat{y}_{i,a} - h(\mathbf{x}_{i,a}) \big)^2 \right] \le \sum_{i \in \mathcal{S}_{\tau-1}} \frac{K}{\gamma_i}.$$

Using this, we can bound term $II(a)$ as follows:

$$\sum_{i \in \mathcal{S}_{\tau-1}} \sum_{a \in [K]} p_{i,a} \Big( h(\mathbf{x}_{i,a}) - h(\mathbf{x}_{i,a_i^*}) \Big) \le \sum_{i \in \mathcal{S}_{\tau-1}} \frac{2K}{\gamma_i} + \sum_{i \in \mathcal{S}_{\tau-1}} \sum_{a \in [K]} \frac{\gamma_i}{4} p_{i,a} \big( \hat{y}_{i,a} - h(\mathbf{x}_{i,a}) \big)^2.$$

Now recall that $\gamma_i = \sqrt{K|\mathcal{S}_i| / (2\text{Reg}_{\text{Sq}}(T) + 16 \log(8\delta^{-1}))}$ and therefore plugging this back in the above equation we get

$$\sum_{i \in \mathcal{S}_{\tau-1}} \sum_{a \in [K]} p_{i,a} \Big( h(\mathbf{x}_{i,a}) - h(\mathbf{x}_{i,a_i^*}) \Big) \le \sum_{i \in \mathcal{S}_{\tau-1}} \frac{2K}{\gamma_i} + \sum_{i \in \mathcal{S}_{\tau-1}} \sum_{a \in [K]} \frac{\gamma_i}{4} p_{i,a} \big( \hat{y}_{i,a} - h(\mathbf{x}_{i,a}) \big)^2$$

$$= 2K \sum_{i \in \mathcal{S}_{\tau-1}} \sqrt{\frac{2\text{Reg}_{\text{Sq}}(T) + 16 \log(8\delta^{-1})}{K|\mathcal{S}_i|}}$$

$$+ \frac{1}{4} \sum_{i \in \mathcal{S}_{\tau-1}} \sqrt{\frac{K|\mathcal{S}_i|}{(2\text{Reg}_{\text{Sq}}(T) + 16 \log(8\delta^{-1}))}} \sum_{a \in [K]} p_{i,a} \big( \hat{y}_{i,a} - h(\mathbf{x}_{i,a}) \big)^2$$

$$\overset{(a)}{\le} 2\sqrt{K(2\text{Reg}_{\text{Sq}}(T) + 16 \log(8\delta^{-1}))} \sum_{i=1}^{m_{\tau-1}} \frac{1}{\sqrt{i}}$$

$$+ \frac{1}{4} \sqrt{\frac{K m_{\tau-1}}{2\text{Reg}_{\text{Sq}}(T) + 16 \log(8\delta^{-1})}} \sum_{i \in \mathcal{S}_{\tau-1}} \sum_{a \in [K]} p_{i,a} \big( \hat{y}_{i,a} - h(\mathbf{x}_{i,a}) \big)^2.$$

In $(a)$, we used the fact that $\gamma_i$ depends on $|\mathcal{S}_i|$ and that $\max_{i \in \mathcal{S}_i} \gamma_i = \sqrt{\frac{K m_{\tau-1}}{2\text{Reg}_{\text{Sq}}(T) + 16 \log(8\delta^{-1})}}$.

Now note that the `C-SquareCB` actions are only played for $i \in \mathcal{S}_T$ and therefore invoking Assumption 3, we can use Lemma 2 in Foster & Rakhlin (2020) to show that with probability $1 - \delta/4$

$$\sum_{i \in \mathcal{S}_{\tau-1}} \sum_{a \in [K]} p_{i,a} \big( \hat{y}_{i,a} - h(\mathbf{x}_{i,a}) \big)^2 \le 2\text{Reg}_{\text{Sq}}(m_{\tau-1}) + 16 \log(8\delta^{-1})$$

Further note that $\sum_{i=1}^{m_{\tau-1}} \frac{1}{\sqrt{i}} \le 2\sqrt{m_{\tau-1}}$ . Therefore term $II(a)$ can be bounded as

$$\sum_{i \in \mathcal{S}_{\tau-1}} \sum_{a \in [K]} p_{i,a} \Big( h(\mathbf{x}_{i,a}) - h(\mathbf{x}_{i,a_i^*}) \Big)$$

$$\le 16 \sqrt{K m_{\tau-1} (\text{Reg}_{\text{Sq}}(T) + \log(8\delta^{-1}))}$$

$$+ \frac{1}{4} \sqrt{\frac{K m_{\tau-1}}{2\text{Reg}_{\text{Sq}}(T) + 16 \log(8\delta^{-1})}} \Big( 2\text{Reg}_{\text{Sq}}(m_{\tau-1}) + 16 \log(8\delta^{-1}) \Big)$$

$$\le 17 \sqrt{K m_{\tau-1}} \Big( \sqrt{\text{Reg}_{\text{Sq}}(T)} + \sqrt{\log(8\delta^{-1})} \Big),$$

where we have used the fact $\text{Reg}_{\text{Sq}}(m_{\tau-1}) \le \text{Reg}_{\text{Sq}}(T)$.

Now consider term $II(b)$. Suppose $\mathbb{E}_{p_i}$ be the expectation with respect to $p_{i,a}$. Then, we may write

$$
\sum_{i \in S_{\tau-1}} \sum_{a \in [K]} p_{i,a}\Big(h(\mathbf{x}_{i,a}) - \hat{y}_{i,a}\Big) = \sum_{i \in S_{\tau-1}} \mathbb{E}_{p_i}\Big[\big(h(\mathbf{x}_{i,a}) - \hat{y}_{i,a}\big)\Big]
$$

$$
= \sum_{i \in S_{\tau-1}} \mathbb{E}_{p_i} \sqrt{\big(h(\mathbf{x}_{i,a}) - \hat{y}_{i,a}\big)^2}
$$

$$
\overset{(a)}{\leq} \sum_{i \in S_{\tau-1}} \sqrt{\mathbb{E}_{p_i}\big(h(\mathbf{x}_{i,a}) - \hat{y}_{i,a}\big)^2}
$$

$$
\overset{(b)}{\leq} \sqrt{m_{\tau-1} \sum_{i \in S_{\tau-1}} \sum_{a \in [K]} p_{i,a}\big(h(\mathbf{x}_{i,a}) - \hat{y}_{i,a}\big)^2},
$$

where $(a)$ follows by Jensen and $(b)$ follows by Cauchy Schwartz. Again, using Lemma 2 in Foster & Rakhlin (2020), with probability $1 - \delta/4$, we have

$$
\sum_{i \in S_{\tau-1}} \sum_{a \in [K]} p_{i,a}\Big(h(\mathbf{x}_{i,a}) - \hat{y}_{i,a}\Big) \leq \sqrt{m_{\tau-1}\big(2\texttt{Reg}_{\texttt{Sq}}(m_{\tau-1}) + 16\log(8\delta^{-1})\big)}.
$$

Finally consider term $III$. Since $0 \leq h(x_{i,a}), \hat{y}_{i,a} \leq 1$, we may write

$$
\sum_{a \in [K]} p_{\tau,a}\big(\hat{y}_{\tau,\tilde{a}_\tau} - h(\mathbf{x}_{\tau,a^*_\tau})\big) \leq 2.
$$

Combining all the bounds, for $K \geq 2$ and $\texttt{Reg}_{\texttt{Sq}}(T) \geq 1$, with probability $1 - \delta/2$, we have

$$
\alpha \sum_{i=1}^{\tau} h(\mathbf{x}_{i,b_i}) \leq -(m_{\tau-1} + 1)\Delta_l + 64\sqrt{K(m_{\tau-1}+1)}\Big(\sqrt{\texttt{Reg}_{\texttt{Sq}}(T)} + \sqrt{\log(8\delta^{-1})}\Big). \quad (20)
$$

Now, using the fact that $m_{\tau-1} + n_{\tau-1} + 1 = \tau$, and Assumption 2, we have $y_l \leq h(\mathbf{x}_{i,b_i}) \leq y_h, \forall i \in [T]$. Therefore,

$$
\alpha \sum_{i=1}^{\tau} h(\mathbf{x}_{i,b_i}) \geq \alpha \left(m_{\tau-1} + n_{\tau-1} + 1\right) y_l.
$$

Combining this with (20), with probability $1 - \delta/2$, we obtain

$$
\alpha n_{\tau-1} y_l \leq -(m_{\tau-1} + 1)(\Delta_l + \alpha y_l) + 64\sqrt{K(m_{\tau-1}+1)}\Big(\texttt{Reg}_{\texttt{Sq}}(T) + \sqrt{\log(8\delta^{-1})}\Big)
$$

$$
= -(m_{\tau-1} + 1)(\Delta_l + \alpha y_l) + 64\sqrt{K}\sqrt{(m_{\tau-1}+1)}\Big(\texttt{Reg}_{\texttt{Sq}}(T) + \sqrt{\log(8\delta^{-1})}\Big).
$$

Finally, using $n_T = n_\tau = n_{\tau-1} + 1$, with probability $1 - \delta/2$, we have

$$
n_T \leq \frac{1}{\alpha y_l}\left\{ -(m_{\tau-1} + 1)(\Delta_l + \alpha y_l) + 64\sqrt{K}\sqrt{(m_{\tau-1}+1)}\left(\sqrt{\texttt{Reg}_{\texttt{Sq}}(T)} + \sqrt{\log(2\delta^{-1})}\right) \right\}.
$$

$\square$

**Lemma 3.3.** *Suppose Assumption 1,2 and 3 holds. Then, with probability $1 - \delta/4$ the number of times the baseline action is played by* `C-SquareCB` *is bounded as follows:*

$$
n_T \leq \mathcal{O}\left(\frac{K(\texttt{Reg}_{\texttt{Sq}}(T) + \log(8\delta^{-1}))}{\alpha y_l(\Delta_l + \alpha y_l)}\right). \quad (8)
$$

*Proof.* Let us define

$$
Q(m_{\tau-1}) = -(m_{\tau-1} + 1)(\Delta_l + \alpha y_l) + 64\sqrt{K}\sqrt{(m_{\tau-1}+1)}\left(\sqrt{\texttt{Reg}_{\texttt{Sq}}(T)} + \sqrt{\log(2\delta^{-1})}\right)
$$

Note that we have

$$Q(m_{\tau-1}) \leq -c_1 m + c_2 \sqrt{m} := f(m)$$

where

$$c_1 = \Delta_l + \alpha r_l \geq 0,$$

$$c_2 = 64\sqrt{K}\left(\sqrt{\texttt{Reg}_{\texttt{Sq}}(T)} + \sqrt{\log(2\delta^{-1})}\right) \geq 0,$$

$$m = m_{\tau-1} + 1.$$

Setting $f'(m) = 0$, and solving we get $m^* = \dfrac{c_2^2}{4c_1^2}$. Now note that $f$ is concave and that $f''(m^*) < 0$ and therefore,

$$Q(m_{\tau-1}) \leq f(m) \leq f(m^*) = -\frac{c_2^2}{4c_1} + \frac{c_2^2}{2c_1}$$

$$= \frac{c_2^2}{4c_1}$$

$$\leq \mathcal{O}\left(\frac{K(\texttt{Reg}_{\texttt{Sq}}(T) + \log(2\delta^{-1}))}{\Delta_l + \alpha y_l}\right).$$

Finally noting that $n_T \leq n_{\tau-1} + 1 \leq \dfrac{Q(m_{\tau-1})}{\alpha y_l} + 1$ completes the proof. $\qquad\square$

**Lemma 3.4.** *Suppose Assumptions 1 and 3 hold. Then, for $\delta > 0$ and $\gamma_t = \sqrt{K|\mathcal{S}_t|/(\texttt{Reg}_{\texttt{Sq}}(T) + \log(4\delta^{-1}))}$, with probability $1 - \delta/4$,* C-SquareCB *guarantees*

$$\sum_{t \in \mathcal{S}_T}\left(h(\mathbf{x}_{t,a_t}) - h(\mathbf{x}_{t,a_t^*})\right) \leq \mathcal{O}\left(\sqrt{Km_T\texttt{Reg}_{\texttt{Sq}}(T)} + \sqrt{Km_T\log(8\delta^{-1})}\right). \qquad (10)$$

*Proof.* Using Lemma 3 from Foster & Rakhlin (2020) for any $i \in [K]$ we have

$$\sum_{a \in [K]} p_{i,a}\left[h(\mathbf{x}_{i,a}) - h(\mathbf{x}_{i,a_i^*}) - \frac{\gamma_i}{4}\left(\hat{y}_{i,a} - h(\mathbf{x}_{i,a})\right)^2\right] \leq \frac{K}{\gamma_i}$$

Now summing for all $i \in \mathcal{S}_T$ we have

$$\sum_{i \in \mathcal{S}_T}\sum_{a \in [K]} p_{i,a}\left[h(\mathbf{x}_{i,a}) - h(\mathbf{x}_{i,a_i^*}) - \frac{\gamma_i}{4}\left(\hat{y}_{i,a} - h(\mathbf{x}_{i,a})\right)^2\right] \leq \sum_{i \in \mathcal{S}_T}\frac{K}{\gamma_i}.$$

Using this get the following bound:

$$\sum_{i \in \mathcal{S}_T}\sum_{a \in [K]} p_{i,a}\left(h(\mathbf{x}_{i,a_i^*}) - h(\mathbf{x}_{i,a})\right) \leq \sum_{i \in \mathcal{S}_T}\frac{2K}{\gamma_i} + \sum_{i \in \mathcal{S}_T}\sum_{a \in [K]}\frac{\gamma_i}{4}p_{i,a}\left(\hat{y}_{i,a} - h(\mathbf{x}_{i,a})\right)^2$$

Now recall that $\gamma_i = \sqrt{K|\mathcal{S}_i|/(\texttt{Reg}_{\texttt{Sq}}(T) + 16\log(8\delta^{-1}))}$ and therefore plugging this back in the above equation we get:

$$\sum_{i \in \mathcal{S}_T}\sum_{a \in [K]} p_{i,a}\left(h(\mathbf{x}_{i,a_i^*}) - h(\mathbf{x}_{i,a})\right) \leq \sum_{i \in \mathcal{S}_T}\frac{2K}{\gamma_i} + \sum_{i \in \mathcal{S}_T}\sum_{a \in [K]}\frac{\gamma_i}{4}p_{i,a}\left(\hat{y}_{i,a} - h(\mathbf{x}_{i,a})\right)^2$$

$$= 2K\sum_{i \in \mathcal{S}_T}\sqrt{\frac{\texttt{Reg}_{\texttt{Sq}}(T) + 16\log(8\delta^{-1})}{K|\mathcal{S}_i|}}$$

$$+ \frac{1}{4}\sum_{i \in \mathcal{S}_T}\sqrt{\frac{K|\mathcal{S}_i|}{(\texttt{Reg}_{\texttt{Sq}}(T) + 16\log(2\delta^{-1}))}}\sum_{a \in [K]} p_{i,a}\left(\hat{y}_{i,a} - h(\mathbf{x}_{i,a})\right)^2$$

$$\overset{(a)}{\leq} 2\sqrt{K(\texttt{Reg}_{\texttt{Sq}}(T) + 16\log(8\delta^{-1}))}\sum_{i=1}^{m_T}\frac{1}{\sqrt{i}}$$

$$+ \frac{1}{4}\sqrt{\frac{Km_T}{\texttt{Reg}_{\texttt{Sq}}(T) + 16\log(8\delta^{-1})}}\sum_{i \in \mathcal{S}_T}\sum_{a \in [K]} p_{i,a}\left(\hat{y}_{i,a} - h(\mathbf{x}_{i,a})\right)^2$$

In $(a)$ we used the fact that $\gamma_i$ depends on $|\mathcal{S}_i|$ and that $\displaystyle\max_{i\in\mathcal{S}_i}\gamma_i = \sqrt{\dfrac{Km_T}{\mathtt{Reg}_{\mathtt{Sq}}(T) + 16\log(8\delta^{-1})}}$. Now note that the `C-SquareCB` actions are only played for $i \in \mathcal{S}_T$ and therefore invoking Assumption 3, we can use Lemma 2 from (Foster & Rakhlin, 2020) to show that with probability $1 - \delta/4$

$$\sum_{i\in\mathcal{S}_T}\sum_{a\in[K]} p_{i,a}\big(\hat{y}_{i,a} - h(\mathbf{x}_{i,a})\big)^2 \le 2\mathtt{Reg}_{\mathtt{Sq}}(m_T) + 16\log(8\delta^{-1})$$

Further note that $\displaystyle\sum_{i=1}^{m_T}\dfrac{1}{\sqrt{i}} \le 2\sqrt{m_T}$. Therefore term $II(a)$ can be bounded as follows

$$\sum_{i\in\mathcal{S}_T}\sum_{a\in[K]} p_{i,a}\Big(h(\mathbf{x}_{i,a_i^*}) - h(\mathbf{x}_{i,a})\Big)$$

$$\le 4\sqrt{Km_T(\mathtt{Reg}_{\mathtt{Sq}}(T) + 16\log(8\delta^{-1}))}$$

$$+ \frac{1}{4}\sqrt{\frac{Km_T}{2\mathtt{Reg}_{\mathtt{Sq}}(T) + 16\log(8\delta^{-1})}}\Big(2\mathtt{Reg}_{\mathtt{Sq}}(m_T) + 16\log(8\delta^{-1})\Big)$$

$$\le 17\sqrt{Km_T}\Big(\sqrt{\mathtt{Reg}_{\mathtt{Sq}}(T)} + \sqrt{\log(8\delta^{-1})}\Big), \tag{21}$$

where we have used the fact $\mathtt{Reg}_{\mathtt{Sq}}(m_T) \le \mathtt{Reg}_{\mathtt{Sq}}(T)$.

Now we can modify the proof of Lemma 2 of Foster & Rakhlin (2020) to take the sum over $i \in \mathcal{S}_T$ instead of $i \in [T]$ to ensure that with probability $1 - \delta/4$

$$\sum_{i\in\mathcal{S}_T} h(\mathbf{x}_{t,a_t}) - h(\mathbf{x}_{t,a_t^*}) \le \sum_{i\in\mathcal{S}_T}\sum_{a\in[K]} p_{i,a}\Big(h(\mathbf{x}_{i,a_i^*}) - h(\mathbf{x}_{i,a})\Big) + \sqrt{2m_T\log(8\delta^{-1})}.$$

Combining with (21) and noting that $\mathtt{Reg}_{\mathtt{Sq}}(T) \ge 1$ we get with probability $1 - \delta/4$

$$\sum_{i\in\mathcal{S}_T} h(\mathbf{x}_{t,a_t}) - h(\mathbf{x}_{t,a_t^*}) \le 32\sqrt{Km_T}\Big(\sqrt{\mathtt{Reg}_{\mathtt{Sq}}(T)} + \sqrt{\log(8\delta^{-1})}\Big)$$

which completes the proof. $\qquad\square$

**Lemma 3.5.** *Let Assumptions 1, 2 and 3 hold. Then, for $\delta > 0$ and $\gamma_t = \sqrt{K|\mathcal{S}_t|/(\mathtt{Reg}_{\mathtt{Sq}}(m_T) + \log(8\delta^{-1}))}$, with probability $1 - \delta/2$, `C-SquareCB` satisfies the performance constraint in (2).*

*Proof.* For $t = 1$ if the condition in line 8 holds then $\tilde{a}_1 = a_1$ and we have that with probability $1 - \delta$

$$\hat{y}_{1,a_1} - 16\sqrt{(m_0 + 1)(1 + \log(1/\delta))} \le (1 + \alpha)h(\mathbf{x}_{1,b_1})$$

Noting that $|\hat{y}_{1,a_1} - h(\mathbf{x}_{i,a_1})| \le 2$ and therefore with probability $1 - \delta$

$$h(\mathbf{x}_{i,a_1}) \le (1 + \alpha)h(\mathbf{x}_{1,b_1}).$$

Further, if the condition in line 8 doesn't hold, then $a_1 = b_1$, and therefore

$$h(\mathbf{x}_{i,a_1}) \le (1 + \alpha)h(\mathbf{x}_{1,b_1}),$$

showing that the performance constraint in Definition 2.2 is satisfied. Now assume that the constraint holds for $t - 1$ and now consider $t \in [T]$. Note that

$$\left|\sum_{i\in\mathcal{S}_{t-1}}\sum_{a\in[K]} p_{i,a}\hat{y}_{i,a} - h(\mathbf{x}_{i,\tilde{a}_i})\right| \le \underbrace{\left|\sum_{i\in\mathcal{S}_{t-1}}\sum_{a\in[K]} p_{i,a}\hat{y}_{i,a} - \sum_{i\in\mathcal{S}_{t-1}}\sum_{a\in[K]} p_{i,a}h(\mathbf{x}_{i,a})\right|}_{I}$$

$$+ \underbrace{\left|\sum_{i\in\mathcal{S}_{t-1}}\sum_{a\in[K]} p_{i,a}h(\mathbf{x}_{i,a}) - h(\mathbf{x}_{i,\tilde{a}_i})\right|}_{II}$$

Consider term $I$. We handle it as in the proof of Lemma 3.2 as follows: Suppose $\mathbb{E}_{p_i}$ be the expectation with respect to $p_{i,a}$. Then

$$
\left| \sum_{i \in S_{t-1}} \sum_{a \in [K]} p_{i,a} \Big( h(\mathbf{x}_{i,a}) - \hat{y}_{i,a} \Big) \right| = \left| \sum_{i \in S_{t-1}} \mathbb{E}_{p_i} \Big[ \big( h(\mathbf{x}_{i,a}) - \hat{y}_{i,a} \big) \Big] \right|
$$

$$
= \left| \sum_{i \in S_{t-1}} \mathbb{E}_{p_i} \sqrt{\big( h(\mathbf{x}_{i,a}) - \hat{y}_{i,a} \big)^2} \right|
$$

$$
\overset{(a)}{\leq} \left| \sum_{i \in S_{t-1}} \sqrt{\mathbb{E}_{p_i} \big( h(\mathbf{x}_{i,a}) - \hat{y}_{i,a} \big)^2} \right|
$$

$$
\overset{(b)}{\leq} \sqrt{m_{t-1} \sum_{i \in S_{t-1}} \sum_{a \in [K]} p_{i,a} \big( h(\mathbf{x}_{i,a}) - \hat{y}_{i,a} \big)^2}
$$

where $(a)$ follows by Jensen and $(b)$ follows by Cauchy Schwartz. Again using Lemma 2 from (Foster & Rakhlin, 2020) with probability $1 - \dfrac{\delta}{2}$

$$
\sum_{i \in S_{t-1}} \sum_{a \in [K]} p_{i,a} \Big( h(\mathbf{x}_{i,a}) - \hat{y}_{i,a} \Big)^2 \leq \sqrt{m_{t-1} \big( 2\mathtt{Reg}_{\mathtt{Sq}}(m_{t-1}) + 16 \log(2\delta^{-1}) \big)} \tag{22}
$$

Next, consider term $II$. Consider the following filtration

$$
\mathcal{F}_{t-1} = \sigma \bigg( (\mathbf{x}_{i,a}, \tilde{a}_i, y_{i,\tilde{a}_i}), \mathbf{x}_{t,a}; 1 \leq i \leq t-1, a \in [K] \bigg).
$$

Note that $\mathbb{E}\big[ h(\mathbf{x}_{t,\tilde{a}_t}) | \mathcal{F}_{t-1} \big] = \sum_{a \in [K]} p_{t,a} h(\mathbf{x}_{t,a})$, and therefore using Azuma-Hoeffding we have that with probability $1 - \dfrac{\delta}{2}$

$$
\left| \sum_{i \in \mathcal{S}_{t-1}} \sum_{a \in [K]} p_{i,a} h(\mathbf{x}_{i,a}) - h(\mathbf{x}_{i,\tilde{a}_i}) \right| \leq 2 \sqrt{m_{t-1} \log \left( \frac{2}{\delta} \right)} \tag{23}
$$

Combing (22) and (23) and taking a union bound we have with probability $1 - \delta$

$$
\left| \sum_{i \in \mathcal{S}_{t-1}} \sum_{a \in [K]} p_{i,a} \hat{y}_{i,a} - h(\mathbf{x}_{i,\tilde{a}_i}) \right| \leq 8 \sqrt{m_{t-1} \Big( \mathtt{Reg}_{\mathtt{Sq}}(m_{t-1}) + \log(2/\delta) \Big)}
$$

Further $|\hat{y}_{t,\tilde{a}_t} - h(\mathbf{x}_{t,\tilde{a}_t})| \leq 2$, and therefore with probability $1 - \delta$

$$
\left| \hat{y}_{t,\tilde{a}_t} + \sum_{i \in \mathcal{S}_{t-1}} \sum_{a \in [K]} p_{i,a} \hat{y}_{i,a} - h(\mathbf{x}_{i,\tilde{a}_i}) - h(\mathbf{x}_{t,\tilde{a}_t}) \right| \leq 16 \sqrt{m_{t-1} \Big( \mathtt{Reg}_{\mathtt{Sq}}(m_{t-1}) + \log(2/\delta) \Big)}. \tag{24}
$$

Now if line 8 of Algorithm 1 holds at time step $t$, then we have

$$
\hat{y}_{t,\tilde{a}_t} + \sum_{i \in \mathcal{S}_{t-1}} \sum_{a \in [K]} p_{i,a} \hat{y}_{i,a} + \sum_{i \in \mathcal{S}_{t-1}^c} h(\mathbf{x}_{i,b_i}) + 16 \sqrt{m_{t-1} \Big( \mathtt{Reg}_{\mathtt{Sq}}(m_{t-1}) + \log(2/\delta) \Big)}
$$

$$
\leq (1 + \alpha) \sum_{i=1}^{t} h(\mathbf{x}_{i,b_t}),
$$

and therefore invoking (24), we have with probability $1 - \delta$

$$
h(\mathbf{x}_{t,\tilde{a}_t}) + \sum_{i \in \mathcal{S}_{t-1}} h(\mathbf{x}_{i,\tilde{a}_i}) + \sum_{i \in \mathcal{S}_{t-1}^c} h(\mathbf{x}_{i,b_i}) \leq (1 + \alpha) \sum_{i=1}^{t} h(\mathbf{x}_{i,b_t})
$$

Now note that for all $i \in S_{t-1}$, $a_i = \tilde{a}_i$, for all $i \in \mathcal{S}_{t-1}^c$, $a_i = b_i$, and using $\mathcal{S}_{t-1} \cup \mathcal{S}_{t-1}^c = [t-1]$, and the fact that the condition in line 8 is satisfied we have with probability $1 - \delta$

$$h(\mathbf{x}_{t,a_t}) + \sum_{i \in [t-1]} h(\mathbf{x}_{i,a_i}) \leq (1+\alpha)\sum_{i=1}^{t} h(\mathbf{x}_{i,b_t}),$$

satisfying the performance condition in Definition 2.2.

Next we consider the case when the condition in line 8 does not hold. Invoking the fact that the performance constraint holds until time $t - 1$, we have with probability $1 - \delta$

$$\sum_{t=1}^{t-1} h(\mathbf{x}_{i,a_i}) \leq (1+\alpha)\sum_{i=1}^{t-1} h(\mathbf{x}_{i,b_t})$$

Adding $h(\mathbf{x}_{t,b_t})$ on both sides of the above equation we get

$$h(\mathbf{x}_{t,b_t}) + \sum_{i=1}^{t-1} h(\mathbf{x}_{i,a_i}) \leq h(\mathbf{x}_{t,b_t}) + (1+\alpha)\sum_{i=1}^{t-1} h(\mathbf{x}_{i,b_t}).$$

Noting that when condition in line 8 does not hold at step $t$, then $a_t = b_t$ and that $\alpha > 0$, we have with probability $1 - \delta$

$$\sum_{i=1}^{t} h(\mathbf{x}_{i,a_i}) \leq (1+\alpha)\sum_{i=1}^{t} h(\mathbf{x}_{i,b_t}),$$

satisfying the performance constraint in Definition 2.2 for step $t$. Using mathematical induction we conclude that the performance constraint holds for all $t \in [T]$, completing the proof. $\qquad\square$

## C    PROOF OF REGRET BOUND FOR `C-FastCB`

*Proof of Theorem 4.1.* The proof of the theorem follows along the following steps, and the proof of the intermediate lemmas can be found at the end of this proof.

1. **Regret Decomposition:** The regret decomposition follows using Lemma 3.1 as in the proof of Theorem 3.1.

   **Lemma 3.1.** *Let Assumptions 1 and 2 hold. Then, the regret defined in (1) can be bounded as*

   $$\texttt{Reg}_{\text{CB}}(T) \leq \sum_{t \in \mathcal{S}_T} \left(h(\mathbf{x}_{t,a_t}) - h(\mathbf{x}_{t,a_t^*})\right) + n_T \Delta_h, \tag{6}$$

   *where the set $\mathcal{S}_T$ consists of the rounds until the horizon $T$ when `C-SquareCB` played an IGW action and $n_T = |\mathcal{S}_T^c|$ is the number of times until $T$ where a baseline action was played.*

2. **Upper Bound on $n_T$ :** The condition in Line 7 determines how many times the baseline action is played. Suppose $m_t = |\mathcal{S}_t|$ and $\tau = \max\{1 \leq t \leq T : a_t = b_t\}$, i.e., the last time step at which `C-FastCB` played an action according to the baseline strategy.

   Before we proceed and give a bound on $n_T$, the number of times the baseline action is played by Algorithm 2, we specify how the exploration factor $\gamma_i$ is chosen. Unlike in Foster & Krishnamurthy (2021) where $\gamma_i = \gamma = \max(\sqrt{KL^*/(3\texttt{Reg}_{\text{KL}}(T))}, 10K)$, for all $i \in [K]$, we need to choose a time dependent $\gamma_i$ to ensure that we control both $n_T$ and the regret by playing the non-conservative actions. However using a different $\gamma_i$ at every step does not lead to a *first-order* regret bound for the first term in (6). Therefore we set $\gamma_i$ in an episodic manner, where $\gamma_i$ remains same in an

episode. More specifically we choose $\gamma_i$ as follows:

$$\gamma_0 = 1, \eta_0 = 1, L_i^* = 0$$
$$\text{for } i \in \mathcal{S}_T$$
$$L_i^* = L_{i-1}^* + h(\mathbf{x}_{t,a_i^*})$$
$$\text{if } \quad L_i^* > 2\eta_{i-1}$$
$$\eta_i = 2\eta_{i-1} \qquad\qquad (\gamma_{\text{i}}\text{-Schedule})$$
$$\text{else}$$
$$\eta_i = \eta_{i-1}$$
$$\gamma_i = \max\left(10K, \sqrt{\frac{K\eta_i}{\text{Reg}_{\text{KL}}(T)}}\right)$$

(a) The following lemma upper-bounds $n_T$ in terms of $m_\tau$, $\sum_{i \in \mathcal{S}_\tau} L^*(i)$, the cumulative cost in the set $\mathcal{S}_{\tau-1}$, and the KL loss $\text{Reg}_{\text{KL}}(T)$, using the above schedule for $\gamma_i$.

**Lemma C.1.** *Suppose Assumption 1,2, 4 holds. Then, the number of times the baseline action is played by* `C-FastCB` *is bounded as*

$$n_T \leq \frac{1}{\alpha y_l}\Bigg\{ -(m_{\tau-1}+1)(\Delta_l + \alpha y_l)$$
$$+ 60\sqrt{K\text{Reg}_{\text{KL}}(T)\log\left(\sum_{i \in \mathcal{S}_{\tau-1}} h(\mathbf{x}_{i,a_i^*})\right)\left(\sum_{i \in \mathcal{S}_{\tau-1}} h(\mathbf{x}_{i,a_i^*}) + 1\right)}\Bigg\}. \quad (25)$$

(b) Now note that since $L^*(i) \in [0,1]$, $\sum_{i \in \mathcal{S}_{\tau-1}} L^*(i) \leq m_{\tau-1}$. Therefore the second term in (25) grows as $\sqrt{m_{\tau-1}\log m_{\tau-1}}$ and that the first term decreases linearly in $m_{\tau-1}$, and therefore one can further bound $n_T$ in the following lemma.

**Lemma C.2.** *Suppose Assumption Assumption 1,2, 4 holds. Then the number of times the baseline action is played by* `C-SquareCB` *is bounded as follows:*

$$n_T \leq \mathcal{O}\left(\frac{K\text{Reg}_{\text{KL}}(T)}{\alpha y_l(\Delta_l + \alpha y_l)}\log\left(\frac{e\sqrt{K\text{Reg}_{\text{KL}}(T)}}{\Delta_l + \alpha y_l}\right)\right). \quad (26)$$

3. **Bounding the Final Regret:** We next move to bounding the first term in (6), with the schedule of $\gamma_i$ as described in Step-2. Note that $\mathcal{D}_T$ only contains the input-output pairs at time steps when the IGW action was picked, i.e., all $t \in \mathcal{S}_T$, and therefore, (11) reduces to

$$\sum_{t \in \mathcal{S}_T} \ell_{\text{KL}}(\hat{y}_{t,a_t}, y_{t,a_t}) - \inf_{g \in \mathcal{H}} \sum_{t \in \mathcal{S}_T} \ell_{\text{KL}}(g(\mathbf{x}_{t,a_t}), y_{t,a_t}) \leq \text{Reg}_{\text{KL}}(T). \quad (27)$$

The next lemma bounds the regret of the first term in (6) with an adaptive $\gamma_i$.

**Lemma C.3.** *Suppose Assumptions 1 and 4 hold. Then for $\gamma_t$ chosen as in ($\gamma_{\text{i}}$-Schedule), we have*

$$\mathbb{E}\sum_{t \in \mathcal{S}_T}\left(h(\mathbf{x}_{t,a_t}) - h(\mathbf{x}_{t,a_t^*})\right) \leq \mathcal{O}\left(\sqrt{K\text{Reg}_{\text{KL}}(T)\log\left(L_{\mathcal{S}_T}^*\right)L_{\mathcal{S}_T}^*}\right). \quad (28)$$

*where $L_{\mathcal{S}_T}^* = \sum_{t \in \mathcal{S}_T} h(\mathbf{x}_{t,a_t^*})$ is the cumulative cost of the optimal policy in the subset $\mathcal{S}_T$.*

Note that $L_{\mathcal{S}_T}^* \leq L^*$ and therefore combining (6), (26), and (28), the regret bound in (12) holds.

4. **Performance Constraint:** Finally the following lemma shows that the condition in Line 7 of `C-SquareCB` ensures that the Performance Constraint in (2) is satisfied.

**Lemma C.4.** *Suppose Assumptions 1 and 4 hold. Then for $\delta > 0$ with $\gamma_i$ chosen according to ($\gamma_i$-Schedule), with probability $1 - \delta$, C-FastCB satisfies the performance constraint in (2).*

Combining all four steps, C-FastCB simultaneously satisfies the performance constraint in (2) with probability $1 - \delta$ and the regret upper-bound in (12), which concludes the proof.

**Lemma C.1.** *Suppose Assumption 1,2, 4 holds. Then, the number of times the baseline action is played by C-FastCB is bounded as*

$$n_T \leq \frac{1}{\alpha y_l} \Bigg\{ - (m_{\tau-1} + 1)(\Delta_l + \alpha y_l)$$

$$+ 60 \sqrt{ K \text{Reg}_{\text{KL}}(T) \log \left( \sum_{i \in \mathcal{S}_{\tau-1}} h(\mathbf{x}_{i,a_i^*}) \right) \left( \sum_{i \in \mathcal{S}_{\tau-1}} h(\mathbf{x}_{i,a_i^*}) + 1 \right)} \Bigg\}. \tag{25}$$

*Proof.* Let $\tau$ be the last round at which the algorithm plays the conservative action, i.e.,

$$\tau = \max\{1 \leq t \leq T | a_t = b_t\}.$$

Recall that $m_t = |\mathcal{S}_t|$ and $n_t = |\mathcal{S}_t^c|$. By the definition of $\tau$, we have that at round $\tau$

$$\hat{y}_{\tau,\tilde{a}_\tau} + \sum_{i \in \mathcal{S}_{\tau-1}} \sum_{a \in [K]} p_{i,a} \hat{y}_{i,a} + \sum_{i \in \mathcal{S}_{\tau-1}^c} h(\mathbf{x}_{i,b_i}) + 16 \sqrt{m_{\tau-1} \text{Reg}_{\text{KL}}(T)}$$

$$> (1 + \alpha) \sum_{i=1}^{\tau} h(\mathbf{x}_{i,b_i}).$$

Therefore, we may write

$$\alpha \sum_{i=1}^{\tau} h(\mathbf{x}_{i,b_i}) < \sum_{i \in \mathcal{S}_{\tau-1}} \sum_{a \in [K]} p_{i,a} \hat{y}_{i,a} + \hat{y}_{\tau,\tilde{a}_\tau} - \sum_{i \in \mathcal{S}_{\tau-1}} (h(\mathbf{x}_{i,b_i}) + h(\mathbf{x}_{\tau,b_\tau}))$$

$$= \sum_{i \in \mathcal{S}_{\tau-1}} \sum_{a \in [K]} p_{i,a} \hat{y}_{i,a} + \hat{y}_{\tau,\tilde{a}_\tau} - \sum_{i \in \mathcal{S}_{\tau-1}} \sum_{a \in [K]} p_{i,a} h(\mathbf{x}_{i,a_i^*})$$

$$+ \sum_{i \in \mathcal{S}_{\tau-1}} \sum_{a \in [K]} p_{i,a} h(\mathbf{x}_{i,a_i^*}) + \sum_{a \in [K]} p_{\tau,a} h(\mathbf{x}_{\tau,a_\tau^*})$$

$$- \sum_{a \in [K]} p_{\tau,a} h(\mathbf{x}_{\tau,a_\tau^*}) - \sum_{i \in \mathcal{S}_{\tau-1}} (h(\mathbf{x}_{i,b_i}) + h(\mathbf{x}_{\tau,b_\tau}))$$

$$+ 16 \sqrt{m_{\tau-1} \text{Reg}_{\text{KL}}(T)}$$

$$= \underbrace{\sum_{i \in \mathcal{S}_{\tau-1}} \left( h(\mathbf{x}_{i,a_i^*}) - h(\mathbf{x}_{i,b_i}) \right) + \left( h(\mathbf{x}_{\tau,a_\tau^*}) - h(\mathbf{x}_{\tau,b_\tau}) \right)}_{I}$$

$$+ \underbrace{\sum_{i \in \mathcal{S}_{\tau-1}} \sum_{a \in [K]} p_{i,a} \left( \hat{y}_{i,a} - h(\mathbf{x}_{i,a_i^*}) \right)}_{II} + \underbrace{\sum_{a \in [K]} p_{\tau,a} \left( \hat{y}_{\tau,\tilde{a}_\tau} - h(\mathbf{x}_{\tau,a_\tau^*}) \right)}_{III} \tag{29}$$

$$+ 16 \sqrt{m_{\tau-1} (\text{Reg}_{\text{KL}}(T) + \log(2/\delta))}$$

First consider term $I$. Using Assumption 2 we have that $\Delta_l \leq h(\mathbf{x}_{i,a_i^*}) - h(\mathbf{x}_{i,b_i}) \leq \Delta_h$. Also recall that $m_{\tau-1} = |S_{\tau-1}|$. Combining these we have:

$$\sum_{i \in \mathcal{S}_{\tau-1}} \left( h(\mathbf{x}_{i,a_i^*}) - h(\mathbf{x}_{i,b_i}) \right) + \left( h(\mathbf{x}_{\tau,a_\tau^*}) - h(\mathbf{x}_{\tau,b_\tau}) \right) < -(m_{\tau-1} + 1)\Delta_l$$

Next consider term $II$. Adding and subtracting $h(\mathbf{x}_{i,a})$, we obtain

$$\sum_{i \in S_{\tau-1}} \sum_{a \in [K]} p_{i,a}\Big(\hat{y}_{i,a} - h(\mathbf{x}_{i,a_i^*})\Big) = \underbrace{\sum_{i \in S_{\tau-1}} \sum_{a \in [K]} p_{i,a}\Big(h(\mathbf{x}_{i,a}) - h(\mathbf{x}_{i,a_i^*})\Big)}_{II(a)}$$

$$+ \underbrace{\sum_{i \in S_{\tau-1}} \sum_{a \in [K]} p_{i,a}\Big(\hat{y}_{i,a} - h(\mathbf{x}_{i,a})\Big)}_{II(b)}$$

Using the AM-GM inequality we can bound term $II(a)$ as follows:

$$\sum_{i \in S_{\tau-1}} \sum_{a \in [K]} p_{i,a}\Big(\hat{y}_{i,a} - h(\mathbf{x}_{i,a_i^*})\Big) \le \sum_{i \in S_{\tau-1}} \sum_{a \in [K]} p_{i,a}\left(\frac{1}{4\beta}(\hat{y}_{i,a} - h(\mathbf{x}_{i,a_i^*})) + \beta \frac{(\hat{y}_{i,a} - h(\mathbf{x}_{i,a_i^*}))^2}{\hat{y}_{i,a} + h(\mathbf{x}_{i,a_i^*})}\right)$$

for any $\beta > 1$. Using Lemma 5 in (Foster & Krishnamurthy, 2021) we have

$$\sum_{i \in S_{\tau-1}} \sum_{a \in [K]} p_{i,a}\hat{y}_{i,a} \le 3 \sum_{a \in [K]} p_{i,a}h(\mathbf{x}_{i,a_i^*}) + \sum_{a \in [K]} p_{i,a}\frac{(\hat{y}_{i,a} - h(\mathbf{x}_{i,a_i^*}))^2}{\hat{y}_{i,a} + h(\mathbf{x}_{i,a_i^*})}$$

Therefore we have the following bound on term $II(b)$:

$$\sum_{i \in S_{\tau-1}} \sum_{a \in [K]} p_{i,a}\Big(\hat{y}_{i,a} - h(\mathbf{x}_{i,a_i^*})\Big) \le \frac{1}{\beta} \sum_{i \in S_{\tau-1}} \sum_{a \in [K]} p_{i,a}h(\mathbf{x}_{i,a_i^*}) + 2\beta \sum_{i \in S_{\tau-1}} \sum_{a \in [K]} p_{i,a}\frac{(\hat{y}_{i,a} - h(\mathbf{x}_{i,a_i^*}))^2}{\hat{y}_{i,a} + h(\mathbf{x}_{i,a_i^*})}$$

Using Proposition 5 from Foster & Krishnamurthy (2021) we have

$$2\beta \sum_{i \in S_{\tau-1}} \sum_{a \in [K]} p_{i,a}\frac{(\hat{y}_{i,a} - h(\mathbf{x}_{i,a_i^*}))^2}{\hat{y}_{i,a} + h(\mathbf{x}_{i,a_i^*})} \le 4\beta \text{Reg}_{\text{KL}}(T)$$

and therefore,

$$\sum_{i \in S_{\tau-1}} \sum_{a \in [K]} p_{i,a}\Big(\hat{y}_{i,a_i^*} - h(\mathbf{x}_{i,a_i^*})\Big) \le \frac{1}{\beta} \sum_{i \in S_{\tau-1}} \sum_{a \in [K]} p_{i,a}h(\mathbf{x}_{i,a_i^*}) + 4\beta \text{Reg}_{\text{KL}}(T)$$

$$= \frac{1}{\beta} \sum_{i \in S_{\tau-1}} h(\mathbf{x}_{i,a_i^*}) + 4\beta \text{Reg}_{\text{KL}}(T)$$

Choosing $\beta = \sqrt{\dfrac{\sum_{i \in S_{\tau-1}} h(\mathbf{x}_{i,a_i^*})}{\text{Reg}_{\text{KL}}(T)}}$ we have

$$\sum_{i \in S_{\tau-1}} \sum_{a \in [K]} p_{i,a}\Big(\hat{y}_{i,a_i^*} - h(\mathbf{x}_{i,a_i^*})\Big) \le 4\sqrt{\sum_{i \in S_{\tau-1}} h(\mathbf{x}_{i,a_i^*})\text{Reg}_{\text{KL}}(T)} \tag{30}$$

Next consider term $II(a)$. We use the per round regret guarantee (Theorem 4) from Foster & Krishnamurthy (2021) as follows:

$$\sum_{a \in [K]} p_{i,a}\Big(h(\mathbf{x}_{i,a_i}) - h(\mathbf{x}_{i,a_i^*})\Big) \le \frac{5K}{\gamma_i} \sum_{a \in [K]} p_{i,a}h(\mathbf{x}_{i,a_i}) + 7\gamma_i \sum_{a \in [K]} p_{i,a}\frac{(\hat{y}_{i,a} - h(\mathbf{x}_{i,a_i}))^2}{\hat{y}_{i,a} + h(\mathbf{x}_{i,a_i})} \tag{31}$$

Adding and subtracting $h(\mathbf{x}_{i,a_i^*})$ we get

$$\sum_{a \in [K]} p_{i,a}\Big(h(\mathbf{x}_{i,a_i^*}) - h(\mathbf{x}_{i,a_i^*})\Big) \le \frac{5K}{\gamma_i} \sum_{a \in [K]} p_{i,a}h(\mathbf{x}_{i,a_i^*}) + \frac{5K}{\gamma_i} \sum_{a \in [K]} p_{i,a}(h(\mathbf{x}_{i,a_i}) - h(\mathbf{x}_{i,a_i^*}))$$

$$+ 7\gamma_i \sum_{a \in [K]} p_{i,a}\frac{(\hat{y}_{i,a} - h(\mathbf{x}_{i,a_i}))^2}{\hat{y}_{i,a} + h(\mathbf{x}_{i,a_i})},$$

and therefore

$$\left(1 - \frac{5K}{\gamma_i}\right) \sum_{a \in [K]} p_{i,a}\left(h(\mathbf{x}_{i,a_i}) - h(\mathbf{x}_{i,a_i^*})\right) \leq \frac{5K}{\gamma_i} \sum_{a \in [K]} p_{i,a}h(\mathbf{x}_{i,a_i^*}) + 7\gamma_i \sum_{a \in [K]} p_{i,a} \frac{(\hat{y}_{i,a} - h(\mathbf{x}_{i,a_i}))^2}{\hat{y}_{i,a} + h(\mathbf{x}_{i,a_i})}$$

Using $\gamma_i \geq 10K$ we have

$$\sum_{a \in [K]} p_{i,a}\left(h(\mathbf{x}_{i,a_i}) - h(\mathbf{x}_{i,a_i^*})\right) \leq \frac{10K}{\gamma_i} \sum_{a \in [K]} p_{i,a}h(\mathbf{x}_{i,a_i^*}) + 14\gamma_i \sum_{a \in [K]} p_{i,a} \frac{(\hat{y}_{i,a} - h(\mathbf{x}_{i,a_i}))^2}{\hat{y}_{i,a} + h(\mathbf{x}_{i,a_i})} \tag{32}$$

Recall that we set the exploration factor $\gamma_i$ as follows:

$$\gamma_0 = 1, \eta_0 = 1, L_i^* = 0$$
$$\text{for } i \in \mathcal{S}_T$$
$$L_i^* = L_{i-1}^* + h(\mathbf{x}_{t,a_i^*})$$
$$\text{if } \quad L_i^* > 2\eta_{i-1}$$
$$\eta_i = 2\eta_{i-1}$$
$$\text{else}$$
$$\eta_i = \eta_{i-1}$$
$$\gamma_i = \max\left(10K, \sqrt{\frac{K\eta_i}{\mathtt{Reg}_{\mathtt{KL}}(T)}}\right)$$

Note that according to the above schedule of $\gamma_i$ there are $E = \log\left(\sum_{i \in \mathcal{S}_{\tau-1}} L_i^*\right)$ episodes, that we denote by $T_e(\mathcal{S}_{\tau-1})$, $e \in [E]$, $\eta_i = \eta_e$ and $\eta_i := \eta_e$ is constant for all $i \in T_e(\mathcal{S}_{\tau-1})$ with the following guarantee

$$\sum_{i \in T_e(\mathcal{S}_{\tau-1})} h(\mathbf{x}_{i,a_i^*}) \leq \eta_e \leq 2 \sum_{i \in T_e(\mathcal{S}_{\tau-1})} h(\mathbf{x}_{i,a_i^*}) \tag{33}$$

Therefore summing up the inequality in (32) for $i \in \mathcal{S}_{\tau-1}$ we get

$$\sum_{i \in \mathcal{S}_{\tau-1}} \sum_{a \in [K]} p_{i,a}\left(h(\mathbf{x}_{i,a_i}) - h(\mathbf{x}_{i,a_i^*})\right) \leq \sum_{i \in \mathcal{S}_{\tau-1}} \frac{10K}{\gamma_i} \sum_{a \in [K]} p_{i,a}h(\mathbf{x}_{i,a_i^*})$$

$$+ \sum_{i \in \mathcal{S}_{\tau-1}} 14\gamma_i \sum_{a \in [K]} p_{i,a} \frac{(\hat{y}_{i,a} - h(\mathbf{x}_{i,a_i}))^2}{\hat{y}_{i,a} + h(\mathbf{x}_{i,a_i})}$$

$$\overset{(a)}{\leq} \sum_{e=1}^{E} \sum_{i \in T_e(\mathcal{S}_{\tau-1})} \frac{10K}{\gamma_i} \sum_{a \in [K]} p_{i,a}h(\mathbf{x}_{i,a_i^*})$$

$$+ 14(\max_{i \in \mathcal{S}_{\tau-1}} \gamma_i) \sum_{i \in \mathcal{S}_{\tau-1}} \sum_{a \in [K]} p_{i,a} \frac{(\hat{y}_{i,a} - h(\mathbf{x}_{i,a_i}))^2}{\hat{y}_{i,a} + h(\mathbf{x}_{i,a_i})}$$

$$\overset{(b)}{=} \sum_{e=1}^{E} \frac{10K}{\gamma_e} \sum_{i \in T_e(\mathcal{S}_{\tau-1})} \sum_{a \in [K]} p_{i,a}h(\mathbf{x}_{i,a_i^*})$$

$$+ 14(\max_{i \in \mathcal{S}_{\tau-1}} \gamma_i) \sum_{i \in \mathcal{S}_{\tau-1}} \sum_{a \in [K]} p_{i,a} \frac{(\hat{y}_{i,a} - h(\mathbf{x}_{i,a_i}))^2}{\hat{y}_{i,a} + h(\mathbf{x}_{i,a_i})}$$

$$\overset{(c)}{=} \sum_{e=1}^{E} 10K\sqrt{\frac{\mathtt{Reg}_{\mathtt{KL}}(T)}{K \sum_{i \in T_e(\mathcal{S}_{\tau-1})} h(\mathbf{x}_{i,a_i^*})}} \sum_{i \in T_e(\mathcal{S}_{\tau-1})} h(\mathbf{x}_{i,a_i^*})$$

$$+ 14(\max_{i \in \mathcal{S}_{\tau-1}} \gamma_i) \sum_{i \in \mathcal{S}_{\tau-1}} \sum_{a \in [K]} p_{i,a} \frac{(\hat{y}_{i,a} - h(\mathbf{x}_{i,a_i}))^2}{\hat{y}_{i,a} + h(\mathbf{x}_{i,a_i})},$$

where $(a)$ follows by changing the sum in $i \in \mathcal{S}_{\tau-1}$ to $\sum_{e=1}^{E} \sum_{i \in T_e(\mathcal{S}_{\tau-1})}$ and noting that $\max_{i \in \mathcal{S}_{\tau-1}} \gamma_i \geq \gamma_i$ for all $i \in \mathcal{S}_{\tau-1}$. Next $(b)$ follows because $\gamma_i$ is constant within an episode $e \in [E]$. Finally $(c)$ follows by our choice of $\gamma_i$ from ($\gamma_{\texttt{i}}$-Schedule) and the property in (38). Therefore we have

$$\sum_{i \in \mathcal{S}_{\tau-1}} \sum_{a \in [K]} p_{i,a}\Big(h(\mathbf{x}_{i,a_i}) - h(\mathbf{x}_{i,a_i^*})\Big) \overset{(d)}{\leq} \sum_{e=1}^{E} 10K\sqrt{\frac{\text{Reg}_{\text{KL}}(T)}{K\sum_{i \in T_e(\mathcal{S}_{\tau-1})} h(\mathbf{x}_{i,a_i^*})}} \sum_{i \in T_e(\mathcal{S}_{\tau-1})} h(\mathbf{x}_{i,a_i^*})$$

$$+ 14\sqrt{K\frac{\sum_{i \in \mathcal{S}_{\tau-1}} h(\mathbf{x}_{i,a_{i^*}})}{\text{Reg}_{\text{KL}}(T)}} \sum_{i \in \mathcal{S}_{\tau-1}} \sum_{a \in [K]} p_{i,a}\frac{(\hat{y}_{i,a} - h(\mathbf{x}_{i,a_i}))^2}{\hat{y}_{i,a} + h(\mathbf{x}_{i,a_i})}$$

$$\overset{(e)}{\leq} 10\sum_{e=1}^{E} \sqrt{K\text{Reg}_{\text{KL}}(T)\sum_{i \in T_e(\mathcal{S}_{\tau-1})} h(\mathbf{x}_{i,a_i^*})}$$

$$+ 14\sqrt{K\frac{\sum_{i \in \mathcal{S}_{\tau-1}} h(\mathbf{x}_{i,a_{i^*}})}{\text{Reg}_{\text{KL}}(T)}\text{Reg}_{\text{KL}}(m_{\tau-1})}$$

$$\overset{(f)}{\leq} 10\sqrt{KE\text{Reg}_{\text{KL}}(T)\sum_{e=1}^{E}\sum_{i \in T_e(\mathcal{S}_{\tau-1})} h(\mathbf{x}_{i,a_i^*})}$$

$$+ 14\sqrt{K\sum_{i \in \mathcal{S}_{\tau-1}} h(\mathbf{x}_{i,a_{i^*}})\text{Reg}_{\text{KL}}(m_{\tau-1})},$$

where $(d)$ again follows by our choice of $\gamma_i$ and (38), $(e)$ follows by Proposition 5 of Foster & Krishnamurthy (2021) and $(f)$ follows by Cauchy-Schwarz inequality. Finally we arrive at the following bound

$$\sum_{i \in \mathcal{S}_{\tau-1}} \sum_{a \in [K]} p_{i,a}\Big(h(\mathbf{x}_{i,a_i}) - h(\mathbf{x}_{i,a_i^*})\Big)$$

$$\leq 25\sqrt{K\text{Reg}_{\text{KL}}(T)\log\left(\sum_{i \in \mathcal{S}_{\tau-1}} h(\mathbf{x}_{i,a_i^*})\right)\sum_{i \in \mathcal{S}_{\tau-1}} h(\mathbf{x}_{i,a_i^*})} \qquad (34)$$

and combining with (30) we have the following bound on term $II$:

$$\sum_{i \in \mathcal{S}_{\tau-1}} \sum_{a \in [K]} p_{i,a}\Big(\hat{y}_{i,a} - h(\mathbf{x}_{i,a_i^*})\Big) \leq 30\sqrt{K\text{Reg}_{\text{KL}}(T)\log\left(\sum_{i \in \mathcal{S}_{\tau-1}} h(\mathbf{x}_{i,a_i^*})\right)\sum_{i \in \mathcal{S}_{\tau-1}} h(\mathbf{x}_{i,a_i^*})}$$

Now consider term $III$. Since $0 \leq h(x_{i,a}), \hat{y}_{i,a} \leq 1$ we have that

$$\sum_{a \in [K]} \hat{y}_{\tau,\tilde{a}_\tau} - p_{\tau,a}h(\mathbf{x}_{\tau,a_\tau^*}) = \sum_{a \in [K]} p_{\tau,a}\big(\hat{y}_{\tau,\tilde{a}_\tau} - h(\mathbf{x}_{\tau,a_\tau^*})\big) \leq 2$$

Combining all the bounds we get for $K \geq 2$ and $\text{Reg}_{\text{KL}}(T) \geq 1$ we have

$$\alpha\sum_{i=1}^{\tau} h(\mathbf{x}_{i,b_i}) \leq -(m_{\tau-1} + 1)\Delta_l$$

$$+ 30\sqrt{K\text{Reg}_{\text{KL}}(T)\log\left(\sum_{i \in \mathcal{S}_{\tau-1}} h(\mathbf{x}_{i,a_i^*})\right)\left(\sum_{i \in \mathcal{S}_{\tau-1}} h(\mathbf{x}_{i,a_i^*}) + 1\right)} \qquad (35)$$

Now, note that $m_{\tau-1} + n_{\tau-1} + 1 = \tau$, and using Assumption 2 we have $y_l \leq h(\mathbf{x}_{i,b_i}) \leq y_h, \forall i \in [T]$. Therefore

$$\alpha\sum_{i=1}^{\tau} h(\mathbf{x}_{i,b_i}) \geq \alpha\left(m_{\tau-1} + n_{\tau-1} + 1\right)y_l.$$

Combining with (35) and noting that $n_T = n_\tau = n_{\tau-1} + 1$ we have

$$n_T \leq \frac{1}{\alpha y_l} \Bigg\{ -(m_{\tau-1} + 1)(\Delta_l + \alpha y_l)$$

$$+ 60 \sqrt{K \text{Reg}_{\text{KL}}(T) \log \left( \sum_{i \in \mathcal{S}_{\tau-1}} h(\mathbf{x}_{i,a_i^*}) \right) \left( \sum_{i \in \mathcal{S}_{\tau-1}} h(\mathbf{x}_{i,a_i^*}) + 1 \right)} \Bigg\}$$

$\square$

**Lemma C.2.** *Suppose Assumption Assumption 1,2, 4 holds. Then the number of times the baseline action is played by* `C-SquareCB` *is bounded as follows:*

$$n_T \leq \mathcal{O}\left( \frac{K \text{Reg}_{\text{KL}}(T)}{\alpha y_l (\Delta_l + \alpha y_l)} \log \left( \frac{e \sqrt{K \text{Reg}_{\text{KL}}(T)}}{\Delta_l + \alpha y_l} \right) \right). \tag{26}$$

*Proof.* Note that we have from Lemma C.1 and using the fact that $h(\cdot) \in [0,1]$ we

$$n_T \leq \frac{1}{\alpha y_l} \Bigg\{ -(m_{\tau-1} + 1)(\Delta_l + \alpha y_l)$$

$$+ 60 \sqrt{K \text{Reg}_{\text{KL}}(T) \log \left( \sum_{i \in \mathcal{S}_{\tau-1}} h(\mathbf{x}_{i,a_i^*}) \right) \left( \sum_{i \in \mathcal{S}_{\tau-1}} h(\mathbf{x}_{i,a_i^*}) + 1 \right)} \Bigg\}$$

$$\leq \frac{1}{\alpha y_l} \Bigg\{ -(m_{\tau-1} + 1)(\Delta_l + \alpha y_l) + 60 \sqrt{K \text{Reg}_{\text{KL}}(T) \log (m_{\tau-1}) (m_{\tau-1} + 1)} \Bigg\}$$

We define $Q(m) := -m\,c_1 + \sqrt{m \log(m)}\,c_2$ where

$$c_1 = \Delta_l + \alpha y_l \geq 0,$$
$$c_2 = 60\sqrt{K \text{Reg}_{\text{KL}}(T)},$$
$$m = m_{\tau-1} + 1$$

Next observe that for $m \geq 3$, we have $-m\,c_1 + \sqrt{m \log(m)}\,c_2 \leq -m\,c_1 + \sqrt{m} \log m\,c_2$. Now we use Lemma 8 from Kazerouni et al. (2017) to conclude that

$$-m\,c_1 + \sqrt{m} \log m\,c_2 \leq \frac{16 c_2^2}{9 c_1} \left[ \log \left( \frac{2 c_2 e}{c_1} \right) \right]^2$$

$$= \mathcal{O}\left( \frac{K \text{Reg}_{\text{KL}}(T)}{\Delta_l + \alpha y_l} \log \left( \frac{e \sqrt{K \text{Reg}_{\text{KL}}(T)}}{\Delta_l + \alpha y_l} \right) \right)$$

which completes the proof. $\square$

**Lemma C.3.** *Suppose Assumptions 1 and 4 hold. Then for $\gamma_t$ chosen as in ($\gamma_i$-Schedule), we have*

$$\mathbb{E} \sum_{t \in \mathcal{S}_T} \left( h(\mathbf{x}_{t,a_t}) - h(\mathbf{x}_{t,a_t^*}) \right) \leq \mathcal{O}\left( \sqrt{K \text{Reg}_{\text{KL}}(T) \log \left( L_{\mathcal{S}_T}^* \right) L_{\mathcal{S}_T}^*} \right). \tag{28}$$

*where $L_{\mathcal{S}_T}^* = \sum_{t \in \mathcal{S}_T} h(\mathbf{x}_{t,a_t^*})$ is the cumulative cost of the optimal policy in the subset $\mathcal{S}_T$.*

*Proof.* The proof follows along similar lines as term $II(a)$ in the proof of Lemma C.1 and is provided here for completeness. We use the per round regret guarantee (Theorem 4) from Foster &

Krishnamurthy (2021) as follows:

$$\sum_{a\in[K]} p_{i,a}\Big(h(\mathbf{x}_{i,a_i}) - h(\mathbf{x}_{i,a_i^*})\Big) \leq \frac{5K}{\gamma_i}\sum_{a\in[K]} p_{i,a}h(\mathbf{x}_{i,a_i}) + 7\gamma_i\sum_{a\in[K]} p_{i,a}\frac{(\hat{y}_{i,a} - h(\mathbf{x}_{i,a_i}))^2}{\hat{y}_{i,a} + h(\mathbf{x}_{i,a_i})} \tag{36}$$

Adding and subtracting $h(\mathbf{x}_{i,a_i^*})$ we get

$$\sum_{a\in[K]} p_{i,a}\Big(h(\mathbf{x}_{i,a_i^*}) - h(\mathbf{x}_{i,a_i^*})\Big) \leq \frac{5K}{\gamma_i}\sum_{a\in[K]} p_{i,a}h(\mathbf{x}_{i,a_i^*}) + \frac{5K}{\gamma_i}\sum_{a\in[K]} p_{i,a}(h(\mathbf{x}_{i,a_i}) - h(\mathbf{x}_{i,a_i^*}))$$
$$+ 7\gamma_i\sum_{a\in[K]} p_{i,a}\frac{(\hat{y}_{i,a} - h(\mathbf{x}_{i,a_i}))^2}{\hat{y}_{i,a} + h(\mathbf{x}_{i,a_i})},$$

and therefore

$$\left(1 - \frac{5K}{\gamma_i}\right)\sum_{a\in[K]} p_{i,a}\Big(h(\mathbf{x}_{i,a_i}) - h(\mathbf{x}_{i,a_i^*})\Big) \leq \frac{5K}{\gamma_i}\sum_{a\in[K]} p_{i,a}h(\mathbf{x}_{i,a_i^*}) + 7\gamma_i\sum_{a\in[K]} p_{i,a}\frac{(\hat{y}_{i,a} - h(\mathbf{x}_{i,a_i}))^2}{\hat{y}_{i,a} + h(\mathbf{x}_{i,a_i})}$$

Using $\gamma_i \geq 10K$ we have

$$\sum_{a\in[K]} p_{i,a}\Big(h(\mathbf{x}_{i,a_i}) - h(\mathbf{x}_{i,a_i^*})\Big) \leq \frac{10K}{\gamma_i}\sum_{a\in[K]} p_{i,a}h(\mathbf{x}_{i,a_i^*}) + 14\gamma_i\sum_{a\in[K]} p_{i,a}\frac{(\hat{y}_{i,a} - h(\mathbf{x}_{i,a_i}))^2}{\hat{y}_{i,a} + h(\mathbf{x}_{i,a_i})} \tag{37}$$

Using the schedule of $\gamma_i$ from ($\gamma_i$-Schedule), there are $E = \log\left(\sum_{i\in\mathcal{S}_T} L_i^*\right)$ episodes, that we denote by $T_e(\mathcal{S}_T)$, $e \in [E]$, $\eta_i = \eta_e$ and $\eta_i := \eta_e$ is constant for all $i \in T_e(\mathcal{S}_T)$ with the following guarantee

$$\sum_{i\in T_e(\mathcal{S}_T)} h(\mathbf{x}_{i,a_i^*}) \leq \eta_e \leq 2\sum_{i\in T_e(\mathcal{S}_T)} h(\mathbf{x}_{i,a_i^*}) \tag{38}$$

Now summing for $i \in \mathcal{S}_T$ as in the proof of Lemma C.1 (cf. equation (34)) we obtain

$$\sum_{i\in\mathcal{S}_T}\sum_{a\in[K]} p_{i,a}\Big(h(\mathbf{x}_{i,a_i}) - h(\mathbf{x}_{i,a_i^*})\Big)$$

$$\leq 25\sqrt{K\text{Reg}_{\text{KL}}(T)\log\left(\sum_{i\in\mathcal{S}_T} h(\mathbf{x}_{i,a_i^*})\right)\sum_{i\in\mathcal{S}_T} h(\mathbf{x}_{i,a_i^*})}$$

$$= 25\sqrt{K\text{Reg}_{\text{KL}}(T)\log\left(L_{\mathcal{S}_T}^*\right)L_{\mathcal{S}_T}^*}$$

where $L_{\mathcal{S}_T}^* = \sum_{t\in\mathcal{S}_T} h(\mathbf{x}_{t,a_t^*})$. Define the following filtration

$$\mathcal{F}_{t-1} = \sigma\Big((\mathbf{x}_{i,a}, \tilde{a}_i, y_{i,\tilde{a}_i}), \mathbf{x}_{t,a}; 1 \leq i \leq t-1, a \in [K]\Big).$$

Note that $\mathbb{E}\left[h(\mathbf{x}_{t,\tilde{a}_t})|\mathcal{F}_{t-1}\right] = \sum_{a\in[K]} p_{t,a}h(\mathbf{x}_{t,a})$ and therefore

$$\mathbb{E}\sum_{t\in\mathcal{S}_T}\Big(h(\mathbf{x}_{t,a_t}) - h(\mathbf{x}_{t,a_t^*})\Big) = \sum_{i\in\mathcal{S}_T}\sum_{a\in[K]} p_{i,a}\Big(h(\mathbf{x}_{i,a_i}) - h(\mathbf{x}_{i,a_i^*})\Big)$$

which completes the proof. $\qquad\square$

**Lemma 3.5.** *Let Assumptions 1, 2 and 3 hold. Then, for $\delta > 0$ and $\gamma_t = \sqrt{K|\mathcal{S}_t|/(\text{Reg}_{\text{Sq}}(m_T) + \log(8\delta^{-1}))}$, with probability $1 - \delta/2$, C-SquareCB satisfies the performance constraint in (2).*

*Proof.* For $t = 1$ if the condition in line 8 holds then $\tilde{a}_1 = a_1$ and we have that with probability $1 - \delta$

$$\hat{y}_{1,a_1} - 16\sqrt{(m_0 + 1)(1 + \log(1/\delta))} \leq (1 + \alpha)h(\mathbf{x}_{1,b_1})$$

Noting that $|\hat{y}_{1,a_1} - h(\mathbf{x}_{i,a_1})| \leq 2$ and therefore with probability $1 - \delta$

$$h(\mathbf{x}_{i,a_1}) \leq (1 + \alpha)h(\mathbf{x}_{1,b_1}).$$

Further, if the condition in line 8 doesn't hold, then $a_1 = b_1$, and therefore

$$h(\mathbf{x}_{i,a_1}) \leq (1 + \alpha)h(\mathbf{x}_{1,b_1}),$$

showing that the performance constraint in Definition 2.2 is satisfied. Now assume that the constraint holds for $t - 1$ and now consider $t \in [T]$. Note that

$$\left| \sum_{i \in \mathcal{S}_{t-1}} \sum_{a \in [K]} p_{i,a}\hat{y}_{i,a} - h(\mathbf{x}_{i,\tilde{a}_i}) \right| \leq \underbrace{\left| \sum_{i \in \mathcal{S}_{t-1}} \sum_{a \in [K]} p_{i,a}\hat{y}_{i,a} - \sum_{i \in \mathcal{S}_{t-1}} \sum_{a \in [K]} p_{i,a}h(\mathbf{x}_{i,a}) \right|}_{I}$$

$$+ \underbrace{\left| \sum_{i \in \mathcal{S}_{t-1}} \sum_{a \in [K]} p_{i,a}h(\mathbf{x}_{i,a}) - h(\mathbf{x}_{i,\tilde{a}_i}) \right|}_{II}$$

Consider term $I$. We handle it as in the proof of Lemma C.1 as follows: Using the AM-GM inequality,

$$\sum_{i \in S_{t-1}} \sum_{a \in [K]} p_{i,a}\left(\hat{y}_{i,a} - h(\mathbf{x}_{i,a_i^*})\right) \leq \sum_{i \in S_{t-1}} \sum_{a \in [K]} p_{i,a}\left( \frac{1}{4\beta}(\hat{y}_{i,a} - h(\mathbf{x}_{i,a_i^*})) + \beta \frac{(\hat{y}_{i,a} - h(\mathbf{x}_{i,a_i^*}))^2}{\hat{y}_{i,a} + h(\mathbf{x}_{i,a_i^*})} \right)$$

for any $\beta > 1$. Using Lemma 5 in (Foster & Krishnamurthy, 2021) we have

$$\sum_{i \in S_{t-1}} \sum_{a \in [K]} p_{i,a}\hat{y}_{i,a} \leq 3 \sum_{a \in [K]} p_{i,a}h(\mathbf{x}_{i,a_i^*}) + \sum_{a \in [K]} p_{i,a} \frac{(\hat{y}_{i,a} - h(\mathbf{x}_{i,a_i^*}))^2}{\hat{y}_{i,a} + h(\mathbf{x}_{i,a_i^*})}$$

Therefore we have the following bound on term $II(b)$:

$$\sum_{i \in S_{t-1}} \sum_{a \in [K]} p_{i,a}\left(\hat{y}_{i,a} - h(\mathbf{x}_{i,a_i^*})\right) \leq \frac{1}{\beta} \sum_{i \in S_{t-1}} \sum_{a \in [K]} p_{i,a}h(\mathbf{x}_{i,a_i^*}) + 2\beta \sum_{i \in S_{t-1}} \sum_{a \in [K]} p_{i,a} \frac{(\hat{y}_{i,a} - h(\mathbf{x}_{i,a_i^*}))^2}{\hat{y}_{i,a} + h(\mathbf{x}_{i,a_i^*})}$$

Using Proposition 5 from Foster & Krishnamurthy (2021) we have

$$2\beta \sum_{i \in S_{t-1}} \sum_{a \in [K]} p_{i,a} \frac{(\hat{y}_{i,a} - h(\mathbf{x}_{i,a_i^*}))^2}{\hat{y}_{i,a} + h(\mathbf{x}_{i,a_i^*})} \leq 4\beta \texttt{Reg}_{\texttt{KL}}(T)$$

and therefore,

$$\sum_{i \in S_{t-1}} \sum_{a \in [K]} p_{i,a}\left(\hat{y}_{i,a_i^*} - h(\mathbf{x}_{i,a_i^*})\right) \leq \frac{1}{\beta} \sum_{i \in S_{t-1}} \sum_{a \in [K]} p_{i,a}h(\mathbf{x}_{i,a_i^*}) + 4\beta \texttt{Reg}_{\texttt{KL}}(T)$$

$$= \frac{1}{\beta} \sum_{i \in S_{t-1}} h(\mathbf{x}_{i,a_i^*}) + 4\beta \texttt{Reg}_{\texttt{KL}}(T)$$

Choosing $\beta = \sqrt{\dfrac{\sum_{i \in S_{t-1}} h(\mathbf{x}_{i,a_i^*})}{\texttt{Reg}_{\texttt{KL}}(T)}}$ we have

$$\sum_{i \in S_{t-1}} \sum_{a \in [K]} p_{i,a}\left(\hat{y}_{i,a_i^*} - h(\mathbf{x}_{i,a_i^*})\right) \leq 4\sqrt{\sum_{i \in S_{t-1}} h(\mathbf{x}_{i,a_i^*})\texttt{Reg}_{\texttt{KL}}(T)}$$

Using the fact that $h(\cdot) \leq 1$ we have

$$\sum_{i \in S_{t-1}} \sum_{a \in [K]} p_{i,a}\left(\hat{y}_{i,a_i^*} - h(\mathbf{x}_{i,a_i^*})\right) \leq 4\sqrt{m_{t-1}\texttt{Reg}_{\texttt{KL}}(T)}$$

Next, consider term $II$. Consider the following filtration

$$\mathcal{F}_{t-1} = \sigma\bigg( (\mathbf{x}_{i,a}, \tilde{a}_i, y_{i,\tilde{a}_i}), \mathbf{x}_{t,a}; 1 \le i \le t-1, a \in [K] \bigg).$$

Note that $\mathbb{E}\left[ h(\mathbf{x}_{t,\tilde{a}_t}) | \mathcal{F}_{t-1} \right] = \sum_{a \in [K]} p_{t,a} h(\mathbf{x}_{t,a})$, and therefore using Azuma-Hoeffding we have that with probability $1 - \delta$

$$\left| \sum_{i \in \mathcal{S}_{t-1}} \sum_{a \in [K]} p_{i,a} h(\mathbf{x}_{i,a}) - h(\mathbf{x}_{i,\tilde{a}_i}) \right| \le 2\sqrt{m_{t-1} \log(2\delta^{-1})} \tag{39}$$

Combining (22) and (39) and taking a union bound we have with probability $1 - \delta$

$$\left| \sum_{i \in \mathcal{S}_{t-1}} \sum_{a \in [K]} p_{i,a} \hat{y}_{i,a} - h(\mathbf{x}_{i,\tilde{a}_i}) \right| \le 8\sqrt{m_{t-1}\bigg( \texttt{Reg}_{\texttt{KL}}(m_{t-1}) + \log(2/\delta) \bigg)}$$

Further $|\hat{y}_{t,\tilde{a}_t} - h(\mathbf{x}_{t,\tilde{a}_t})| \le 2$, and therefore with probability $1 - \delta$

$$\left| \hat{y}_{t,\tilde{a}_t} + \sum_{i \in \mathcal{S}_{t-1}} \sum_{a \in [K]} p_{i,a} \hat{y}_{i,a} - h(\mathbf{x}_{i,\tilde{a}_i}) - h(\mathbf{x}_{t,\tilde{a}_t}) \right| \le 16\sqrt{m_{t-1}\bigg( \texttt{Reg}_{\texttt{KL}}(m_{t-1}) + \log(2/\delta) \bigg)}. \tag{40}$$

Now if line 8 of Algorithm 2 holds at time step $t$, then we have

$$\hat{y}_{t,\tilde{a}_t} + \sum_{i \in \mathcal{S}_{t-1}} \sum_{a \in [K]} p_{i,a} \hat{y}_{i,a} + \sum_{i \in \mathcal{S}_{t-1}^c} h(\mathbf{x}_{i,b_i}) + 16\sqrt{m_{t-1}\bigg( \texttt{Reg}_{\texttt{KL}}(m_{t-1}) + \log(2/\delta) \bigg)}$$

$$\le \ (1+\alpha) \sum_{i=1}^{t} h(\mathbf{x}_{i,b_t}),$$

and therefore invoking (40), we have with probability $1 - \delta$

$$h(\mathbf{x}_{t,\tilde{a}_t}) + \sum_{i \in \mathcal{S}_{t-1}} h(\mathbf{x}_{i,\tilde{a}_i}) + \sum_{i \in \mathcal{S}_{t-1}^c} h(\mathbf{x}_{i,b_i}) \le (1+\alpha) \sum_{i=1}^{t} h(\mathbf{x}_{i,b_t})$$

Now note that for all $i \in S_{t-1}$, $a_i = \tilde{a}_i$, for all $i \in \mathcal{S}_{t-1}^c$, $a_i = b_i$, and using $\mathcal{S}_{t-1} \cup \mathcal{S}_{t-1}^c = [t-1]$, and the fact that the condition in line 8 is satisfied we have with probability $1 - \delta$

$$h(\mathbf{x}_{t,a_t}) + \sum_{i \in [t-1]} h(\mathbf{x}_{i,a_i}) \le (1+\alpha) \sum_{i=1}^{t} h(\mathbf{x}_{i,b_t}),$$

satisfying the performance condition in Definition 2.2.

Next we consider the case when the condition in line 8 does not hold. Invoking the fact that the performance constraint holds until time $t - 1$, we have with probability $1 - \delta$

$$\sum_{t=1}^{t-1} h(\mathbf{x}_{i,a_i}) \le (1+\alpha) \sum_{i=1}^{t-1} h(\mathbf{x}_{i,b_t})$$

Adding $h(\mathbf{x}_{t,b_t})$ on both sides of the above equation we get

$$h(\mathbf{x}_{t,b_t}) + \sum_{i=1}^{t-1} h(\mathbf{x}_{i,a_i}) \le h(\mathbf{x}_{t,b_t}) + (1+\alpha) \sum_{i=1}^{t-1} h(\mathbf{x}_{i,b_t}).$$

Noting that when condition in line 8 does not hold at step $t$, then $a_t = b_t$ and that $\alpha > 0$, we have with probability $1 - \delta$

$$\sum_{i=1}^{t} h(\mathbf{x}_{i,a_i}) \le (1+\alpha) \sum_{i=1}^{t} h(\mathbf{x}_{i,b_t}),$$

satisfying the performance constraint in Definition 2.2 for step $t$. Using mathematical induction we conclude that the performance constraint holds for all $t \in [T]$, completing the proof. $\square$

$\square$

## D   PROOF FOR REGRET BOUNDS FOR NEURAL CONSERVATIVE BANDITS

**Theorem 5.1** (**Regret bound for Neural** `C-SquareCB`). *We instantiate* `Sq-Alg` *with the predictor* $\hat{y}_{t,a_t} = \tilde{f}^{(S)}\big(\theta; \mathbf{x}_t, \boldsymbol{\varepsilon}^{(1:S)}\big)$ *from* (15) *and update the parameters using OGD in* (17). *Under Assumptions* 1,2, 5 *and* 6 *with* $\gamma_t$ *as in Theorem* 5.1, *step-size sequence* $\{\eta_t\}$, *width* $m$, *perturbation constant* $c_p$, *and projection ball* $B$, *with high probability* $(1 - \mathcal{O}(\delta))$, *the performance constraint in* (2) *is satisfied by* `C-SquareCB` *and the regret is given by*

$$\texttt{Reg}_{\text{CB}}(T) \leq \mathcal{O}\bigg(\sqrt{KT \log T} + \sqrt{KT \log(16\delta^{-1})} + \frac{K(\log T + \log(16\delta^{-1}))}{\alpha y_l(\Delta_l + \alpha y_l)}\bigg).$$

*Proof.* We set the width of the network $m = \max\big(\mathcal{O}(T^5), \mathcal{O}(\frac{4LT}{\delta})\big)$ and the projection set $B = B_{\rho,\rho_1}^{\text{Frob}}(\theta_0)$, the layer-wise Frobenius ball around the initialization $\theta_0$ with radii $\rho, \rho_1$ which is defined as

$$B_{\rho,\rho_1}^{\text{Frob}}(\theta_0) := \{\theta \in \mathbb{R}^p : \|\operatorname{vec}(W^{(l)}) - \operatorname{vec}(W_0^{(l)})\|_2 \leq \rho, l \in [L], \|\mathbf{v} - \mathbf{v}_0\|_2 \leq \rho_1\}. \quad (41)$$

We set $\rho$ and $\rho_1$ according to Theorem 3.2 in Deb et al. (2024a), and the perturbation constant $c_p = \mathcal{O}(\sqrt{\lambda})$, where $\lambda$ is the Lipschitz parameter of the loss. Now, invoking Theorem 3.2 in Deb et al. (2024a) we get with probability $1 - \mathcal{O}(\delta)$ over the randomness of initialization and $\{\boldsymbol{\varepsilon}\}_{s=1}^S$, the regret of projected OGD with loss $\mathcal{L}_{\text{Sq}}^{(S)}\big(y_t, \{\tilde{f}(\theta; \mathbf{x}_t, \boldsymbol{\varepsilon}_s)\}_{s=1}^S\big)$ for online regression with squared loss is bounded by $\mathcal{O}(\log T)$ i.e.,

$$\sum_{t=1}^{T} \ell_{\text{sq}}(\hat{y}_{t,a_t}, y_{t,a_t}) - \inf_{g \in \mathcal{H}} \sum_{t=1}^{T} \ell_{\text{sq}}(g(\mathbf{x}_{t,a_t}), y_{t,a_t}) \leq \mathcal{O}(\log T)$$

Therefore with probability $1 - \mathcal{O}(\delta)$ Assumption 3 is satisfied with $\texttt{Reg}_{\text{Sq}} \leq \mathcal{O}(\log T)$.

Before proceeding further we note that Foster & Rakhlin (2020) invokes Assumption-3 (Assumption 2a in Foster & Rakhlin (2020)) for all sequences. In the proof of Lemma 2 in Foster & Rakhlin (2020), Appendix B, using this assumption, the authors conclude that SqAlg guarantees that with probability 1

$$\sum_{t=1}^{T} \ell_{\text{sq}}(\hat{y}_{t,a_t}, y_{t,a_t}) - \inf_{g \in \mathcal{H}} \sum_{t=1}^{T} \ell_{\text{sq}}(g(\mathbf{x}_{t,a_t}), y_{t,a_t}) \leq \mathcal{O}(\log T)$$

In our analysis this would hold in high probability, i.e., with probability $1 - \mathcal{O}(\delta)$ (this randomness is over the initialization and the perturbation of the network). Subsequently we invoke Freedman's Inequality (Lemma 1 in Foster & Rakhlin (2020)) that holds with probability $(1 - \delta)$ and take a union bound of both the high probability events to conclude that with probability $(1 - (\delta + \mathcal{O}(\delta)))$

$$\sum_{t=1}^{T} \sum_{a \in \mathcal{A}} p_{t,a} \left(\hat{y}_t(x_t, a_t) - f^*(x_t, a_t)\right)^2 \leq 2\texttt{Reg}_{\text{Sq}}(T) + 16\log(\delta^{-1}).$$

Note that the the $1 - \delta$ high probability event is with respect to the randomness of the arm algorithm. Thereafter the analysis follows as in Foster & Rakhlin (2020). Therefore for any sequence of contexts and costs, our regret bound holds in high probability over the randomness of initialization and the perturbation of the network and the randomness of the arm choices.

Invoking Theorem 3.1 we get with with probability $1 - \delta/2$

$$\texttt{Reg}_{\text{CB}}(T) = \mathcal{O}\bigg(\sqrt{KT \log T} + \sqrt{KT \log(16\delta^{-1})} + \frac{K(\log T + \log(16\delta^{-1}))}{\alpha y_l(\Delta_l + \alpha y_l)}\bigg).$$

Taking a union bound over all the high probability events, we have with probability $1 - \mathcal{O}(\delta)$ over all the randomness in the Algorithm the performance constraint in (2) is satisfied and,

$$\texttt{Reg}_{\text{CB}}(T) = \mathcal{O}\bigg(\sqrt{KT}\Big(\sqrt{\texttt{Reg}_{\text{Sq}}(T)} + \sqrt{\log(16\delta^{-1})}\Big) + \frac{K(\texttt{Reg}_{\text{Sq}}(T) + \log(16\delta^{-1}))}{\alpha y_l(\Delta_l + \alpha y_l)}\bigg)$$

$\square$

**Theorem 5.2** (**Regret bound for Neural** `C-FastCB`). *We instantiate Sq-Alg with the predictor* $\hat{y}_{t,a_t} = \tilde{f}^{(S)}\left(\theta; \mathbf{x}_t, \varepsilon^{(1:S)}\right)$ *from* (15) *and update the parameters using OGD in* (17). *Under Assumptions 1,2, 4, 5 and 6 with $\gamma_t$ chosen as in ($\gamma_{\mathtt{i}}$-Schedule), step-size sequence $\{\eta_t\}$, width $m$, perturbation constant $c_p$, and projection ball $B$, with probability $(1 - \mathcal{O}(\delta))$, the performance constraint in (2) is satisfied by C-FastCB and the expected regret is given by*

$$\mathbb{E}\,\mathtt{Reg}_{\mathtt{CB}}(T) \leq \mathcal{O}\left(\sqrt{KL^* \log L^* \log T} + K \log T + \frac{K \log T}{\alpha y_l(\Delta_l + \alpha y_l)}\right).$$

*Proof.* As in the previous Theorem, we set the width of the network $m = \max\left(\mathcal{O}(T^5), \mathcal{O}(\frac{4LT}{\delta})\right)$ and the projection set $B = B_{\rho,\rho_1}^{\mathrm{Frob}}(\theta_0)$, the layer-wise Frobenius ball around the initialization $\theta_0$ with radii $\rho, \rho_1$ which is defined as

$$B_{\rho,\rho_1}^{\mathrm{Frob}}(\theta_0) := \{\theta \in \mathbb{R}^p : \|\operatorname{vec}(W^{(l)}) - \operatorname{vec}(W_0^{(l)})\|_2 \leq \rho, l \in [L], \|\mathbf{v} - \mathbf{v}_0\|_2 \leq \rho_1\}. \tag{42}$$

We set $\rho$ and $\rho_1$ according to Theorem 3.3 in Deb et al. (2024a), and the perturbation constant $c_p = \mathcal{O}(\sqrt{\lambda})$, where $\lambda$ is the Lipschitz parameter of the loss. Now, invoking Theorem 3.3 in Deb et al. (2024a) we get with probability $1 - \mathcal{O}(\delta)$ over the randomness of initialization and $\{\varepsilon\}_{s=1}^S$, the regret of projected OGD with loss $\mathcal{L}_{\mathrm{Sq}}^{(S)}\left(y_t, \{\tilde{f}(\theta; \mathbf{x}_t, \varepsilon_s)\}_{s=1}^S\right)$ for online regression with KL loss is bounded by $\mathcal{O}(\log T)$ i.e.,

$$\sum_{t=1}^T \ell_{\mathrm{KL}}(\hat{y}_{t,a_t}, y_{t,a_t}) - \inf_{g \in \mathcal{H}} \sum_{t=1}^T \ell_{\mathrm{KL}}(g(\mathbf{x}_{a_t}), y_{t,a_t}) \leq \mathcal{O}(\log T)$$

Therefore with probability $1 - \mathcal{O}(\delta)$, Assumption 4 is satisfied with $\mathtt{Reg}_{\mathtt{Sq}} \leq \mathcal{O}(\log T)$.

Before proceeding further we note that Foster & Krishnamurthy (2021) invokes Assumption-3 (Assumption 2 in Foster & Krishnamurthy (2021)) for all sequences. In the proof of Theorem 1 in Foster & Krishnamurthy (2021), using this assumption, the authors conclude that $E[\bar{\mathrm{Reg}}_{KL}(T)] \leq \bar{\mathrm{Reg}}_{KL}(T)$, where $\bar{\mathrm{Reg}}_{KL}(T)$ is the conditional expectation of of the KL regret with respect to $\mathcal{F}_{t-1} = \sigma((\mathbf{x}_{i,a_i}, y_{i,a_i}), i \leq t - 1)$. In our case $\bar{\mathrm{Reg}}_{KL}(T) \leq \mathcal{O}(\log T)$ holds with probability $1 - \mathcal{O}(\delta)$ and we need to provide an expected bound. Now note that $\bar{\mathrm{Reg}}_{KL}(T) \leq T$, for all sequences therefore setting $\mathcal{O}(\delta) = 1/T$ we get that

$$E[\bar{\mathrm{Reg}}_{KL}(T)] \leq \mathcal{O}(\log(T))\left(1 - \frac{1}{T}\right) + 1 = \mathcal{O}(\log T)$$

Thereafter the analysis follows as in Foster & Krishnamurthy (2021). Now invoking Theorem 4.1

$$\mathbb{E}\,\mathtt{Reg}_{\mathtt{CB}}(T) \leq \mathcal{O}\left(\sqrt{KL^* \log L^* \log T} + K \log T + \frac{K \log T}{\alpha y_l(\Delta_l + \alpha y_l)}\right).$$

Taking a union bound over all the high probability events, we have with probability $1 - \mathcal{O}(\delta)$ over over the randomness of initialization and $\{\varepsilon\}_{s=1}^S$ the expected regret is bounded by

$$\mathtt{Reg}_{\mathtt{CB}}(T) = \mathcal{O}\left(\sqrt{KT}\left(\sqrt{\mathtt{Reg}_{\mathtt{Sq}}(T)} + \sqrt{\log(16\delta^{-1})}\right) + \frac{K(\mathtt{Reg}_{\mathtt{Sq}}(T) + \log(16\delta^{-1}))}{\alpha y_l(\Delta_l + \alpha y_l)}\right)$$

while simultaneously with probability $1 - \mathcal{O}(\delta)$ over all the randomness in the Algorithm the performance constraint in (2) is satisfied. $\qquad\square$

# E  UNKNOWN BASELINE COSTS

In this section we relax the the assumption of knowing the true baseline cost values to having a noisy observation for the baseline cost $y_{t,b_t}$. More formally, consider the following filtration

$$\mathcal{F}_{t-1} = \sigma\left((\mathbf{x}_{i,a}, a_i, y_{i,\tilde{a}_i}), \mathbf{x}_{t,a}; 1 \leq i \leq t - 1, a \in [K]\right).$$

Then we assume that

$$\mathbb{E}[y_{t,b_t}|\mathcal{F}_{t-1}] = h(\mathbf{x}_{t,b_t}), \quad \forall t \in [T]$$

We can slightly modify our algorithms to retain the same regret bound and performance constraint guarantees. Consider `C-SquareCB` (Algorithm 1) and replace the *safety condition* in (4) by the following more stringent condition:

$$\hat{y}_{t,\tilde{a}_t} + \sum_{i \in \mathcal{S}_{t-1}} \sum_{a \in [K]} p_{i,a}\hat{y}_{i,a} + 16\sqrt{m_{t-1}\Big(\text{Reg}_{\text{Sq}}(m_{t-1}) + \log(4/\delta)\Big)}$$

$$- \alpha \min\left(\sum_{i \in \mathcal{S}_{t-1} \cup t} y_{i,b_i} - 5\sqrt{(m_{t-1}+1)\ln\left(\frac{16}{\delta}\right)}, m_{t-1}y_l\right)$$

$$\leq \sum_{i \in \mathcal{S}_{t-1} \cup t} y_{i,b_t} - 5\sqrt{(m_{t-1}+1)\ln\left(\frac{16}{\delta}\right)}$$

$$+ \alpha \max\left(\sum_{i \in \mathcal{S}_{t-1}^c} y_{i,b_t} - 5\sqrt{n_{t-1}\ln\left(\frac{16}{\delta}\right)}, n_{t-1}y_l\right). \tag{43}$$

The next theorem shows that our modified algorithm satisfies the same regret bound as in Theorem 5.1 while satisfying the performance constraint.

**Theorem E.1** (**Regret for `C-SquareCB` with unknown baseline cost**). *Suppose Assumptions 1,2 and 3 hold. With probability at least $1-\delta$, `C-SquareCB` (Algorithm 1) with the safety condition (4) replaced by (43) satisfies the performance constraint in (2) and has the following regret bound:*

$$\text{Reg}_{\text{CB}}(T) = \mathcal{O}\left(\sqrt{KT}\left(\sqrt{\text{Reg}_{\text{Sq}}(T)} + \sqrt{\log(8\delta^{-1})}\right) + \frac{K(\text{Reg}_{\text{Sq}}(T) + \log(8\delta^{-1}))}{\alpha^2 y_l^2}\right). \tag{44}$$

*Proof of Theorem E.1.* We first start by showing that the modified safety condition (43) ensures that with high probability the performance constraint in (2) is satisfied.

**Lemma E.2.** *Let Assumptions 1, 2 and 3 hold. Then, for $\delta > 0$ and $\gamma_t = \sqrt{K|\mathcal{S}_t|/(\text{Reg}_{\text{Sq}}(m_T) + \log(8\delta^{-1}))}$, with probability $1-\delta/2$, `C-SquareCB` satisfies the performance constraint in (2).*

*Proof of Lemma E.2.* We start with our safety condition in (4) for the known baseline case and show that it is satisfied with high probability whenever the new safety condition in (43) is satisfied, i.e., the new condition is strictly more conservative than the previous one. Recall that from (4) we have

$$\underbrace{\hat{y}_{t,\tilde{a}_t} + \sum_{i \in \mathcal{S}_{t-1}} \sum_{a \in [K]} p_{i,a}\hat{y}_{i,a}}_{(A)} + \underbrace{\sum_{i \in \mathcal{S}_{t-1}^c} h(\mathbf{x}_{i,b_i})}_{(B)} + \underbrace{16\sqrt{m_{t-1}\Big(\text{Reg}_{\text{Sq}}(m_{t-1}) + \log(4/\delta)\Big)}}_{(C)}$$

$$\leq (1+\alpha)\sum_{i=1}^{t} h(\mathbf{x}_{i,b_t}).$$

Now moving term $(B)$ to the other side we have that the above condition is equivalent to

$$\hat{y}_{t,\tilde{a}_t} + \sum_{i \in \mathcal{S}_{t-1}} \sum_{a \in [K]} p_{i,a}\hat{y}_{i,a} + 16\sqrt{m_{t-1}\Big(\text{Reg}_{\text{Sq}}(m_{t-1}) + \log(4/\delta)\Big)}$$

$$\leq (1+\alpha)\sum_{i=1}^{t} h(\mathbf{x}_{i,b_t}) - \sum_{i \in \mathcal{S}_{t-1}^c} h(\mathbf{x}_{i,b_i})$$

$$= (1+\alpha)\sum_{i \in \mathcal{S}_{t-1} \cup t} h(\mathbf{x}_{i,b_t}) + \alpha \sum_{i \in \mathcal{S}_{t-1}^c} h(\mathbf{x}_{i,b_t}),$$

where we have used the fact that $[t] = \mathcal{S}_{t-1} \cup \mathcal{S}_{t-1}^c \cup t$. We can further write the above condition as:

$$\hat{y}_{t,\tilde{a}_t} + \sum_{i \in \mathcal{S}_{t-1}} \sum_{a \in [K]} p_{i,a} \hat{y}_{i,a} + 16\sqrt{m_{t-1}\Big(\text{Reg}_{\text{Sq}}(m_{t-1}) + \log(4/\delta)\Big)} - \alpha \sum_{i \in \mathcal{S}_{t-1} \cup t} h(\mathbf{x}_{i,b_i})$$
$$\leq \sum_{i \in \mathcal{S}_{t-1} \cup t} h(\mathbf{x}_{i,b_t}) + \alpha \sum_{i \in \mathcal{S}_{t-1}^c} h(\mathbf{x}_{i,b_t}) \tag{45}$$

Now note that $\mathbb{E}[y_{t,b_t}|\mathcal{F}_{t-1}] = h(\mathbf{x}_{t,b_t})$, $\forall t \in [T]$. Therefore $X_t = y_{t,b_t} - h(\mathbf{x}_{t,b_t})$ is a martingale difference sequence and since $X_t \in [-1, 2]$ we use Azuma-Hoeffding to ensure that for any $\epsilon > 0$ and all $T > 0$,

$$P\left(\sum_{t=1}^T |y_{t,b_t} - h(\mathbf{x}_{t,b_t})|\right) \leq 2\exp\left(\frac{-\epsilon^2}{18T}\right).$$

Therefore with probability $1 - \delta/8$

$$\sum_{i \in \mathcal{S}_{t-1} \cup t} |y_{t,b_t} - h(\mathbf{x}_{t,b_t})| \leq 5\sqrt{(m_{t-1} + 1)\ln\left(\frac{16}{\delta}\right)}$$

and, with probability $1 - \delta/8$

$$\sum_{i \in \mathcal{S}_{t-1}^c} |y_{t,b_t} - h(\mathbf{x}_{t,b_t})| \leq 5\sqrt{n_{t-1}\ln\left(\frac{16}{\delta}\right)}.$$

Further $\sum_{i \in \mathcal{S}_{t-1}^c} h(\mathbf{x}_{t,b_t}) \geq n_{t-1}y_l$, and $\sum_{i \in \mathcal{S}_{t-1} \cup t} h(\mathbf{x}_{t,b_t}) \geq (m_{t-1} + 1)y_l$; therefore with probability $1 - \delta/8$ we have the following lower bound for the rhs of (45):

$$\sum_{i \in \mathcal{S}_{t-1}} h(\mathbf{x}_{i,b_t}) + \alpha \sum_{i \in \mathcal{S}_{t-1}^c \cup t} h(\mathbf{x}_{i,b_t}) \geq \sum_{i \in \mathcal{S}_{t-1} \cup t} y_{i,b_t} - 5\sqrt{(m_{t-1} + 1)\ln\left(\frac{16}{\delta}\right)}$$
$$+ \alpha \max\left(\sum_{i \in \mathcal{S}_{t-1}^c} y_{i,b_t} - 5\sqrt{n_{t-1}\ln\left(\frac{16}{\delta}\right)}, n_{t-1}y_l\right)$$

Next, with probability $1 - \delta/8$ we have the following upper bound on the lhs of (45)

$$\hat{y}_{t,\tilde{a}_t} + \sum_{i \in \mathcal{S}_{t-1}} \sum_{a \in [K]} p_{i,a} \hat{y}_{i,a} + 16\sqrt{m_{t-1}\Big(\text{Reg}_{\text{Sq}}(m_{t-1}) + \log(4/\delta)\Big)} - \alpha \sum_{i \in \mathcal{S}_{t-1}^c} h(\mathbf{x}_{i,b_i})$$
$$\leq \hat{y}_{t,\tilde{a}_t} + \sum_{i \in \mathcal{S}_{t-1}} \sum_{a \in [K]} p_{i,a} \hat{y}_{i,a} + 16\sqrt{m_{t-1}\Big(\text{Reg}_{\text{Sq}}(m_{t-1}) + \log(4/\delta)\Big)}$$
$$- \alpha \min\left(\sum_{i \in \mathcal{S}_{t-1} \cup t} y_{i,b_i} - 5\sqrt{(m_{t-1} + 1)\ln\left(\frac{16}{\delta}\right)}, m_{t-1}y_l\right)$$

Therefore if the following condition holds

$$\hat{y}_{t,\tilde{a}_t} + \sum_{i \in \mathcal{S}_{t-1}} \sum_{a \in [K]} p_{i,a} \hat{y}_{i,a} + 16 \sqrt{m_{t-1} \left( \text{Reg}_{\text{Sq}}(m_{t-1}) + \log(4/\delta) \right)}$$

$$- \alpha \min \left( \sum_{i \in \mathcal{S}_{t-1} \cup t} y_{i,b_i} - 5 \sqrt{(m_{t-1}+1) \ln \left( \frac{16}{\delta} \right)}, m_{t-1} y_l \right)$$

$$\leq \sum_{i \in \mathcal{S}_{t-1} \cup t} y_{i,b_t} - 5 \sqrt{(m_{t-1}+1) \ln \left( \frac{16}{\delta} \right)}$$

$$+ \alpha \max \left( \sum_{i \in \mathcal{S}_{t-1}^c} y_{i,b_t} - 5 \sqrt{n_{t-1} \ln \left( \frac{16}{\delta} \right)}, n_{t-1} y_l \right)$$

then with probability $1 - \delta/4$

$$\hat{y}_{t,\tilde{a}_t} + \sum_{i \in \mathcal{S}_{t-1}} \sum_{a \in [K]} p_{i,a} \hat{y}_{i,a} + \sum_{i \in \mathcal{S}_{t-1}^c} h(\mathbf{x}_{i,b_i}) + 16 \sqrt{m_{t-1} \left( \text{Reg}_{\text{Sq}}(m_{t-1}) + \log(4/\delta) \right)}$$

$$\leq (1+\alpha) \sum_{i=1}^{t} h(\mathbf{x}_{i,b_t}).$$

Now we invoke Lemma 4 with $\delta$ substituted by $\delta/4$ and take a union bound with the above high probability even to conclude that with probability $1 - \delta/2$ C-SquareCB (Algorithm 1) with the safety condition (43) satisfies the performance constraint in (2). $\square$

Next we show that the regret of the modified C-SquareCB algorithm satisfies the same regret bound. Note that the regret decomposition in (6) and the bound on term $I$ in (10) still hold. Therefore our objective to complete the proof of the Theorem is to bound $n_T$ as in Step-2 of the proof of Theorem 5.1. The following Lemma bounds $n_T$, the number of times the baseline action is played with the modified safety condition in (43).

**Lemma E.3.** *Suppose Assumption 1,2 and 3 hold. Then, with probability $1 - \delta/4$ the number of times the baseline action is played by* C-SquareCB *with the safety condition (43) is bounded as follows:*

$$n_T \leq \mathcal{O} \left( \frac{K(\text{Reg}_{\text{Sq}}(T) + \log(8\delta^{-1}))}{\alpha y_l (\Delta_l + \alpha y_l)} \right). \tag{46}$$

*Proof.* Let $\tau$ be the last round at which the algorithm plays the conservative action, i.e.,

$$\tau = \max\{1 \leq t \leq T | a_t = b_t\}.$$

Recall that $m_t = |\mathcal{S}_t|$ and $n_t = |\mathcal{S}_t^c|$. By the definition of $\tau$, we have that at round $\tau$

$$\hat{y}_{t,\tilde{a}_t} + \sum_{i \in \mathcal{S}_{t-1}} \sum_{a \in [K]} p_{i,a} \hat{y}_{i,a} + 16 \sqrt{m_{t-1} \left( \text{Reg}_{\text{Sq}}(m_{t-1}) + \log(4/\delta) \right)}$$

$$- \alpha \min \left( \sum_{i \in \mathcal{S}_{t-1} \cup t} y_{i,b_i} - 5 \sqrt{(m_{t-1}+1) \ln \left( \frac{16}{\delta} \right)}, m_{t-1} y_l \right)$$

$$> \sum_{i \in \mathcal{S}_{t-1} \cup t} y_{i,b_t} - 5 \sqrt{(m_{t-1}+1) \ln \left( \frac{16}{\delta} \right)}$$

$$+ \alpha \max \left( \sum_{i \in \mathcal{S}_{t-1}^c} y_{i,b_t} - 5 \sqrt{n_{t-1} \ln \left( \frac{16}{\delta} \right)}, n_{t-1} y_l \right)$$

Re-arranging we can write the above condition as follows:

$$\alpha n_{\tau-1} y_l \le \hat{y}_{t,\tilde{a}_\tau} + \sum_{i \in \mathcal{S}_{\tau-1}} \sum_{a \in [K]} p_{i,a} \hat{y}_{i,a} + 5 \sqrt{(m_{t-1}+1) \ln \left( \frac{16}{\delta} \right)}$$

$$+ 16 \sqrt{m_{\tau-1} \left( \text{Reg}_{\text{Sq}}(m_{\tau-1}) + \log(4/\delta) \right)} - \alpha (m_{t-1}+1) y_l$$

Hereafter following the analysis as in the proof of Lemma 3.2 we can show that with probability $1 - \delta/2$

$$\alpha n_{\tau-1} y_l \le -(m_{\tau-1}+1)\alpha y_l + 64 \sqrt{K(m_{\tau-1}+1)} \left( \text{Reg}_{\text{Sq}}(T) + \sqrt{\log(16\delta^{-1})} \right)$$

$$= -(m_{\tau-1}+1)\alpha y_l + 64 \sqrt{K} \sqrt{(m_{\tau-1}+1)} \left( \text{Reg}_{\text{Sq}}(T) + \sqrt{\log(16\delta^{-1})} \right).$$

Now using the analysis as in the proof of Lemma 3.3 with $m = m_{\tau-1}+1$, $c_1 = \alpha y_l$, $c_2 = 64\sqrt{K} \left( \text{Reg}_{\text{Sq}}(T) + \sqrt{\log(16\delta^{-1})} \right)$, with probability $1 - \delta/4$ we can bound bound $n_T$ as follows:

$$n_T \le \mathcal{O} \left( \frac{K(\text{Reg}_{\text{Sq}}(T) + \log(2\delta^{-1}))}{\alpha^2 y_l^2} \right).$$

$\square$

To complete the proof of Theorem E.1 we combine Lemma E.2 and Lemma E.3 with Lemma 3.4. $\square$

Next consider `C-FastCB` (Algorithm 2) and replace the *safety condition* by the following more stringent condition:

$$\hat{y}_{t,\tilde{a}_t} + \sum_{i \in \mathcal{S}_{t-1}} \sum_{a \in [K]} p_{i,a} \hat{y}_{i,a} + 16 \sqrt{m_{t-1} \text{Reg}_{\text{KL}}(T)}$$

$$- \alpha \min \left( \sum_{i \in \mathcal{S}_{t-1} \cup t} y_{i,b_i} - 5 \sqrt{(m_{t-1}+1) \ln \left( \frac{16}{\delta} \right)}, m_{t-1} y_l \right)$$

$$\le \sum_{i \in \mathcal{S}_{t-1} \cup t} y_{i,b_t} - 5 \sqrt{(m_{t-1}+1) \ln \left( \frac{16}{\delta} \right)}$$

$$+ \alpha \max \left( \sum_{i \in \mathcal{S}_{t-1}^c} y_{i,b_t} - 5 \sqrt{n_{t-1} \ln \left( \frac{16}{\delta} \right)}, n_{t-1} y_l \right). \tag{47}$$

Then we have the following regret bound for `C-FastCB`.

**Theorem E.4** (**Regret Bound for C-FastCB** for unknown baseline). *Let Assumptions 1, 2 and 4 hold. With probability $1 - \delta$,* `C-FastCB` *(Algorithm 2) with $\gamma_i$ chosen in ($\gamma_\text{i}$-Schedule), and with the modified safety condition in (47) satisfies the performance constraint in (2) and has the following bound on the expected regret (expectation is for the action distributions):*

$$\mathbb{E}\left[\text{Reg}_{\text{CB}}(T)\right] = \mathcal{O}\left(\sqrt{KL^* \log(L^*)\,\text{Reg}_{\text{KL}}(T)} + \frac{K\text{Reg}_{\text{KL}}(T)}{\alpha^2 y_l^2} \log\left(\frac{e\sqrt{K\text{Reg}_{\text{KL}}(T)}}{\alpha^2 y_l^2}\right)\right). \quad (48)$$

*Proof.* The proof follows along the same lines as in proof of Theorem E.1. By upper bounding and lower bounding the safety condition we can show that when (47) is satisfied then with high probability the safety condition in Algorithm 2 is satisfied.

Further the additional terms in (47) can be handled exactly as in proof of Lemma E.3 and combining with the proof of Lemma C.2 we complete the proof. □

## F  ADDITIONAL EXPERIMENTAL DETAILS

### F.1  EXPLORATION PARAMETER $\gamma_i$

Since the optimal loss $L_i^*$ is not known in advance, the exploration parameter $\gamma_i$ is treated as a hyper-parameter in our experiments. A heuristic choice is to substitute $\sum_{i=1}^{t} L_i^*$ by the sum of the observed losses until time $t - 1$, i.e., $\sum_{i=1}^{t-1} L_i$ to choose $\gamma_t$. Note that at time $t$, $\sum_{i=1}^{t-1} L_i$ is known to the user. The other choice is to treat $\gamma_i$ as a single parameter $\gamma$ and tune it for different values. In our experiments we tune $\gamma$ in $\{10, 20, 50, 100, 200, 500, 1000\}$. We plot the corresponding cumulative regret for all these choices in Figure 3 and we note that the heuristic choice of $\sum_{i=1}^{t-1} L_i$ produces good results in the majority of environments.

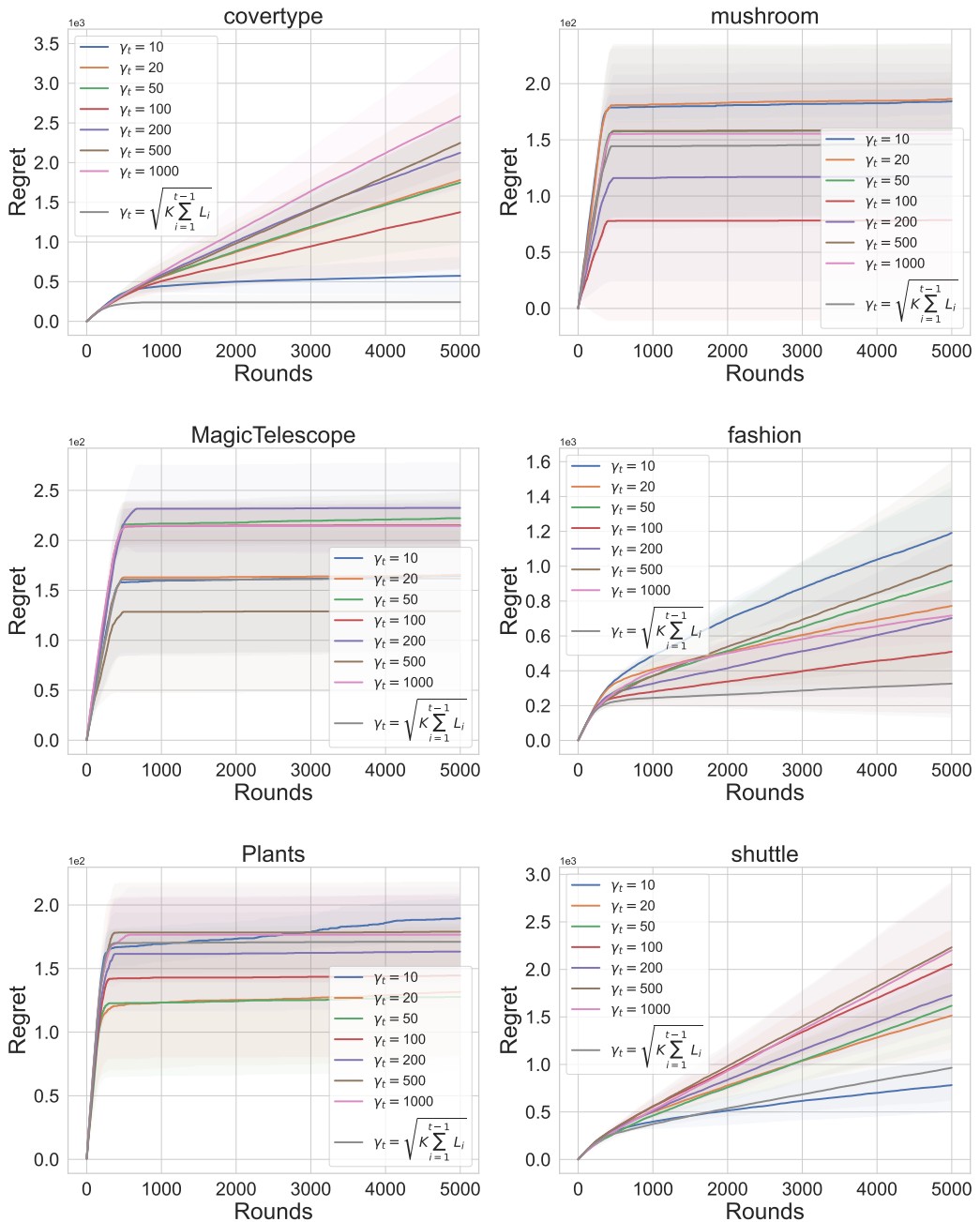

Figure 3: Comparison of cumulative regret of `C-FastCB` with various choices of the exploration parameter $\gamma_t$ on openml datasets (averaged over 5 runs).

## F.2 DEPENDENCE ON THE NETWORK WIDTH

For Theorem 5.1 and 5.2 we use one specific instance of an online regression oracle, namely Online Gradient Descent with overparameterized neural networks. Note that the width requirements in the theorem statements are sufficient conditions, but not necessary. Therefore to address concerns about practicality of the algorithms, we provide additional evidence here that shows the performance of the algorithms for different choices of the width of the network. Figure 4 and Figure 5 shows that for practical purposes, fixed width networks suffice for both Algorithm 1 and 2.

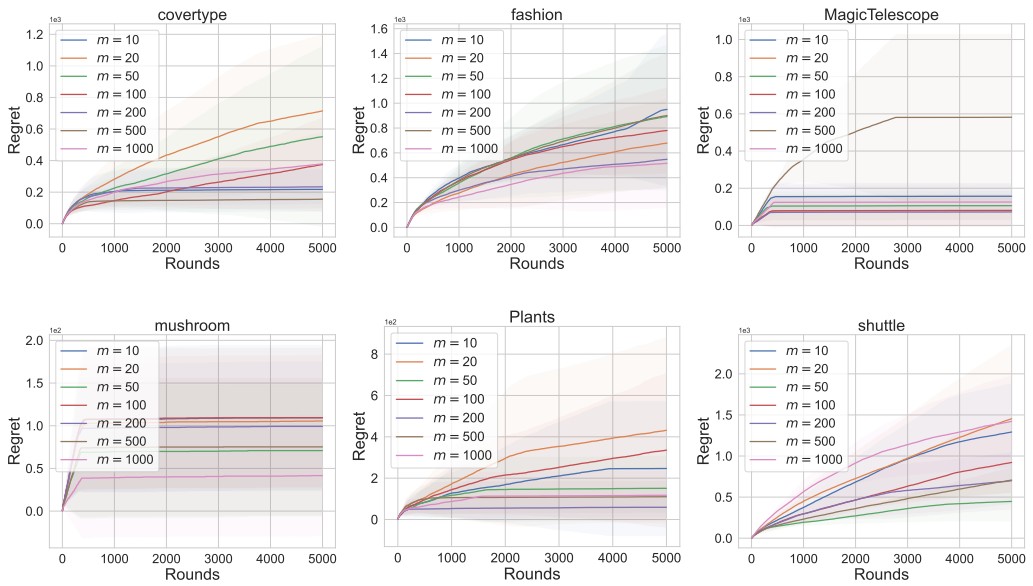

Figure 4: Comparison of cumulative regret of `C-SquareCB` with various choices of the network width $m$ on openml datasets (averaged over 5 runs).

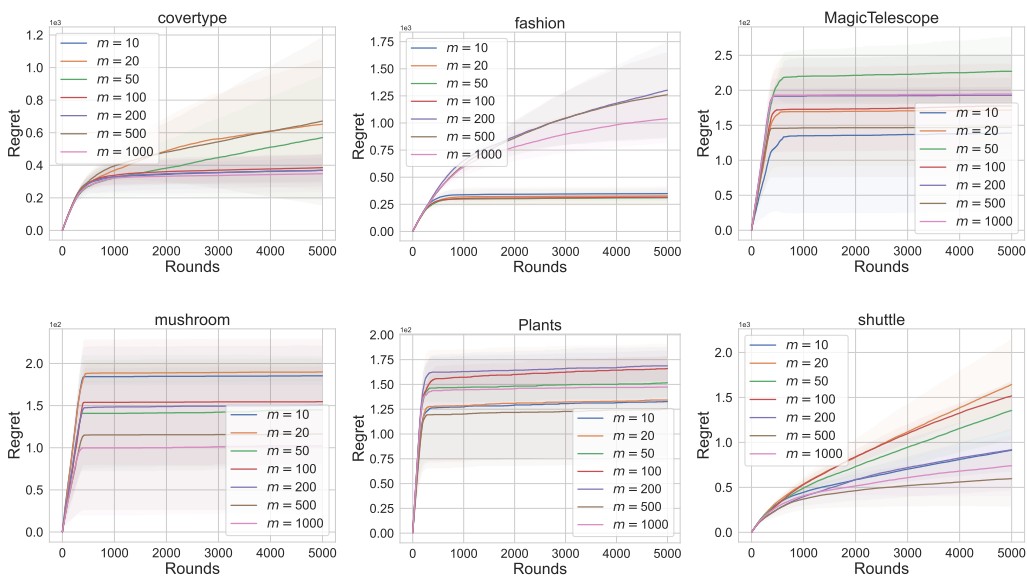

Figure 5: Comparison of cumulative regret of `C-FastCB` with various choices of the network width $m$ on openml datasets (averaged over 5 runs).

