# OpenReview forum: "Conservative Contextual Bandits: Beyond Linear Representations"
_ICLR.cc/2025/Conference — ICLR 2025 Poster_

### Official Review · Reviewer_wAuW · 2024-10-31

**Soundness:** 3
**Presentation:** 2
**Contribution:** 2
**Rating:** 6
**Confidence:** 3

**Summary:**

This paper proposes algorithms for the conservative contextual bandit problem under a non-linear reward model.  The proposed algorithms apply IGW exploration based on an online regression oracle, enabling the selection of either an exploratory or a conservative policy based on a safety condition.  The proposed algorithms achieve sub-linear regret with respect to the total time step $T$, and its performance is supported by numerical experiments.

**Strengths:**

- The proposed algorithm introduces the first conservative contextual bandit algorithm for a general reward model by adapting the assumption of access to a regression oracle and leveraging the IGW algorithm, which previously applied only to linear reward models. The authors also present regret analysis for the algorithm. Although I was unable to rigorously verify all proofs, the results appear consistent, achieving a regret bound comparable to that of the linear case.
- The methodology is illustrated with examples using an online gradient descent regression oracle and feed-forward neural networks. Real-world data experiments further support the algorithm’s performance.

**Weaknesses:**

1. This paper introduces multiple algorithms and theoretical results for the conservative contextual bandit problem with a general non-linear cost function. Consequently, much of the main content is focused on the operation of the algorithms, assumptions required for the theoretical results, and descriptions of the outcomes, with limited discussion of the technical challenges arising from handling non-linear cost functions in CCB and how these challenges were addressed. It seems that Algorithm 1 and Algorithm 2 are almost identical aside from the use of different regression oracles. Moreover, if, as the authors state, Algorithm 2 has a tighter regret guarantee than Algorithm 1, it may be beneficial to focus more on Algorithm 2, detailing the challenges and technical novelties associated with it.

2. The main theorem statements lack completeness. For example, in the statement of Theorem 3.1, there is no indication of how to set the input parameter $\gamma$ for the algorithm to achieve the stated regret bound. Similarly, Theorem 5.1 could benefit from more precise language rather than vague expressions like “appropriate choice of parameter”. Additionally, based on my understanding, the exploration parameter $\gamma_i$ in Algorithm 2 appears to depend on $\eta_i$ (line 1218). However, $\eta_i$ in turn depends on $L_i^*$, which is the true cost value of the optimal action and unknown to the agent. Further clarification is needed on how $\gamma_i$ is determined.

3. The description for experiment reproducibility is lacking. Additional information, such as how the hyperparameters for each algorithm are set, would be helpful.

**Questions:**

1. In Algorithm 1, if the safety condition is not met, the baseline policy ($b_t$) is used, and its noiseless cost $h(x_{t,b_t})$ is observed. However, this data is not used in the oracle. Since the baseline policy provides high-quality data with zero noise, why is it not utilized? Would using this data yield a better estimator? Additionally, in the linear cost case (Kazerouni et al., 2017), a noisy reward is observed for actions chosen by the baseline policy (e.g., $y_{t,b_t}$). What would happen if, in this paper’s setting, the algorithm observed $y_{t,b_t}$ instead of $h(x_{t,b_t})$?

2. The algorithm introduced in Section 4 is referred to as having a “first-order regret bound.” Why is it called “first-order”? How would second-order and third-order bounds be defined?

3. When defining the Neural Tangent Kernel (NTK) matrix, is $x_t$ the feature chosen by the algorithm at time $t$? If so, it seems unclear how the NTK can be defined given that the chosen features are unknown prior to starting the algorithm. It may be better to define the NTK matrix based on the context of all actions at all times, as in Zhou et al. (2020). Additionally, the regret bounds in Theorems 5.1 and 5.2 are independent on $\lambda_0$. Why is this assumption necessary, and does it not affect the regret bound?

4. Most literature on Neural Tangent Kernel-based neural bandits shows a dependency of regret on the effective dimension of the NTK. In contrast, the proposed paper presents a dependency on $K$ instead of the effective dimension. Could you explain what enables this distinction?

5. In the Neural Conservative Bandit, the activation function in the first-order bound changed from a smooth & Lipschitz function $\phi$ to a sigmoid function $\sigma$. What are the differences between $\phi$ and $\sigma$?

6. There do not appear to be terms related to the perturbation constant or the number of ensemble $S$ in the regret bound. Do these quantities not affect the regret? Additionally, how are these hyperparameters chosen in the experiments, and how do they impact performance?

---

**[Typo]**

- line 295: “Line 7” should be “Line 8.”

**Details Of Ethics Concerns:**

There are no specific ethics concerns.

---

> ### Author Response · Authors · 2024-11-22
> **Response to questions by Reviewer wAuW**
>
> ---
> ---
> ## **_Details of Technical Contribution_**
> ___
> We summarize our technical and algorithmic contributions below. We have also added two remarks in the updated draft detailing these. \
> \
> **1. Reduction using Squared loss (C-SquareCB):**
>   - The analysis in [Foster & Rakhlin (2020)](https://proceedings.mlr.press/v119/foster20a) does not have a safety condition and therefore our analysis bounding $n_T$ (the number of times the baseline action is played in Step 2) and the performance constraint satisfaction (in Step 4) of proof of Theorem 3.1 are original contributions.
>   - Note that unlike in the linear case [Kazerouni et al. (2017)](https://arxiv.org/abs/1611.06426), we cannot maintain high probability confidence bounds around parameter estimates in the general case. Therefore in our analysis we crucially relate $n_T$ to squared loss and thereafter give a reduction to online regression. We emphasize that such a reduction does not exist in the literature. Further our analysis in Step 4 leads to the construction of the safety condition in (4), that maintains the performance guarantee in high probability.
>   - Finally even the analysis from [Foster & Rakhlin (2020)](https://proceedings.mlr.press/v119/foster20a) cannot be directly used to bound the regret for the time steps when the IGW actions were picked (term $I$ in eq (5) of the draft). This is because we need to carefully choose a time dependent exploration parameter $\gamma_t$, to simultaneously ensure that term $I$ is $\mathcal{O}({\sqrt{T}})$ while ensuring that $n_T$ is small. Therefore in the process, we extend the analysis in [Foster & Rakhlin (2020)](https://proceedings.mlr.press/v119/foster20a) to time-dependent $\gamma_t$ and bound the regret in term $I$.
>
>
> **2. Reduction using KL loss (C-FastCB)**
>   - For the first-order regret bound in Theorem 4.1, we face similar challenges as above. Note that a UCB based analysis for conservative bandits ensuring a first-order regret bound does not exist even in the linear case. Therefore, in our analysis of Theorem 4.1 we show a reduction of $n_T$ to online regression with KL loss and ensures the performance constraint is satisfied by modifying the safety condition.
>   - Note that in [Foster & Krishnamurthy (2021)](https://proceedings.neurips.cc/paper_files/paper/2021/hash/9d684c589d67031a627ad33d59db65e5-Abstract.html) , the exploration parameter $\gamma_t$ is set to a fixed value $\gamma$ but as in the previous case, in our analysis we need a time dependent $\gamma_t$ to ensure that we can bound the regret contributed by both the IGW and baseline actions. *However, unlike the previous case, we crucially need to set $\gamma_t$ in an episodic manner to ensure that the final regret does not have a $\sqrt{T}$ dependence.* We construct $\log (L^*)$ such episodes, keep $\gamma_t$ constant within an episode and modify our analysis in Step 2 to bound $n_T$. Further we extend the analysis in [Foster & Rakhlin (2020)](https://proceedings.mlr.press/v119/foster20a)  with such an episodic schedule of $\gamma_t$ to bound term $I$ and show that it has only an additional $\sqrt{\log(L^*)}$ factor.
>
> **3. Regret Bounds with Neural Networks**
> - Finally to give complete regret bounds, we instantiate the online regression oracle with Online Gradient Descent (OGD) and the function approximator with a feed-forward neural network to give an end-to-end regret bound of $O\big(\sqrt{KT \log (T)} + K \log(T)/\alpha\big)$ for Algorithm 1 (Theorem 5.1) and $O\big(\sqrt{KL^* \log (L^*) \log(T)} + K\log T + K \log(T)/\alpha\big)$ for Algorithm 2 (Theorem 5.2). We also compare our proposed algorithms with existing baselines for conservative bandits and show that our algorithms consistently perform better on real world datasets.
> ___

---

> > ### Author Response · Authors · 2024-11-22
> >
> > ___
> > ___
> > ## **_Choice of exploration parameter $\gamma_i$:_**
> > ___
> > - We have specified the choice of $\gamma_i$ in the statement of Theorem 3.1 and for 4.1 we have specified the choice with a forward reference to ($\gamma_i$ schedule) in Appendix C owing to space constraints.
> >
> > - Indeed in Theorem 4.1, the exploration parameter $\gamma_i$ depends on $L^*$. Note that the dependence of $\gamma_i$ on $L_i^*$ is to provably guarantee a sub-linear regret. We first emphasize that even in the unconstrained case, [Foster & Krishnamurthy (2021)](https://proceedings.neurips.cc/paper_files/paper/2021/hash/9d684c589d67031a627ad33d59db65e5-Abstract.html)  also set $\gamma$ using $L^*$ to give a provable first order regret bound. Further, even in the full information setting, with exponential weights algorithm [Freund & Schapire (1997)](https://www.sciencedirect.com/science/article/pii/S002200009791504X)  one needs to know $L^*$ to set the exploration parameter to get a *first-order bound*. The choice of $\gamma_i$ is in principle similar to optimal step-size schedule choices that depend on problem parameters in optimization results (e.g., strong convexity constant, lipschitz constant, etc.). However, in practice one does not need to use $L_i^*$ to obtain good performance and can work with heuristic choices (see our next response).
> > ___
> > ___
> > ## **_Hyper-parameter:_**
> > ___
> > **Choice of $\gamma_i$:** Since the optimal loss $L_i^*$ is not known in advance, the exploration parameter $\gamma_i$ is treated as a hyper-parameter in our experiments. A heuristic choice is to substitute $\sum_{i=1}^t L_i^*$ by the sum of the observed losses until time $t-1$, i.e., $\sum_{i=1}^{t-1}L_i$ to choose $\gamma_t$. Note that at time $t$, $\sum_{i=1}^{t-1}L_i$ is known to the user. The other choice is to treat $\gamma_i$ as a single parameter $\gamma$ and tune it for different values. In our experiments, we tune $\gamma$ in {10,20,50,100,200,500,1000}. We have plotted the corresponding cumulative regret for all these choices in Figure 3 (Appendix F) and we note that the heuristic choice of $\sum_{i=1}^{t-1}L_i$ produces good results in the majority of the environments.
> > ___

---

> ### Author Response · Authors · 2024-11-22
> **Response to Question by Reviewer wAuW**
>
> ___
> ___
> ## **_Unknown Baseline:_**
> ___
> - **Using the baseline cost $h(\mathbf{x}_{t,b_t})$ in the oracle:** We first note that our analysis separates the time steps into $S_{T}$ and $S_{T}^c$, denoting the subsets containing the time-steps until round $T$ when the IGW and baseline actions were played, respectively, and the oracle is used only for $t \in S_T$. Therefore, not using the baseline cost $h(\mathbf{x_{t,b_t}})$ for $t \in S_T^c$ does not impact our regret guarantees in Theorems 3.1 and 4.1, and we retain the same bounds even if the oracle utilizes the baseline costs. For empirical evaluation one can utilize the baseline costs as well. We compare the performance when the baseline costs are used by the algorithm, report the results in Figure 6 in Appendix F.3 and note that there is no significant performance improvement. This could possibly be because our analysis shows that the number of times the baseline action is played is $O(\log T)$, which is small.
>
> - **Extension to Unknown Baseline costs:** If one does not know the baseline cost $h(\mathbf{x_{t,b_t}})$ and instead observes a noisy version $y_{t,b_t}$, then we can still guarantee a similar regret bound for $\mathtt{C}-\mathtt{SquareCB}$ by modifying the safety condition. We briefly describe the safety condition below and provide a complete proof of the regret bound and performance constraint in Appendix E along with a remark and forward reference in the main paper.
> \
>    Consider the original safety condition in Eq. (4) of the paper:
> $$
> \quad \hat{y_{t,\tilde{a_t}}} + \sum_{i \in S_{t-1}^c} h(\mathbf{x_{i,b_i}}) + 16 \sqrt{m_{t-1}(Reg_{Sq}(m_{t-1}) + \log (4/\delta))} \leq (1+\alpha) \sum_{i=1}^{t} h(\mathbf{x_{i,b_i}})
> $$
> We show in Appendix E that we can re-write this as
> $$
> \hat{y_{t,\tilde{a_t}}} + \sum_{i \in S_{t-1}^c} h(\mathbf{x_{i,b_i}}) + 16 \sqrt{m_{t-1}(Reg_{Sq}(m_{t-1}) + \log (4/\delta))} - \alpha \sum_{i \in S_{t-1} \cup t} h(\mathbf{x_{i,b_i}}) \leq \sum_{i \in S_{t-1} \cup t} h(\mathbf{x_{i,b_i}}) + \alpha \sum_{i \in S_{t-1}^c \cup t} h(\mathbf{x_{i,b_i}})
> $$
> Further with high probability we upper bound the $rhs$ by
> $$
> rhs \geq \sum_{i \in S_{t-1} \cup t} y_{i,b_i} - 5 \sqrt{(m_{t-1} + 1)\ln(16/\delta)} + \alpha \max(\sum_{i \in S_{t-1}^c \cup t} y_{i,b_i}  - 5 \sqrt{n_{t-1}\ln(16/\delta)} , n_{t-1}y_{l})
> $$
> and lower bound the $lhs$ by
> $$
> lhs \leq \hat{y_{t,\tilde{a_t}}} + \sum_{i \in S_{t-1}^c} h(\mathbf{x_{i,b_i}}) + 16 \sqrt{m_{t-1}(Reg_{Sq}(m_{t-1}) + \log (4/\delta))} - \alpha \min(\sum_{i \in S_{t-1} \cup t} y_{i,b_i} - 5 \sqrt{(m_{t-1} + 1)\ln(16/\delta)},m_{t-1}y_{l})
> $$
> Therefore, if the following condition holds:
> $$
> \hat{y_{t,\tilde{a_t}}} + \sum_{i \in S_{t-1}^c} h(\mathbf{x_{i,b_i}}) + 16 \sqrt{m_{t-1}(Reg_{Sq}(m_{t-1}) + \log (4/\delta))} - \alpha \min(\sum_{i \in S_{t-1} \cup t} y_{i,b_i} - 5 \sqrt{(m_{t-1} + 1)\ln(16/\delta)},m_{t-1}y_{l})
> $$
> $$
> \qquad\qquad \qquad \leq \sum_{i \in S_{t-1} \cup t} y_{i,b_i} - 5 \sqrt{(m_{t-1} + 1)\ln(16/\delta)} + \alpha \max(\sum_{i \in S_{t-1}^c \cup t} y_{i,b_i}  - 5 \sqrt{n_{t-1}\ln(16/\delta)} , n_{t-1}y_{l})
> \tag{1}
> $$
> then with high probability our original safety condition holds, i.e.,
> $$
> \hat{y_{t,\tilde{a_t}}} + \sum_{i \in S_{t-1}^c} h(\mathbf{x_{i,b_i}}) + 16 \sqrt{m_{t-1}(Reg_{Sq}(m_{t-1}) + \log (4/\delta))} \leq (1+\alpha) \sum_{i=1}^{t} h(\mathbf{x_{i,b_i}})
> $$
> With this modified safety condition (1), in Appendix~E, we show that we can maintain the same regret bound. A similar analysis holds for the first order regret bound for Algorithm 2.

---

> > ### Author Response · Authors · 2024-11-22
> > **Response to Question by Reviewer wAuW**
> >
> > ___
> > ___
> > ## **_First and Second Order Regret Bound:_**
> > ___
> > - In *first-order regret bounds,* the regret scales as in $L_* = \sum_{t=1}^{T} \ell_{t,a_t^*}$, the cumulative loss of the optimal policy. It has a rich history, with [Freund & Schapire (1997)](https://www.sciencedirect.com/science/article/pii/S002200009791504X) proving the first such bound for the full information setting using Exponential Weights algorithm. Subsequent developments were made in the bandit literature in  [Agarwal et al. (2016)](https://www.semanticscholar.org/paper/A-Multiworld-Testing-Decision-Service-Agarwal-Bird/e71f6ce292c307b95d4845bfcc542b9a08b3baa0), [Agarwal et al. (2017)](https://proceedings.mlr.press/v65/agarwal17a.html), [Allen-Zhu et al. (2018)](https://proceedings.mlr.press/v80/allen-zhu18b.html) and [Foster & Krishnamurthy (2021)](https://proceedings.neurips.cc/paper_files/paper/2021/hash/9d684c589d67031a627ad33d59db65e5-Abstract.html). Separately, [Cesa-Bianchi et al. (2006)](https://arxiv.org/abs/math/0602629) first posed the question of whether further improvements could be achieved by deriving *second-order* (variance-like) bounds on the regret for the full information setting. They provided two choices for second order bounds, one that depends on $\sum_{t=1}^T \ell_{t,a_t^*}^2$ (variance across time) and another that depends on $\sum_{k \leq K} p_{k,t} (\hat{\ell_t} - \ell_{k,t})^2$ (variance across actions), where $\hat{\ell_t} = \sum_{k=1}^K p_{t,k} \ell_{t,k}$, and $p_{k,t}$ is the probability with which expert $k$ is chosen in round $t$.
> >
> > - We have added a new paragraph describing these developments in more detail in the Related Works Section (see Appendix A) along with a forward reference to it in Section 4.
> > ___
> > ___
> > ## **_NTK and $\lambda_0$:_**
> > ___
> > - **NTK Definition:** Note that the NTK matrix is only defined for the analysis and the algorithm does not need access to it and therefore even with a dependence on the selected action, it is well defined. In fact the NTK is defined for a specific trajectory/sequence of $\mathbf{x}_t$'s where $\mathbf{x}_t$ depends on the choice of arms played, and our assumption on the NTK matrix is for all trajectories, which is equivalent to the assumption for the $(TK \times TK)$ NTK matrix as in [Zhou et al. (2020)](https://proceedings.mlr.press/v119/zhou20a); [Zhang et al. (2021)](https://openreview.net/forum?id=tkAtoZkcUnm).
> >
> > - **Use of $\lambda_0:$** The regret bound in Theorems 5.1 and 5.2 do not depend on $\lambda_0,$ but rather the choice of the width of the network $m$ depends on $\lambda_0$ (see the proof of Theorem 3.2 in  [Deb et al. (2024)](https://openreview.net/forum?id=5ep85sakT3)). This is similar to the standard width requirements in Neural UCB ([Zhou et al. (2020)](https://proceedings.mlr.press/v119/zhou20a)) and Neural Thompson Sampling ([Zhang et al. (2021)](https://openreview.net/forum?id=tkAtoZkcUnm)). In our case, the assumption is necessary to ensure a $\log T$ regret bound for the online regression regret.
> >
> > We have specified both these points in the updated draft.
> > ___
> > ___
> > ## **_Effective dimension:_**
> > ___
> > Indeed the regret bounds in  Neural UCB ([Zhou et al. (2020)](https://proceedings.mlr.press/v119/zhou20a)) and Neural Thompson Sampling ([Zhang et al. (2021)](https://openreview.net/forum?id=tkAtoZkcUnm)) depend on the effective dimension $\tilde{d}$. In contrast, our regret bounds do not depend on $\tilde{d}$ specifically because the regret of online gradient descent is independent of $\tilde{d}$. Further, [Deb et al. (2024)](https://openreview.net/forum?id=5ep85sakT3) recently showed that the regret bounds in Neural UCB and Neural Thompson Sampling with the $\tilde{d}$ can be made $\Omega(T\sqrt{K})$ by an oblivious adversary (see Appendix A in [Deb et al. (2024)](https://openreview.net/forum?id=5ep85sakT3)). This is precisely why we do not extend the linear UCB algorithm from [Kazerouni et al. (2017)](https://arxiv.org/abs/1611.06426) to the neural setup.
> >
> > We had specified this in the draft (see Lines 061 to 067 in the draft).
> > ___
> > ___
> > ## **_Activation Function:_**
> > $\phi$ is the general notation for any (point-wise) smooth and Lipschitz activation function, while $\sigma$ is specifically the sigmoid activation function.
> > ___
> > ___
> > ## **_Perturbation Constant:_**
> > ___
> > The perturbation constant is chosen in the analysis and is subsumed in the final regret bound. We have specified the choice in the proof of Theorems 4.1 and 4.2 in Appendix F. [Deb et al. (2024)](https://openreview.net/forum?id=5ep85sakT3) tuned for different choices of the perturbation constant (see Appendix F in [Deb et al. (2024)](https://openreview.net/forum?id=5ep85sakT3)) and show that the unperturbed version perform almost as well as the perturbed ones empirically, and are computationally more efficient. We saw a similar behavior in our experiments and report the final plots for the unperturbed networks. We have specified this in Section 6 of the updated draft.

---

> > > ### Author Response · Authors · 2024-11-24
> > >
> > > Dear reviewer, we again thank you for taking the time to review our paper. Since the end of the discussion period is approaching, we request you to kindly check our responses to your questions. We would be happy to provide any further clarifications you need.
> > >
> > > We have also provided a brief response higlighting the significance of this work to the broader community [(Link)](https://openreview.net/forum?id=SThJXvucjQ&noteId=i9bMk8ga90) and would greatly appreciate you reading it.

---

> > > > ### Author Response · Authors · 2024-11-25
> > > >
> > > > Dear Reviewer, with only one day left until the end of the discussion phase, we request you to kindly check our responses to your questions.
> > > >
> > > > We have worked very hard on the paper and subsequently on the responses to meticulously address every question you raised. We would be glad to address any further clarification questions you might have to positively impact your impression of our work.

---

> ### Comment · Reviewer_wAuW · 2024-11-26
>
> I sincerely apologize for the delayed response and deeply appreciate the authors for their kind and thorough answers despite the numerous questions. I have reviewed all the responses, which greatly helped me in understanding the paper, and accordingly, I have updated my rating to '6'.
>
> However, I would like to leave a few additional comments.
>
> First, I still find the issue related to the $\gamma$'s dependence on $L^*$ problematic. While I understand, as the authors pointed out, that even in previous first-order bounds, the parameters required for theoretical guarantees were dependent on $L^*$, and that this technique was carried over to this paper, inheriting the same $L^*$ dependence. However, I do not believe this entirely justifies the weakness related to $L^*$ dependence. The main contribution of this paper lies in proposing an algorithm with regret guarantees for general function classes. Nevertheless, the fact that partial information about the optimal arm is needed to ensure the regret guarantee remains a limitation in my view. If, as the authors suggested, the regret guarantee can be achieved with $\sum_i L_i$, that would be a significant improvement. (Although I understand that addressing this point is not the main contribution of this paper.)
>
> Additionally, regarding the effective dimension, as I understand from the authors' response, the reason the effective dimension can be replaced with $K$ is due to the use of online gradient descent instead of (stochastic) gradient descent. Is this correct?

---

> > ### Author Response · Authors · 2024-12-03
> >
> > Dear reviewer, we sincerely thank you for revieweing our responses, and your comments. We were conducting an in-depth review of the literature and thinking deeply to respond to your points.
> >
> > Regarding unknown $L^*$, we have thoroughly reviewed the literature, and to the best of our knowledge, for the bandit information case, even in the multi-armed case, existing algorithms: EXP3LIGHT [[Stoltz (2005)]](https://theses.hal.science/tel-00009759), GREEN [[Allenberg et al. (2006)]](https://www.szit.bme.hu/~oti/publications/hannan.pdf)
> > need $L^*$ to prove a first order regret bound. We will add this discussion to the final draft. However, as you said, this is not the point of focus for this work, and we think that this problem needs independent attention staring from the multi-armed case, and subsequently to the linear contextual and the general case. Alternatively, one might need to prove an information theoretic lower bound that such a first order regret bound in the bandit case requires knowing $L^*$.
> >
> > **As we have highlighted in the significance response [(Link)](https://openreview.net/forum?id=SThJXvucjQ&noteId=i9bMk8ga90), we think our work provides a safe way for existing linear implementations of bandit algorithms to transition to neural models for better modeling, and would be of practical use to the community. As such a more positive evaluation from you would enable us to engage a wider audience and contribute more effectively to advancing practical applications.**
> >
> > Regarding the effective dimension comment, - yes, using online gradient descent let's us provide a bound that is independent of the dimension of the contexts or effective dimension, and the $\sqrt{K}$ dependence comes from the bandit to online regression reduction. As we have highlighted in Remark 3.2, for infinite arms, one can use a straightforward extension of our
> > results using the analysis of Theorem 1 in [Foster et al. (2020)](https://arxiv.org/abs/2107.05745) to give a regret that scales with
> > the dimension of the action space instead of $K$.
> >
> > We again thank the reviewer for their engament and all the valuable comments that have improved the quality of our draft.

---

### Official Review · Reviewer_BHSm · 2024-11-01

**Soundness:** 2
**Presentation:** 3
**Contribution:** 3
**Rating:** 6
**Confidence:** 3

**Summary:**

The paper studies the conservative contextual bandit problem, where the goal is to minimize the regret as in classical contextual bandit problems and, with high probability, be competitive against a baseline policy. Importantly, the cost functions are not assumed to be linear in the contexts. This extends previous study of conservative bandit problems beyond the multi-armed setting and the contextual linear setting.

The two proposed algorithms, C-SquareCB and C-FastCB, combine the inverse gap weighting based exploration and online regression oracles (e.g. function approximation using neural network+ gradient descent), and achieve regret which is sublinear in the horizon T or sublinear in the optimal loss L*, while being (1+alpha)-competitive against the baseline with high probability. Experiments using real world data show that C-SquareCB and C-FastCB have smaller regret compared to algorithms for contextual linear bandit problems, and are more competitive against the baselines.

**Strengths:**

The paper is overall well written, with clearly presented problem formulations, algorithms and results. The setup considered fills in the gap of current conservative bandit literature, and the proposed algorithms have provably sublinear (in T or L*) regret while being (1+alpha) competitive against the baseline. Experiments further demonstrate their superior performance as compared to algorithms designed for conservative linear contextual bandits and for classical settings without baselines.

**Weaknesses:**

The proofs/assumptions may lack rigor. In particular, the proofs cite results in other works without carefully checking the assumptions under which those results hold. For instance, in line 914-915, lemma 2 in Foster & Rakhlin (2020) is invoked. If my understanding is correct, that lemma requires Assumption 3 to hold for all possible sequences. Nevertheless, in line 1736-1737, Assumption 3 is only proved to hold with high probability.

One contribution of the work is to use neural network for function approximation to deal with the non-linearity in the cost functions. However, to achieve the stated performance as in Theorem 5.2, the width (and thus the number of parameters) of the neural network is Omega(poly(T)), which might be too large for long-horizon problems, making the algorithms less practical.

The paper could benefit from more detailed discussion on the significance of the regret bounds in Theorem 3.1 and Theorem 4.1. In particular, it appears that for any algorithm which has regret upper bounded by alpha * y_l * t, under Assumption 2, equation (2) is automatically satisfied. Some discussions on the range of parameters might help the readers better understand the significance of the results.

**Questions:**

In Definition 2.2, line 124, alpha is chosen to be <1. I’m wondering if this is just a simplifying assumption, or there is difficulty in extending the results of this paper to settings where alpha>=1?

In line 160, what is H in the ``inf’’?

---

> ### Author Response · Authors · 2024-11-22
> **Response to questions by Reviewer BHSm**
>
> ___
> ___
> ## **_On the use of Assumption-3_**
> ___
> We thank the reviewer for checking the details of the proof. The reviewer raised a question about the use of a high probability version of Assumption 3 in the proof of Theorem 5.1. We clarify why this is rigorous.
>
> **We first note that  [Foster & Rakhlin (2020)](https://proceedings.mlr.press/v119/foster20a) provide their regret bounds in high probability over the randomness of the algorithm and make the squared loss regret assumption for all choice of sequences $\mathbf{x_{t,a}}, y_{t,a_t}, t\in [T], a\in[K]$. Our main results in Theorems 3.1 and 4.1 are of the same nature and do not make use of a high probability version of Assumption 3.**
>
> Next, in Theorem 5.1, we indeed show that Assumption 3 holds with high probability, where the randomness is with respect to the initialization and the perturbation of the neural network. Therefore, given a sequence, $\mathbf{x}_{t,a}$, our final regret bound holds in high probability with respect to the randomness of the network and the randomness of the algorithm.
>
> To see this more concretely, we first outline the specific part of the analysis in [Foster & Rakhlin (2020)](https://proceedings.mlr.press/v119/foster20a) that invokes the Assumption for all sequences. Here we shall use the notation from [Foster & Rakhlin (2020)](https://proceedings.mlr.press/v119/foster20a) for consistency. We refer the reviewer to the proof of Lemma 2 in  [Foster & Rakhlin (2020)](https://proceedings.mlr.press/v119/foster20a),  Appendix B, specifically to the part that says "Now, suppose Assumption 2a holds, so that SqAlg guarantees that with probability 1,"
> $$
> \sum_{t=1}^T ( \hat{y_t} (x_t, a_t) - \ell_t(a_t) )^2 - \sum_{t=1}^T \left( f^*(x_t, a_t) - \ell_t(a_t) \right)^2  \leq Reg_{Sq}(T).
> $$
> In our analysis, this would hold in high probability, i.e., with probability $1-\delta_1$, for some $\delta_1>0$ (this randomness is over the initialization and the perturbation of the network). Subsequently, we invoke the Freedman's inequality (Lemma 1 in [Foster & Rakhlin (2020) ](https://proceedings.mlr.press/v119/foster20a) that holds with probability $(1-\delta_2)$, for some $\delta_2>0$, and take a union bound of both the high probability events to conclude that with probability $(1-(\delta_1+\delta_2))$
> $$
> \sum_{t=1}^T \sum_{a \in \mathcal{A}} p_{t,a} \left( \hat{y_t}(x_t, a_t) - f^*(x_t, a_t) \right)^2
>         \leq 2 Reg_{Sq}(T)(T) + 16 \log(\delta_2^{-1}).
> $$
> Note that the $1-\delta_2$ high probability event is with respect to the randomness of the arm selection by IGW. Thereafter, the analysis follows as in [Foster & Rakhlin (2020)](https://proceedings.mlr.press/v119/foster20a). Therefore, for any sequence of contexts and costs, our regret bound holds in high probability over the randomness of initialization and the perturbation of the network and the randomness of the arm choices. In the theorem statement, $1-O(\delta)$ corresponds to $1-(\delta_1 + \delta_2)$ as in the above argument, where $O$ only hides global constants.
>
> We have updated the statement of the theorem to specify that the randomness is with respect to both the algorithm and the randomness of the neural network. We have also added more details in the proof to make this part of the analysis more transparent, and once again appreciate the clarification question from the reviewer.
> ___

---

> > ### Author Response · Authors · 2024-11-22
> > **Response to questions by Reviewer BHSm (cont.)**
> >
> > ___
> > ___
> > ## **_Width Requirements:_**
> > ___
> > The reviewer raises questions about the width requirements and the practicality of the algorithms for long horizon problems. We respond to this as follows.
> >
> > - **We first note that our main regret bounds in Theorems 3.1 and 4.1 are for general (potentially non-linear) cost functions as in [Foster & Rakhlin (2020)](https://proceedings.mlr.press/v119/foster20a); [Foster & Krishnamurthy (2021)](https://proceedings.neurips.cc/paper_files/paper/2021/hash/9d684c589d67031a627ad33d59db65e5-Abstract.html); [Simchi-Levi & Xu (2020)](https://arxiv.org/abs/2003.12699), and do not assume anything about the width of a network.**
> >
> > - For Theorems 5.1 and 5.2, we use one specific instance of an online regression oracle, namely Online Gradient Descent with overparameterized neural networks. Note that the width requirements in the theorem statements are sufficient conditions, but not necessary. Therefore to address the reviewers' concern about practicality of the algorithms, we provide additional evidence in Appendix F.2 that shows the performance of the algorithms for different choices of the width of the network. Figures 4 and 5 show that for practical purposes, fixed width networks suffice for both Algorithms 1 and 2.
> >
> > - Further we also highlight that when using neural networks, all bandit algorithms assume a large width to give provable regret bounds. While our work requires $m = \Omega(T^5)$, other works require far larger width. For example,
> > 1. Neural Thompson Sampling [(Zhang et al. (2021))](https://openreview.net/forum?id=tkAtoZkcUnm) assumes  $m (\log m)^{-3} \geq T^{13}$ (see Condition 4.1 in their paper).
> > 2. Neural UCB [(Zhou et al. (2020))](https://proceedings.mlr.press/v119/zhou20a) assumes $m \geq T^{16}$. Theorem 4.5's regret bound states that the width must be $\tilde{\Omega}(\text{poly}(T))$, and from the proof, specifically the final term $m^{-1/6}\sqrt{\log m} T^{8/3} \lambda^{-2/3}L^3 \leq 1$, it follows that $m$ must be at least $T^{16}$.
> > 3. EE-Net [(Ban et al., 2022)](https://openreview.net/forum?id=X_ch3VrNSRg) assumes $m \geq T^{30}$. From the proof of Theorem 1, specifically the term $\xi_1 = \Theta(\frac{t^4}{m^{1/6}})$ in Eq. (C.4) and its sum $\Theta(\frac{T^5}{m^{1/6}})$ over $t=1$ to $T$, it follows that they need $m^{1/6} \geq (T^5)$ or $m$ must be at least $T^{30}$ to make this term $\mathcal{O}(1)$.
> > ___
> > ___
> > ## **_Trivial policy:_**
> > ___
> > Indeed any policy that has regret upper bounded by $\alpha y_{l} t$, as suggested by the reviewer would satisfy the performance constraint in (2). However, **the regret of this policy is not $\tilde{\mathcal{O}}(\sqrt{T})$**. In fact the trivial policy that always selects the baseline action also satisfies (2) but does not guarantee the desired regret bound.
> >
> > The objective in conservative bandits are two fold: (1) satisfy the performance constraint (2) generate a good policy with sub-linear regret. Our construction in Algorithm 1 ensures that the generated policy is good, i.e., has a regret bound of $\tilde{\mathcal{O}}(\sqrt{T})$ while simultaneously satisfying the performance constraint. Further in Algorithm~2, the generated policy has an even tighter regret bound of $\tilde{\mathcal{O}}(\sqrt{L^*})$ while maintaining the performance constraint.
> > ___
> > ___
> > ## **_Assumption $\alpha < 1$:_**
> > ___
> > Our analysis does not depend on $\alpha < 1$ and works even for $\alpha \geq 1$. Note that the regret in Theorems 3.1 and 4.1 depend on $1/\alpha$, and therefore, it only decreases with increasing $\alpha$. We have updated the range of $\alpha$ in the current draft.
> > ___
> > ___
> > ## **_Definition of $\mathcal{H}$:_**
> > ___
> > $\mathcal{H}$ refers to the function class to which $h$ belongs. We have specified this in the updated draft. Our regret defined in (3) is competitive against the $\inf$ in $\mathcal{H}$, and therefore, is immediately competitive against $h$.

---

> > > ### Author Response · Authors · 2024-11-24
> > >
> > > Dear reviewer, we again thank you for taking the time to review our paper. Since the end of the discussion period is approaching, we request you to kindly check our responses to your questions. We would be happy to provide any further clarifications you need.
> > >
> > > We have also provided a brief response higlighting the significance of this work to the broader community [(Link)](https://openreview.net/forum?id=SThJXvucjQ&noteId=i9bMk8ga90) and would greatly appreciate you reading it.

---

> > > > ### Comment · Reviewer_BHSm · 2024-11-24
> > > > **response from reviewer BHSm**
> > > >
> > > > Thank the authors for the detailed answers and updates to the paper, which have addressed most of my questions, and I've raised the score to 6. However, I'm still a bit concerned about the trivial policies: consider a policy which IGNORES the performance constraints and has $O(\sqrt{T})$ regret. For this policy, the regret is sublinear, and so the performance constraints are automatically satisfied for large $T$ if $\alpha y_l$ does not depend on $T$. In this case, the performance constraints seem to be redundant constraints, and the problem degenerates to a classical contextual bandit problem.

---

> > > > > ### Author Response · Authors · 2024-11-25
> > > > > **Response to further clarification question**
> > > > >
> > > > > Dear Reviewer, we thank you for looking at our responses and the clarification question. We respond to it as follows.
> > > > >
> > > > > - Note that the performance constraint is for every $t \in [T]$. Even if the policy in the long run is good, it would deviate from the existing baseline in order to explore the environment, incurring high losses during the exploration phase, which might not be acceptable in many applications.
> > > > > - For the multi-armed and linear case existing methods [[Wu et al. (2016)]](https://arxiv.org/abs/1602.04282) [[Kazerouni et al. (2017)]](https://arxiv.org/abs/1611.06426) build confidence sets around estimates to set up the safety condition while we use the properties of the sampling algorithm and a reduction to online regression.
> > > > > - In essence this specific formulation gives agents a way to conservatively explore the environment to find a better policy while not incurring huge losses when exploring, when compared to the existing policy in use. Take the example we described in our general response - a lot of deployed bandit algorithms (in advertisement and recommendation systems) use linear reward modeling with high dimensional sparse contexts. Our algorithms provide a way to transition to neural models for better modeling while at the same time ensuring that when the neural model is exploring the environment to find a superior policy, the performance does not significantly decline at every step, compared to existing linear baselines.
> > > > >
> > > > > Please let us know if you have any other questions.

---

> > > > > > ### Comment · Reviewer_BHSm · 2024-11-25
> > > > > >
> > > > > > Thank you for the response and clarification. I will maintain my score (supporting acceptance).

---

> ### Author Response · Authors · 2024-12-03
>
> Dear reviewer, we again sincerely thank you for your engagement and the thoughtful comments to help increase the quality of our draft. We bring a few points to your attention to further help you evaluate our work.
>
> - As requested by Reviewer wAuW we have further extended our algorithm to the case when the true baseline cost is not known, but rather a noisy version is recieved.
>
> - As requested by Reviewer wAuW and 2J6i we have provided empirical evaluation for tuning the exploration parameter $\gamma_t$ and showed that replacing $\sum_{i=1}^{t} L_i^*$ by $\sum_{i=1}^{t-1} L_i$, the loss of the current policy produces good results in the majority of environments (see Appendix F). This is practical since $\sum_{i=1}^{t-1} L_i$ is known to the learner.
>
> - **As we have highlighted in the significance response [(Link)](https://openreview.net/forum?id=SThJXvucjQ&noteId=i9bMk8ga90), we think our work provides a safe way for existing linear implementations of bandit algorithms to transition to neural models for better modeling, and would be of practical use to the community. As such a more positive evaluation from you would enable us to engage a wider audience and contribute more effectively to advancing practical applications.**

---

### Official Review · Reviewer_2J6i · 2024-11-04

**Soundness:** 4
**Presentation:** 3
**Contribution:** 3
**Rating:** 8
**Confidence:** 3

**Summary:**

This authors consider Conservative Contextual Bandits with general value function class and develop two algorithms C-SquareCB and C-FastCB, using Inverse Gap Weighting and online regression oracles. They show that the safety constraint is satisfied with high probability (the performance is not worse than a baseline policy by more than $(1+\alpha)$ factor). They also show the regret for C-SquareCB is $\tilde O(\sqrt{KT}+K/\alpha)$ and the regret for C-FastCB is $\tilde O(\sqrt{KL^*}+K/\alpha)$, where $L^*$ is the cumulative loss of the optimal
policy. The efficacy of the proposed algorithms is validated on real-world data.

**Strengths:**

- The paper is clearly written and mostly easy to follow, with proof roadmap and intuition moderately provided.
- The provided solution nicely connects safe conservative bandits and contextual bandits with general functions class.
- Analysis is sound and rigorous. Numerical experiment is convincing.

**Weaknesses:**

As a paper that combines two established sub-fields in bandits, it is a bit unclear the novelty in algorithmic design and theoretical analysis. I would like to see authors provide and emphasize more detailed discussions if possible. In particular, what is your technical/methodological contribution compared to Kazerouni et al. (2017), Foster & Rakhlin (2020), Foster & Krishnamurthy (2021)? What are the challenges of adapting/extending their tools? From what I understand, the novelty appears in: a delicate safety criterion and a time-varying exploration schedule. The authors can add more if necessary.

**Questions:**

Some clarification questions apart from the concern in Weaknesses:

- Data-dependent bound
  - When $L^*=O(1)$, it seems the regret bound becomes $\tilde O(\ln T)$. Is there any instance-dependent metric (similar to the mean gap in the stochastic MAB setting) hiding in the constant term?
  - $\gamma_t$ seems to be requiring the knowledge of not only $L^*$ but also each component that sums up to $L^*$. Would that be too strong?

- Experiments
  - Can you provide more details on how you set $\delta$ and tune hyperparameters? How do you run C-FastCB if the optimal cost is not known?

- Typos:
  - Algorithm 2 Line 8: Is there a term related with $\sqrt{\log\delta^{-1}}$ missing?
  - Line 836 missing a sqrt term consisting of $m_{\tau-1}$ and $\texttt{Reg}$.

---

> ### Author Response · Authors · 2024-11-22
> **Response to questions by Reviewer 2J6i**
>
> ---
> ---
> ## **_Details of Technical Contribution_**
> ___
> We summarize our technical and algorithmic contributions below. We have also added two remarks in the updated draft detailing these. \
> \
> **1. Reduction using Squared loss (C-SquareCB):**
>   - The analysis in [Foster & Rakhlin (2020)](https://proceedings.mlr.press/v119/foster20a) does not have a safety condition and therefore our analysis bounding $n_T$ (the number of times the baseline action is played in Step 2) and the performance constraint satisfaction (in Step 4) of proof of Theorem 3.1 are original contributions.
>   - Note that unlike in the linear case [Kazerouni et al. (2017)](https://arxiv.org/abs/1611.06426), we cannot maintain high probability confidence bounds around parameter estimates in the general case. Therefore in our analysis we crucially relate $n_T$ to squared loss and thereafter give a reduction to online regression. We emphasize that such a reduction does not exist in the literature. Further our analysis in Step 4 leads to the construction of the safety condition in (4), that maintains the performance guarantee in high probability.
>   - Finally even the analysis from [Foster & Rakhlin (2020)](https://proceedings.mlr.press/v119/foster20a) cannot be directly used to bound the regret for the time steps when the IGW actions were picked (term $I$ in eq (5) of the draft). This is because we need to carefully choose a time dependent exploration parameter $\gamma_t$, to simultaneously ensure that term $I$ is $\mathcal{O}({\sqrt{T}})$ while ensuring that $n_T$ is small. Therefore in the process, we extend the analysis in [Foster & Rakhlin (2020)](https://proceedings.mlr.press/v119/foster20a) to time-dependent $\gamma_t$ and bound the regret in term $I$.
>
>
> **2. Reduction using KL loss (C-FastCB)**
>   - For the first-order regret bound in Theorem 4.1, we face similar challenges as above. Note that a UCB based analysis for conservative bandits ensuring a first-order regret bound does not exist even in the linear case. Therefore, in our analysis of Theorem 4.1 we show a reduction of $n_T$ to online regression with KL loss and ensure that the performance constraint is satisfied by modifying the safety condition.
>   - Note that in [Foster & Krishnamurthy (2021)](https://proceedings.neurips.cc/paper_files/paper/2021/hash/9d684c589d67031a627ad33d59db65e5-Abstract.html) , the exploration parameter $\gamma_t$ is set to a fixed value $\gamma$ but as in the previous case, in our analysis we need a time dependent $\gamma_t$ to ensure that we can bound the regret contributed by both the IGW and baseline actions. *However, unlike the previous case, we crucially need to set $\gamma_t$ in an episodic manner to ensure that the final regret does not have a $\sqrt{T}$ dependence.* We construct $\log (L^*)$ such episodes, keep $\gamma_t$ constant within an episode and modify our analysis in Step 2 to bound $n_T$. Further we extend the analysis in [Foster & Krishnamurthy (2021)](https://proceedings.neurips.cc/paper_files/paper/2021/hash/9d684c589d67031a627ad33d59db65e5-Abstract.html)  with such an episodic schedule of $\gamma_t$ to bound term $I$ and show that it has only an additional $\sqrt{\log(L^*)}$ factor.
>
> **3. Regret Bounds with Neural Networks**
> - Finally to give complete regret bounds, we instantiate the online regression oracle with Online Gradient Descent (OGD) and the function approximator with a feed-forward neural network to give an end-to-end regret bound of $O\big(\sqrt{KT \log (T)} + K \log(T)/\alpha\big)$ for Algorithm 1 (Theorem 5.1) and $O\big(\sqrt{KL^* \log (L^*) \log(T)} + K\log T + K \log(T)/\alpha\big)$ for Algorithm 2 (Theorem 5.2). We also compare our proposed algorithms with existing baselines for conservative bandits and show that our algorithms consistently perform better on real world datasets.
>
> ---
> ---
>
> ## **_Data Dependent Bound_**
> ___
> - **Dependence on Instance Dependent Metrics:** No, the first order regret bound does not depend on any other instance dependent metric.
> - **Choice of $\gamma_i$:** Note that the dependence of $\gamma_i$ on $L_i^*$ is to provably guarantee a sub-linear regret. We first emphasize that even in the unconstrained case, [Foster & Krishnamurthy (2021)](https://proceedings.neurips.cc/paper_files/paper/2021/hash/9d684c589d67031a627ad33d59db65e5-Abstract.html)  also set $\gamma$ using $L^*$ to give a provable first order regret bound. Further, even in the full information setting, with exponential weights algorithm [Freund & Schapire (1997)](https://www.sciencedirect.com/science/article/pii/S002200009791504X)  one needs to know $L^*$ to set the exploration parameter to get a *first-order bound*. Specifically regarding the $i$ dependence, our empirical study shows that one can work with a constant $\gamma$, or use a heuristic schedule and still ensure very low regret (see next response).
> ---

---

> > ### Author Response · Authors · 2024-11-22
> > **Response to questions by Reviewer 2J6i (cont.)**
> >
> > ---
> > ---
> > ## **_Experiment Details_**
> > ___
> > **Tuning of $\gamma_i$:** Since the optimal loss $L_i^*$ is not known in advance, the exploration parameter $\gamma_i$ is treated as a hyper-parameter in our experiments. A heuristic choice is to substitute $\sum_{i=1}^t L_i^*$ by the sum of the observed losses until time $t-1$, i.e., $\sum_{i=1}^{t-1}L_i$ to choose $\gamma_t$. Note that at time $t$, $\sum_{i=1}^{t-1}L_i$ is known to the user. The other choice is to treat $\gamma_i$ as a single parameter $\gamma$ and tune it for different values. In our experiments we tune $\gamma$ in {10,20,50,100,200,500,1000}. We have plotted the corresponding cumulative regret for all these choices in Figure~3 (Appendix F) and we note that the heuristic choice of $\sum_{i=1}^{t-1}L_i$ produces good results in the majority of environments.
> >
> > **Choice of $\delta$:** The failure probability $\delta$ is set by the user and is not tuned as a hyper-parameter. A cautious user might want to set a very small value of $\delta$ to ensure that the performance constraint is satisfied with very high probability. In our experiments, $\delta$ is set to the value of $0.1$.
> > ___
> > ___
> > ## **_Minor Typos_**
> > ___
> > We thank the reviewer for reading the proofs in detail and pointing out the two typos. We have fixed them in the updated version.

---

> > > ### Author Response · Authors · 2024-11-24
> > >
> > > Dear reviewer, we again thank you for taking the time to review our paper. Since the end of the discussion period is approaching, we request you to kindly check our responses to your questions. We would be happy to provide any further clarifications you need.
> > >
> > > We have also provided a brief response higlighting the significance of this work to the broader community [(Link)](https://openreview.net/forum?id=SThJXvucjQ&noteId=i9bMk8ga90) and would greatly appreciate you reading it.

---

> ### Comment · Reviewer_2J6i · 2024-11-25
>
> I would like to thank the authors for the detailed response that have addressed my questions. I have raised the score to 8. Please include your response accordingly in your updated version to particularly highlight your technical contributions as well as experimental details.

---

> > ### Author Response · Authors · 2024-11-25
> >
> > We sincerely thank the reviewer for their feedback. We have already included our response to the draft, and will further update them to reflect all clarifications, post the discussion phase.

---

### Author Response · Authors · 2024-11-22
**General Response**

We sincerely thank all the reviewers for their valuable and thoughtful feedback on our submission. We are gald that the reviewers found the paper **clearly written**, **easy to follow** and **clearly presented formulation, algorithms and results** (Reviewers 2J6i, BHSm), and that they believe that our work **fills in the current gaps** in conservative bandits (BHSm), which applies **only to linear reward models** (wAuW), by making it **work for general function classes** (Reviewers 2J6i, BHSm). We are encouraged that they find the **analysis sound and rigorous** (2J6i) and that they **appear to be consistent** (wAuW). Finally we are also glad that they find our **experiments convincing** (2J6i), that they **demonstrate superior performance** (BHSm) on **real world data** (wAuW).


The reviewers have posed several insightful questions and offered valuable suggestions to improve the quality of the current draft. We have incorporated all the suggestions in our modified draft and we address each question in our individual responses. We have highlighted the incorporated changes in color in the draft of main paper. Below we specifically discuss the significance of our algorithm and results for the broader community. For a discussion on our novel technical contributions see [**[Link]**](https://openreview.net/forum?id=SThJXvucjQ&noteId=c9tiwWyZ37).
___
___
### **_Significance:_**
___
- Safety is a crucial consideration that significantly enhances the practical use of bandit algorithms in the real-world. Our study ensures **safety with respect to a baseline**, meaning the performance is not worse than a baseline policy (e.g., the policy that the company has in production) by more than $(1+\alpha)$ factor.  A lot of deployed bandit algorithms (in advertisement and recommendation systems) use linear reward modelling with high dimensional sparse contexts. **Our algorithms provide a way to transition to neural models for better modelling while at the same time ensuring that the performance does not significantly decline compared to existing linear baselines.**

- Since our algorithms work for general cost functions, they are very broadly applicable to real-world datasets. Further our second algorithm simultaneously provides a very tight first order regret while maintaining the safety condition.

- We combine both our algorithms with neural networks, give provable regret bounds and provide empirical evaluation on several real-world datasets. We show that they indeed maintain low regret while not violating the safety constraint (see Figures 1 and 2 in the draft). Further we do extensive hyper-parameter tuning (see Appendix F) and show that the performance of the algorithms are far superior over their linear counterpart.

---

### Meta-Review · Area_Chair_5PGK · 2024-12-22

**Metareview:**

This paper examines the conservative contextual bandit problem, including nonlinear cost functions, and proposes an algorithm that utilizes a regression oracle satisfying certain assumptions. The authors provide guarantees for the regret upper bound of the proposed algorithm and the probability of violating the safety constraint. Its effectiveness is demonstrated through numerical experiments. The algorithms and results are clearly written, and the extension to arbitrary function classes for the cost function is a significant contribution.

On the other hand, there are parts where the proofs and assumptions lack rigor, as well as weaknesses in technical novelty. However, most of these concerns were resolved through discussions with the reviewers, and the authors have promised to incorporate these revisions. Under the premise that these issues will be adequately addressed in the final version, I support the acceptance of this paper.


----
Additional Comments:

Based on the authors’ comments, it seems that some related works have been overlooked. I would like to share the following references as helpful information.
> Regarding unknown $L^*$, we have thoroughly reviewed the literature, and to the best of our knowledge, for the bandit information case, even in the multi-armed case, existing algorithms: EXP3LIGHT [Stoltz (2005)], GREEN [Allenberg et al. (2006)] need $L^*$ to prove a first order regret bound. We will add this discussion to the final draft. However, as you said, this is not the point of focus for this work, and we think that this problem needs independent attention staring from the multi-armed case, and subsequently to the linear contextual and the general case. Alternatively, one might need to prove an information theoretic lower bound that such a first order regret bound in the bandit case requires knowing $L^*$.

For several bandit problems, algorithms that achieve bounds dependent on $L^*$ without prior knowledge of $L^*$ are known. For multi-armed bandit problems, refer to Exercise 28.14 in [M1], [M2], or the blog article [M3]. Similar results are also known for combinatorial semi-bandit problems [M4], linear bandits [M5], and contextual linear bandits [M6].

These results are typically achieved through approaches like the doubling trick or adaptive learning rates. However, it is not trivial to apply these approaches to the problem setting of this paper. Therefore, the fact that the proposed algorithm is not $L^*$-agnostic should not be considered a reason for rejection. I hope the authors will clarify their descriptions regarding prior work and remaining challenges to avoid potential misunderstandings.

* [M1]: Lattimore, T., & Szepesvári, C. (2020). Bandit algorithms. Cambridge University Press.
* [M2]: Wei, C. Y., & Luo, H. (2018). More adaptive algorithms for adversarial bandits. In Conference On Learning Theory (pp. 1263-1291). PMLR.
* [M3]: [First order bounds for k-armed adversarial bandits](https://banditalgs.com/2019/02/16/first-order-bounds/)
* [M4]: Tsuchiya, T., Ito, S., & Honda, J. (2023). Further adaptive best-of-both-worlds algorithm for combinatorial semi-bandits. In International Conference on Artificial Intelligence and Statistics (pp. 8117-8144). PMLR.
* [M5]: Ito, S., Hirahara, S., Soma, T., & Yoshida, Y. (2020). Tight first-and second-order regret bounds for adversarial linear bandits. Advances in Neural Information Processing Systems, 33, 2028-2038.
* [M6]: Olkhovskaya, J., Mayo, J., van Erven, T., Neu, G., & Wei, C. Y. (2024). First-and second-order bounds for adversarial linear contextual bandits. Advances in Neural Information Processing Systems, 36.

**Additional Comments On Reviewer Discussion:**

The reviewers expressed concerns about the lack of rigor in certain proofs and assumptions, as well as the technical novelty of the work. However, most of these concerns were resolved during discussions with the reviewers, and the authors have committed to incorporating these revisions. Under the assumption that these issues will be adequately addressed in the final version, I support the acceptance of this paper.

---

### Decision · Program_Chairs · 2025-01-22

Accept (Poster)